# Therapy-induced secretion of spliceosomal components mediates pro-survival crosstalk between ovarian cancer cells

Ovarian cancer often develops resistance to conventional therapies, hampering their effectiveness. Here, using ex vivo paired ovarian cancer ascites obtained before and after chemotherapy and in vitro therapy-induced secretomes, we show that molecules secreted by ovarian cancer cells upon therapy promote cisplatin resistance and enhance DNA damage repair in recipient cancer cells. Even a short-term incubation of chemonaive ovarian cancer cells with therapy-induced secretomes induces changes resembling those that are observed in chemoresistant patient-derived tumor cells after long-term therapy. Using integrative omics techniques, we find that both ex vivo and in vitro therapy-induced secretomes are enriched with spliceosomal components, which relocalize from the nucleus to the cytoplasm and subsequently into the extracellular vesicles upon treatment. We demonstrate that these molecules substantially contribute to the phenotypic effects of therapy-induced secretomes. Thus, SNU13 and SYNCRIP spliceosomal proteins promote therapy resistance, while the exogenous U12 and U6atac snRNAs stimulate tumor growth. These findings demonstrate the significance of spliceosomal network perturbation during therapy and further highlight that extracellular signaling might be a key factor contributing to the emergence of ovarian cancer therapy resistance.

Despite recent therapeutic advancements, ovarian cancer continues to exhibit the highest mortality rate among gynecological malignancies. In most cases, ovarian cancer is managed through repeated cycles of platinum-based chemotherapy. Although tumors initially respond to this treatment, they inevitably develop drug resistance, leading to relapse in approximately 80% of patients within 6-12 months after chemotherapy[1]. The acquisition of drug resistance during therapy has long been viewed as a process entirely confined to the molecular events occurring within cancer cells. Indeed, some tumor cells manage to evade therapeutic compounds by stimulation of drug efflux (owing to enhanced expression of multidrug resistance genes: *MDR1*, *MRP1*, *BCRP*, etc.), downregulation of apoptotic pathways (through de novo mutations in tumor suppressor genes like *TP53*, *BRCA1/2*, *ATM*), modification of targeted proteins or activation of alternative signaling pathways[2–4]. However, recent data suggest an interesting scenario where the cancer cell death induced by initial therapy promotes subsequent expansion of the survived clones[5–7].

Ovarian tumors exhibit significant intercellular heterogeneity in response to drugs, with most cancer cells succumbing to therapeutic pressure while some persistent cells survive drug exposure[6,8–14]. Single-cell RNA sequencing data has unveiled that these resistant subpopulations of ovarian cancer cells either exist at the beginning of tumorigenesis and expand after chemotherapy[15,16] or are generated during chemotherapy[12]. However, the precise mechanism driving the accelerated expansion of therapy-resistant clones remains elusive.

During chemotherapy, dying tumor cells become integral to the tumor microenvironment, releasing molecules that stimulate the proliferation of the remaining cancer cells[5,7,17,18]. This phenomenon,

✉ e-mail: victoria.shender@gmail.com

which ensures rapid tumor repopulation after initial treatment response, has been recognized in medical practice but is poorly studied at the molecular level. Understanding the nature of these extracellular signaling networks and developing approaches to inhibit them could shed light on the mechanism behind tumor cell adaptation to treatment and enhance the efficacy of standard chemotherapy. However, most studies have primarily focused on identifying signaling cascades within cancer cells that are activated in response to therapy-induced secretomes[5,17,18] or conducting proteomic analyses of secretomes without evaluating their biological significance[19,20]. Furthermore, the analysis of tumor secretomes is usually performed using cell cultures or animal models[5,7,17,21], while research on patient-derived tumor secretomes ex vivo faces challenges due to limited access to suitable biological material. Notably, ovarian carcinomas provide a unique opportunity for such investigations[22]. Advanced ovarian cancer is often accompanied by ascites, excessive fluid accumulation in the abdominal cavity[23], and routine medical care involves ascites drainage. Chemotherapy have been observed to reduce ascites volume, reaching up to an 80% reduction, with regression defined as an ascites volume below 500 ml. However, cases of intractable ascites may persist, particularly in patients with chemoresistant or recurrent disease, indicating a challenging clinical scenario with an associated poor prognosis[24–26]. Within cancer ascites, tumor cells, along with tumor-associated immune and stromal cells, create a unique microenvironment that fosters cancer progression, chemoresistance development, and immune system suppression[27].

In this work, we conducted a comprehensive analysis of changes in ovarian cancer cell secretomes induced by chemotherapy ex vivo and in vitro. Our findings unveil a molecular mechanism contributing to the acquisition of therapy resistance in cancer cells.

## Results

### Ovarian cancer ascites after chemotherapy contributes to de novo therapy resistance

To explore alterations in ovarian cancer cells and their microenvironment in response to neoadjuvant chemotherapy, we obtained a collection of paired ascitic fluids from the same ovarian cancer patients before and after chemotherapy (Supplementary Data 1, Sheets 1-2). We separated ascites into extracellular fluid and cancer cells, using the latter to establish primary cultures of ovarian cancer cells (Fig. 1A, left panel). Established primary cultures were stained for ovarian cancer markers CA125, EpCam, and CD44 (Supplementary Fig. 1A). We assessed the sensitivity of these low-passage cultures to cisplatin, one of the main drugs in ovarian cancer treatment. As expected, our results demonstrated that tumor primary cultures obtained after chemotherapy exhibited significantly higher resistance to cisplatin compared to their chemotherapy-naive counterparts (Fig. 1B; Supplementary Fig. 1B).

Previous studies have convincingly shown that tumor cells that die during therapy can reduce therapy efficacy by releasing various signaling molecules that accelerate the repopulation of residual tumor cells[5–7]. To investigate the role of this mechanism in ovarian cancer, we incubated chemotherapy-naive primary ovarian cancer cell cultures with autologous ascitic fluids (25% v/v) obtained from patients before and after chemotherapy. Then, we evaluated various parameters, including sensitivity to cisplatin treatment, cell migration, and whole transcriptome and proteome alterations (Fig. 1A, right panel). Our data revealed that ascitic fluids obtained after chemotherapy led to a four-fold increase in cisplatin resistance in cancer cells (Fig. 1D) and stimulated tumor cell motility (Fig. 1E) compared to chemonaive ascites. For a more precise characterization of effects of ovarian cancer ascites, we conducted RNAseq and proteomic analyses of several primary ovarian cancer cell cultures incubated with autologous pre- or post-chemotherapy ascitic fluids for three days (Supplementary Data 3). Among these cultures, the majority represented epithelial ovarian

cancer (EOC), while one represented neuroendocrine ovarian tumor (Supplementary Data 3, Sheet 2). Enrichment analysis of differentially expressed genes and proteins unveiled prominent upregulation DNA repair and cell cycle regulation pathways in tumor cells incubated with EOC ascitic fluids after chemotherapy (Fig. 1F, G). Consistent results were observed in the primary cell culture from neuroendocrine ovarian cancer (Supplementary Fig. 1C).

Thus, we demonstrated that ascitic fluids after chemotherapy contain components that substantially contribute to the development of a more aggressive cancer cell phenotype. To determine which ascitic fluid fraction is responsible for the observed effects, we fractionated post-chemotherapy ascitic fluid using ultrafiltration cartridges with molecular weight cut-offs of 3, 10, 30, and 100 kDa Supplementary Fig. 1D). Our data indicated that fraction greater than 100 kDa exhibited the most pronounced stimulation of chemoresistance, suggesting that the primary contributors to chemoresistance acquisition may be large protein complexes or components of extracellular vesicles circulating within ascitic fluids.

### Malignant ascitic fluids after chemotherapy are enriched with spliceosomal components

To determine signaling molecules contributing to therapy resistance development, we examined changes in cancer cell secretome in response to chemotherapy in a natural microenvironment. Employing LC-MS/MS, we scrutinized the proteomic profiles of paired ascitic fluids obtained from 10 ovarian cancer patients before and after chemotherapy (Fig. 2A; Supplementary Data 1, Sheets 1–2). Intriguingly, clustering of the proteomic data was not determined by individual patients but rather by the presence or absence of therapy (Fig. 2B; Supplementary Fig. 1E), suggesting that neoadjuvant chemotherapy induces significant changes in ascites composition, and these changes exhibit similarity across different patients.

In total, we detected 2,258 proteins within tumor ascites (Fig. 2C; Supplementary Data 4, Sheet 1). Enrichment analysis of differentially abundant proteins (at least 2-fold) revealed that ascites after chemotherapy are enriched with the clusters of ribosomal, spliceosomal, and proteasomal proteins (Supplementary Fig. 1F). As an additional control to exclude proteins not related to the therapy-induced secretomes of cancer cells we also compared the proteins identified in ovarian cancer ascites before and after chemotherapy with those previously found in cirrhosis ascites (non-malignant untreated control samples[22]) (Fig. 2C). Our analysis pinpointed 531 proteins that specifically emerged in tumor ascites after chemotherapy. Functional annotation via the KEGG and Gene Ontology databases revealed their predominant association with the pre-mRNA splicing process (Fig. 2C; Supplementary Fig. 1G).

Pre-mRNA splicing is orchestrated by the spliceosome, a dynamic ribonucleoprotein (RNP) complex comprising five uridine-rich non-coding small nuclear RNAs (U snRNAs) and up to 250 proteins[28,29]. To validate our proteomic data, we examined the presence of U snRNAs in ascitic fluids before and after chemotherapy using RT-qPCR analysis. Our findings demonstrated elevated levels of all spliceosomal U snRNAs, including those from the minor spliceosome, in ascitic fluids after chemotherapy (Fig. 2D), aligning with the proteomic profiling results. The presence of spliceosomal components in the extracellular space is of considerable interest since spliceosomal components typically reside intracellularly.

Intracellular proteins may enter the extracellular space either through active secretion or due to plasma membrane rupture upon cell death. In the latter case, these proteins are usually found as a mixture of partially degraded peptides[30,31]. To ascertain the integrity of proteins identified in ascitic fluids, we utilized our previously published peptidomic dataset[32] and assessed the ascitic fluid degradome. Our data revealed that the majority of peptides within ovarian cancer ascites after chemotherapy were related to ribosomal proteins

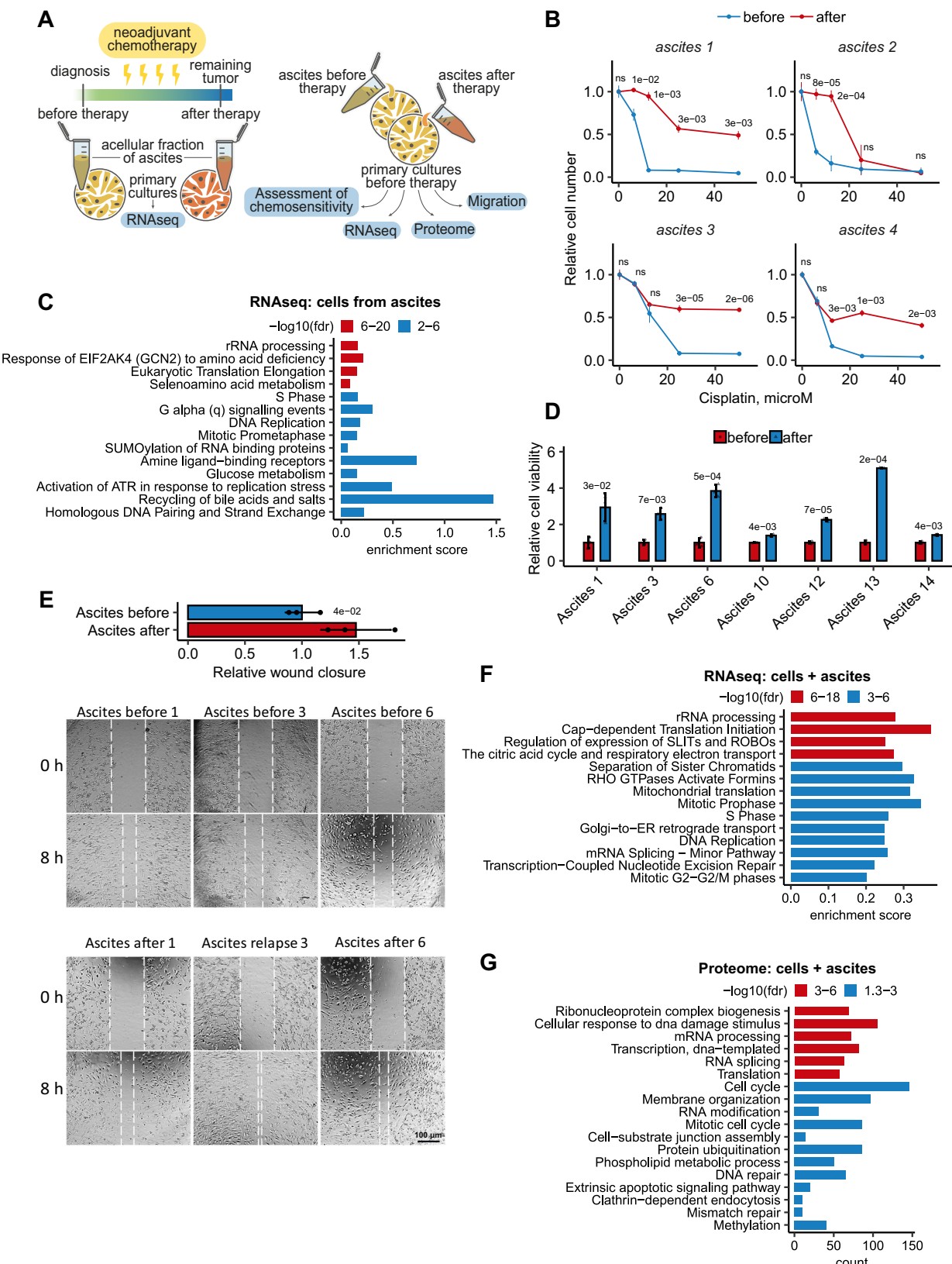

(Supplementary Fig. 1H). Importantly, we did not observe degradation products of spliceosomal proteins. This suggests that the enrichment of ascitic fluids with ribosomal proteins likely stems from cell destruction and the subsequent release of these proteins into the extracellular space. In contrast, the presence of spliceosomal proteins hints at active export from cells. Western blot analysis further confirmed the presence of several undegraded spliceosomal proteins in ascites after therapy (Supplementary Fig. 1I).

Active protein export can occur via transport through the cell membrane or encapsulation within extracellular vesicles[33]. To explore the mechanism of spliceosomal component export, we attempted to deplete U snRNPs from ascitic fluids using antibodies against specific

**Fig. 1 | Ovarian cancer ascites after chemotherapy contributes to de novo therapy resistance. A** Scheme showing the collection and processing of paired ascitic fluids isolated from patients before and after chemotherapy (left panel). Experimental workflow used to assess the effect of paired ascitic fluids before and after chemotherapy on phenotype and behavior of primary cultures of ovarian cancer cells (right panel). **B** In vitro viability assay of ovarian cancer cells isolated from paired ascites before (blue curves) and after (red curves) chemotherapy and subsequently treated with different concentrations of cisplatin. Dose-response curves of ovarian cancer cells were determined by an MTT assay on day 2 after cisplatin adding. The data represent the mean values ± SD (standard deviation) ($n = 3$ biologically independent experiments). **C** GSEA analysis of gene expression in ovarian cancer cells isolated from ascites after chemotherapy versus cells isolated from ascites before therapy. The X-axis represents GSEA enrichment score. **D** Primary cultures of ovarian cancer cells were pre-incubated for 3 days with autologous ascites before (blue bars) and after (red bars) chemotherapy, and then cancer cells were treated with cisplatin (10 μM). In vitro cell viability was assessed on day 2 after cisplatin adding using MTT assay. The data represent the mean values ± SD ($n = 3$ biologically independent experiments). **E** Wound healing assay of primary cultures of ovarian cancer cells that were pre-incubated for 3 days with autologous ascites before and after chemotherapy. The width of the wound area

was measured immediately after scratching (0 h) and the relative closure was measured after 8 h for three primary cultures of ovarian cancer cells. The bar graph illustrates wound closure, expressed as the fold change, denoting the ratio of mean values of wound widths between two states: cell cultures pre-treated with ascites after chemotherapy relative to cells pre-treated with ascites before chemotherapy. The data represent the mean values ± SD ($n = 3$ biologically independent experiments). **F** GSEA analysis of gene expression in ovarian cancer cells pre-incubated for 3 days with autologous ascites after chemotherapy versus ascites before therapy. The X-axis represents GSEA enrichment score (p-values are indicated by colors). **G** Gene Ontology enrichment analysis of proteins whose abundance increased at least 2 times in ovarian cancer cell cultures incubated for 3 days with autologous ascites after chemotherapy versus ascites before therapy. The X-axis represents the number of proteins associated with each pathway (p-values are indicated by colors). The p-value was obtained by two-tailed unpaired Student's t test (**B**, **D**, **E**). ClusterProfiler was used for functional enrichment analysis with all genes as background (**G**). Gene expression signature analysis was performed using the "signatureSearch" packages in "R" against the Reactome database (**C**, **F**). A hypergeometric test was carried out and all significant categories (false discovery rate < 0.05, after correction for multiple testing using the Benjamini–Hochberg procedure) are displayed. Source data are provided as a Source Data file.

---

spliceosomal proteins (U2AF1, SF3B1, PRPF8) or antibodies targeting CD63, an extracellular vesicle surface marker. Our results (Fig. 2E) demonstrated that only anti-CD63 antibodies efficiently removed U snRNPs from ascitic fluids, suggesting that spliceosomal proteins are primarily exported within extracellular vesicles. This finding aligns well with our earlier observation that the ascitic fluid fraction exceeding 100 kDa had the most pronounced biological effect (Supplementary Fig. 1D). To reinforce these findings, we investigated changes in extracellular vesicle concentrations in ascitic fluids before and after therapy using nanoparticle tracking analysis (NTA). The results demonstrated that the number of vesicles increased at least 2-fold after therapy (Fig. 2F; Supplementary Fig. 2A).

In summary, our findings demonstrate an augmentation of spliceosomal proteins and non-coding spliceosomal RNAs within ascites from ovarian cancer patients after chemotherapy (Fig. 2). These components are likely actively secreted by tumor cells within extracellular vesicles.

## Therapy-induced secretomes promote chemoresistance in ovarian cancer cells in vitro

Ascitic fluids contain a variety of cell types as well as extracellular components produced by these cells[34]. To explore whether cancer cells' signals play a role in mediating chemoresistance, we examined the impact of therapy-induced secretomes obtained in vitro from homogeneous ovarian cancer cell lines. We incubated recipient ovarian adenocarcinoma SKOV3 cells with secretomes from SKOV3 donor cells treated or untreated with cisplatin. In this experimental setup, SKOV3 donor cells were initially incubated with a cisplatin-containing medium for 7 h, then the cells were washed and incubated with a cisplatin-free medium. After 41 h, we collected the secretomes and added to recipient SKOV3 cells (Fig. 3A). Cisplatin concentration was chosen to cause the death of approximately 50% of donor cells at the time of secretome collection (Supplementary Fig. 2B, C). The absence of cisplatin in collected therapy-induced secretome (TIS) was confirmed by LC-MS/MS analysis (Supplementary Data 1, Sheet 4). Our findings revealed that pre-incubation of recipient SKOV3 cells with TIS significantly increased their resistance to subsequent cisplatin treatment (Fig. 3B). Moreover, this effect was accompanied by an increase in cell motility (Fig. 3C). Similar results were observed in the experiments with other ovarian cancer cell lines representing different tumor subtypes, including the serous ovarian cancer cell line (OVCAR3), ovarian cystadenocarcinoma cell line (MESOV) and clear cell ovarian cancer primary culture isolated from ascites (26 cells) (Fig. 3B; Supplementary Fig. 2D).

We then explored which pathways in donor cells might be responsible for the protective effect of their secretomes. To this end, we treated donor SKOV3 cells with cisplatin in the presence of various inhibitors: caspase inhibitor (Z-VAD-FMK; as cisplatin primarily activates caspase-dependent apoptosis[35]), inhibitor of vesicle formation (Brefeldin A; since we detected increased number of extracellular vesicles in ascites after therapy), or inhibitor of nuclear export (Leptomycin B; since we detected abundant nuclear proteins in ascitic fluids after therapy) (Supplementary Fig. 2E–G). Our results indicated that inhibition of apoptosis and vesicular transport in donor cells attenuated the effects of the secretomes (Fig. 3D), suggesting that protective components are secreted from apoptotic cells via vesicular transport.

Additionally, we examined secretomes from non-cancerous "normal" cells, such as the keratinocyte-derived epithelial cell line HaCaT, fallopian tube secretory epithelia cells (hTERT FT282), and primary culture of normal skin fibroblasts. We found that the corresponding TIS from hTERT FT282, HaCaT and fibroblasts had little or no effect on the cisplatin sensitivity of the parental cell lines (Fig. 3B) and slightly increased cisplatin resistance of ovarian cancer cells (Fig. 3E). Surprisingly, pre-incubation of normal fibroblasts and HaCaT cells with tumor cell secretomes reduced their resistance to cisplatin (Fig. 3F). This data suggests that TIS from cancer cells predominantly affect neighboring tumor cells but have limited influence on normal cells.

In summary, we demonstrated that therapy-induced secretomes obtained from ovarian cancer cells in vitro recapitulated effect of ex vivo ascitic fluids collected after the therapy. Importantly, in vitro experiments demonstrated that TIS from cancer cells can increase cisplatin resistance of ovarian cancer cells. This phenomenon is also observed in normal cells to a lesser extent, which could be attributed to differential extracellular vesicle uptake by normal and cancer cells[36] or the enhanced ability of cancer cells to adapt their phenotype in response to microenvironmental changes.

## In vitro therapy-induced secretomes of ovarian cancer cells are enriched with spliceosomal components

In order to elucidate the molecules that mediate the protective effects of therapy-induced secretomes (TIS)), we conducted a comprehensive analysis of the proteomic profiles within secretomes derived from various ovarian cancer cell lines, representing distinct tumor subtypes (SKOV3, MESOV, OVCAR3, and clear cell ovarian cancer primary culture) before and 48 h after cisplatin treatment. We also examined secretomes from normal fibroblasts, hTERT FT282 and HaCaT cells as non-cancerous controls (Fig. 3G; Supplementary Fig. 3A).

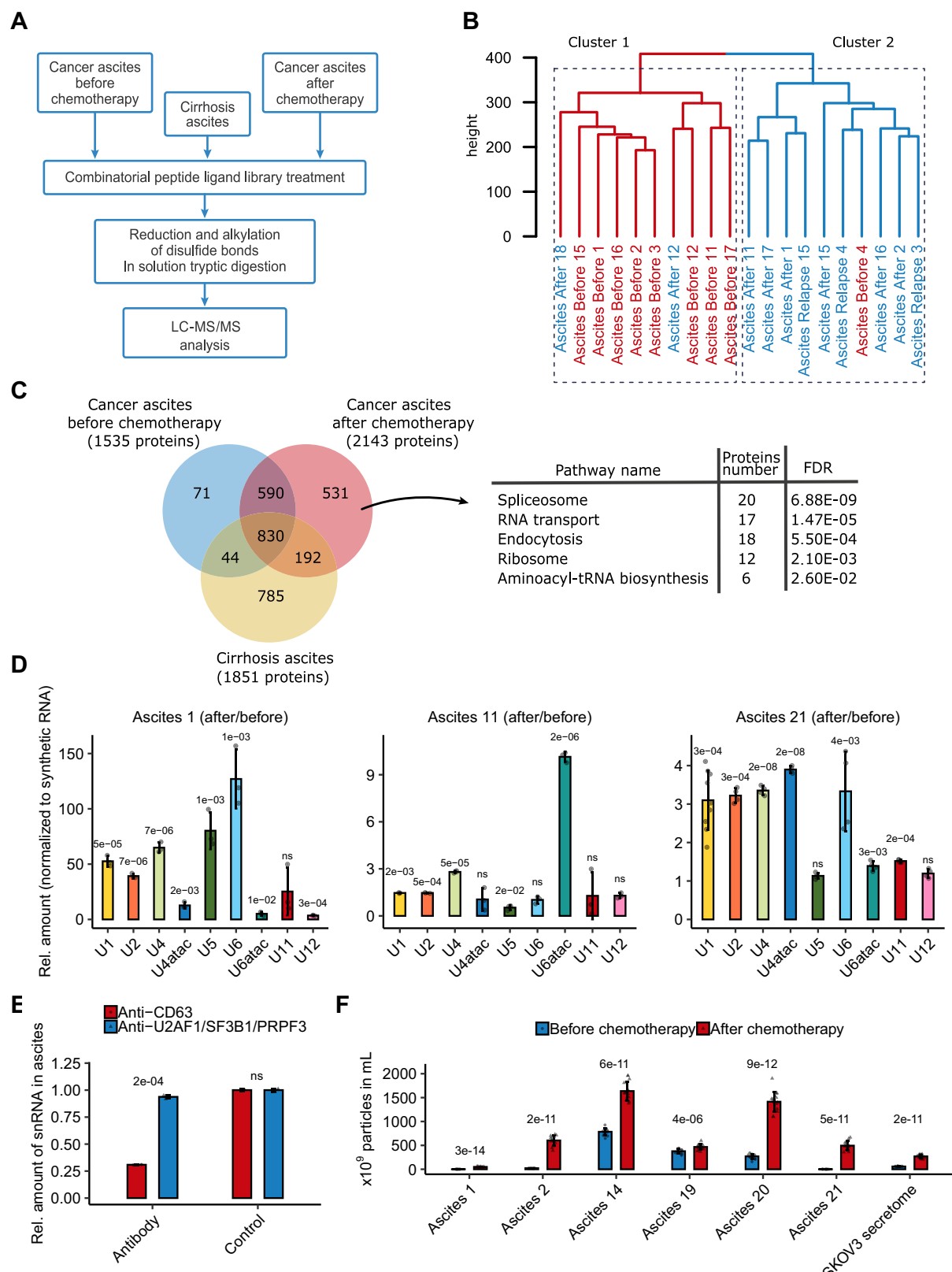

Our analysis revealed a higher diversity of proteins in the secretomes from cancer cell lines treated with cisplatin, compared to corresponding control secretomes from untreated cancer cells (Supplementary Fig. 3A; Supplementary Data 4, Sheets 2–3). These findings align with our earlier LC-MS/MS analysis of ascitic fluids (Fig. 2C). Enrichment analysis of proteins whose abundance increased

at least 2 times in therapy-induced secretomes from all cancer cell lines revealed a predominant association with splicing regulation (Fig. 3H; Supplementary Data 4, Sheet 4). The presence of various spliceosomal proteins was confirmed by western blotting (Supplementary Fig. 3B). Notably, we detected less number of spliceosomal proteins in TIS from HaCaT, hTERT FT282 and fibroblasts compared to TIS from cancer

**Fig. 2 | Malignant ascitic fluids after chemotherapy are enriched with spliceosomal components. A** Experimental scheme for proteomic analysis of the ascites samples. **B** Hierarchical clustering dendrogram of proteomic profiles of paired ascitic fluids before (red) and after (blue) chemotherapy. **C** Left panel: Venn diagram representing the proteins identified in ovarian cancer ascitic fluids before chemotherapy (blue circle), ovarian cancer ascitic fluids after chemotherapy (red circle), and cirrhosis ascitic fluids (yellow circle, according to our previously published data). Right panel: Results of the KEGG enrichment analysis of proteins identified only in ovarian cancer ascites after chemotherapy (p-values are indicated). **D** RT-qPCR analysis of 9 spliceosomal snRNAs in paired ascitic fluids before and after chemotherapy. Bars represent the level of each snRNA in ascitic fluid after therapy compared to samples before therapy ($n = 3$ biologically independent

samples). Data represent the mean values ± SD (standard deviation). **E** RT-qPCR analysis of spliceosomal snRNAs in pool of ascitic fluids from 3 patients after immunodepletion of snRNP complexes (using a mix of antibodies against U2AF1, SF3B1, and PRPF8) or extracellular vesicles (using an antibody against CD63) ($n = 3$ biologically independent samples). Data represent the mean values ± SD. **F** Nanoparticle tracking analysis of extracellular vesicles isolated from paired ovarian cancer ascitic fluids or secretomes (secretomes were collected as indicated in Fig. 3B) before (blue) and after (red) chemotherapy ($n = 14$ biologically independent samples). Data represent the mean values ± SD. The p-value was obtained by two-tailed unpaired Student's t test (**D**, **E**, **F**). Source data are provided as a Source Data file.

cells which correlates with less intensive protective effect of the secretomes from normal cells.

A total of 128 splicing-related proteins were identified in the therapy-induced secretome of SKOV3 cells (Supplementary Data 4, Sheet 5). These proteins included ubiquitously expressed core spliceosomal components as well as regulatory splicing factors (SR and hnRNP family members) that influence splice site selection in a cell-type-specific manner[37,38]. Further validation via RT-qPCR revealed elevated levels of all spliceosomal snRNAs in the therapy-induced secretomes from SKOV3 cells (Fig. 3I) with no increase in U35a, SNORD114, and HY1 snoRNAs, structurally similar to spliceosomal snRNAs but not involved in RNA splicing (Supplementary Fig. 4A).

We investigated whether spliceosomal proteins are encapsulated within extracellular vesicles in in vitro TIS, similar to what we observed in ascitic fluids. To this end, we conducted a proteinase K protection assay, demonstrating that proteinase K digested the analyzed spliceosomal proteins (DHX9, SRSF2, SRSF3, and SYNCRIP) only in the presence of detergent, suggesting their encapsulation in extracellular vesicles (Supplementary Fig. 4F).

Next, we analyzed the proteomic profiles of fibroblast secretomes, which did not confer a protective effect on recipient cells (Supplementary Fig. 3A, C, D). Notably, around 60% of proteins in the secretomes of untreated fibroblasts overlapped with those of untreated SKOV3 cells. However, secretomes obtained after cisplatin treatment exhibited pronounced differences between these two cell lines, with only 33% protein overlap (Supplementary Fig. 3C). Enrichment analysis highlighted an abundance of cell adhesion and extracellular matrix proteins in the secretomes of fibroblasts before treatment (Supplementary Fig. 4B), while lysosomal proteins and wound healing-related growth factors were upregulated after cisplatin treatment (Supplementary Fig. 4C). Notably, spliceosomal proteins were minimally present in fibroblast secretomes and exhibited little changes following chemotherapy (Fig. 3H). RT-qPCR confirmed the absence of differences in U snRNA levels in fibroblasts' secretomes after therapy (Supplementary Fig. 4D).

As expected, SKOV3 cells, but not fibroblasts, secreted various pro-oncogenic growth factors, including FAM3C protein, granulins (GRN), growth/differentiation factor 15 (GDF15), CXCL1, EFNA5, EFNB2, TGFB2, IGF2, vascular endothelial growth factors VEGFB and VEGFC, EFEMP1, LOXL2, BMP1, TGFB1, FSTL1, MYDGF, CYR61, and CSF1, involved in proliferation, migration, angiogenesis, and tissue remodeling[39] (Supplementary Fig. 4E). Surprisingly, the abundance of these proteins decreased sharply in the secretomes of cancer cells exposed to cisplatin. In contrast, in fibroblast secretomes, most of these proteins, including GDF15 and CTGF, were exclusively detected after chemotherapy (Supplementary Fig. 4E). Comparing these findings with the biological effects of secretomes suggests that despite a reduced content of soluble growth factors, tumor TIS have a more pronounced impact on recipient cells than secretomes obtained before therapy (Fig. 3B).

In summary, our findings indicate that, akin to ascitic fluids, in vitro cultures of ovarian cancer cells secrete a variety of

spliceosomal components encapsulated within extracellular vesicles following cisplatin treatment.

## Splicing factors from drug-stressed cells are transferred to recipient cancer cells

Spliceosomal proteins which predominantly located within the nucleus and therefore have to be exported to the cytoplasm for subsequent secretion from the cells. In this context, we investigated the changes in protein distribution between the cytoplasm and nucleus in SKOV3 cells following cisplatin treatment (Fig. 4A). To validate the quality of separation of the nuclear and cytoplasmic fractions, we evaluated the abundance of proteins that are frequently used as nuclear fraction markers (lamin-B1 and RPA194) (Supplementary Fig. 5A). Our analysis identified 4,085 proteins (Supplementary Data 5, Sheet 1). Proteins whose abundance differed two-fold or more were considered differentially present. Thus, 442 proteins were upregulated and 578 proteins were downregulated in the nuclear fraction after cisplatin treatment. Notably, the abundance of spliceosomal proteins decreased in the nucleus but increased in the cytoplasm in response to cisplatin treatment (Fig. 4B). Conversely, abundance of proteins related to the proteasome, endocytosis, amino acid biosynthesis, and pyrimidine metabolism increased in the nucleus but decreased in the cytoplasm after cisplatin treatment (Fig. 4B). We confirmed that spliceosomal proteins are exported from the nucleus during the therapy by co-transfecting SKOV3 cells with plasmids encoding spliceosomal proteins SRSF4 and TIA1 fused with RFP and EGFP, respectively (Fig. 4C). Additionally, using immunocytochemistry, we showed nuclear transport of spliceosomal protein SRSF1 while nuclear protein RPA2 retained in the nuclei after cisplatin treatment (Supplementary Fig. 5B). Our data demonstrated that spliceosomal proteins were transferred from the nucleus to the cytoplasm at early time points after cisplatin exposure, preceding nuclear fragmentation. This hypothesis aligns with prior research indicating the dissociation of certain spliceosomal proteins from chromatin during apoptosis[40,41], implying that nuclear export may be the initial step in the secretion of spliceosomal proteins from the cell.

To uncover the fate of splicing factors in the cytoplasm, we modified a method of stable isotope labeling with amino acids in cell culture (SILAC) (Fig. 4D). SKOV3 cells were cultured in a medium containing heavy isotope-labeled lysine and arginine. Once label incorporation exceeded 98%, cells were treated with cisplatin or left untreated (control). After 48 h, extracellular vesicles containing labeled proteins were isolated. These vesicles were added to unlabeled SKOV3 recipient cells, and after 10 or 48 h recipient cells were lysed and subjected to LC-MS/MS analysis to identify labeled proteins. A total of 189 heavy-labeled proteins were identified in recipient cells after 10 h of incubation with vesicles. Among them, 101 proteins were more abundant in cells incubated with vesicles from dying cancer cells, while only 29 proteins were more prevalent in cells exposed to control vesicles (Supplementary Data 5, Sheet 2). The functional annotation of heavy-labeled proteins whose abundance was elevated in the samples incubated with vesicles from dying cancer cells revealed the highest

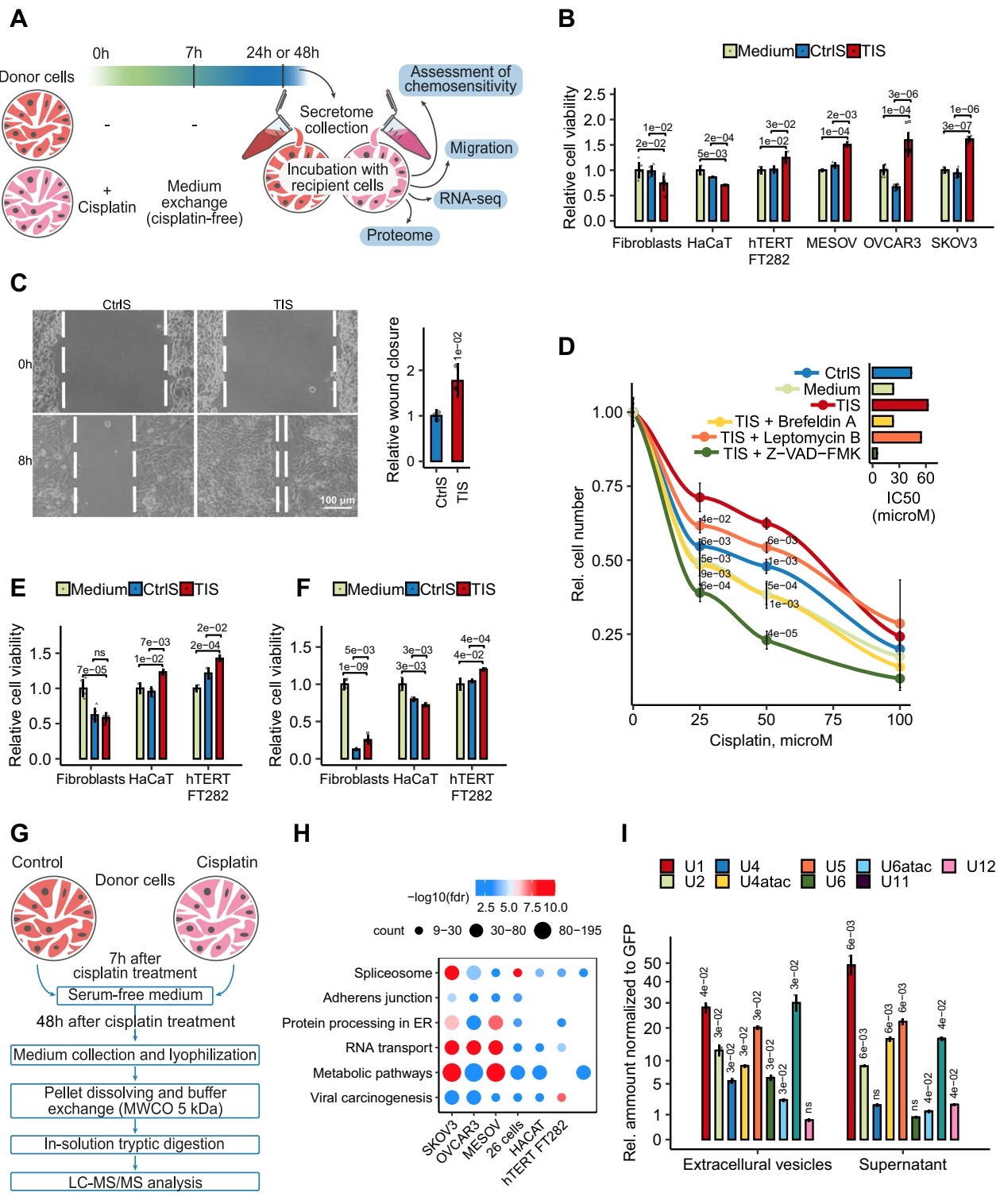

enrichment in the cluster of spliceosome-related proteins (Fig. 4E). Comparing data from cells incubated with vesicles for 10 and 48 h revealed that exogenous spliceosomal proteins exhibited greater stability in recipient cells compared to non-splicing proteins (Supplementary Fig. 5C). Next, we calculated the H/L ratio (heavy vs. light isotopes) for spliceosomal proteins to estimate how significantly vesicles can change the level of these proteins within the cell (Fig. 4F). Our results demonstrated that intracellular levels of several spliceosomal proteins (RBM15, SRSF4, DDX46, STRAP, and CDC40) was increased by more than two folds due to the incubation with the vesicles from apoptotic cells. We confirmed that various spliceosomal

proteins (TIA1, SFSF4, SYNCRIP, and SNU13) can be secreted from dying cancer cells and transferred into recipient cells using GFP- and RFP- fusion constructs and subsequent confocal microscopy and western blotting (Fig. 4G, H; Supplementary Fig. 5D). Furthermore, the presence of SRSF4-RFP, SYNCRIP-GFP, and SNU13-GFP in extracellular vesicles was confirmed through flow cytometry analysis of CD9 positive extracellular vesicles from dying cancer cells (Supplementary Fig. 5E).

In summary, our results indicate that in response to cispaltin treatment, spliceosomal proteins translocate from the nucleus to the cytoplasm and subsequently to extracellular vesicles. Uptake of these

**Fig. 3 | Therapy-induced secretomes of ovarian cancer cells promote cell chemoresistance in vitro. A** Experimental workflow used to assess the effect of therapy-induced secretomes on the behavior of ovarian cancer cells. **B** Secretomes were obtained from donor Fibroblasts, HaCaT, hTERT FT282, SKOV3, MESOV or OVCAR3 cells that were treated or untreated with cisplatin (40 μM for SKOV3 cells, 60 μM for MESOV cells, 25 μM for OVCAR3 and HaCaT cells, and 80 μM for Fibroblasts and hTERT FT282) for 7 h, then washed three times with PBS and cultured in serum-free media for 41 h. Recipient cells were incubated with corresponding therapy-induced (TIS; red bars) or control (CtrlS; blue bars) secretomes, or with fresh culture media (Medium; green bars) for 3 days, and then treated with cisplatin (10 μM for SKOV3, MESOV, hTERT FT282, and HaCaT cells, 7 μM for OVCAR3, and 40 μM for Fibroblasts). In vitro cell viability was assessed on day 2 after cisplatin adding. The data represent the mean values ± SD ($n = 3$ biologically independent experiments). **C** Wound healing assay of SKOV3 cells that were pre-incubated with TIS or CtrlS for 3 days (secretomes were collected as indicated in Fig. 3B). The width of the wound area was measured immediately after scratching (0 h) and the relative closure was measured after 8 h. The bar graph illustrates wound closure, expressed as the fold change, denoting the ratio of mean values of wound widths between two states: cells pre-treated with TIS relative to cells pre-treated with CtrlS. The data represent the mean values ± SD ($n = 3$ biologically independent experiments).
**D** Donor SKOV3 cells were exposed to 40 μM of cisplatin in the presence of Z-VAD-FMK (50 μM), Brefeldin A (6 μg/ml), or Leptomycin B (37 nM) for 7 h and then cells were washed three times with PBS and incubated in fresh serum-free media for 17 h. Recipient SKOV3 cells were incubated for 3 days with secretomes from treated (TIS) or untreated (CtrlS) donor cells, then recipient cells were treated with different concentrations of cisplatin for an additional 48 h. Dose-response curves and half-

maximal inhibitory concentration (IC50) values of cisplatin were determined using MTT assay. The data represent the mean values ± SD ($n = 3$ biologically independent experiments). **E** Recipient SKOV3 cells were incubated for 3 days with TIS or CtrlS from donor Fibroblasts, hTERT FT282, or HaCaT cells (secretomes were collected as indicated in Fig. 3B). After incubation, recipient SKOV3 cells were treated with cisplatin (10 μM). In vitro cell viability assay was performed on day 2 after cisplatin adding. The data represent the mean values ± SD ($n = 3$ biologically independent experiments). **F** Recipient Fibroblasts, hTERT FT282, or HaCaT cells were incubated for 3 days with TIS or CtrlS from donor SKOV3 cells (secretomes were collected as indicated in B). After incubation, recipient cells were treated with cisplatin (10 μM for HaCaT and hTERT FT282 cells and 40 μM for Fibroblasts). In vitro cell viability assay was performed on day 2 after cisplatin adding. The data represent the mean values ± SD ($n = 3$ biologically independent experiments).
**G** Experimental workflow used for proteomic analysis of cell secretomes. **H** Dot plot shows the KEGG enrichment analysis of proteins whose abundance increased at least 2 times TIS compared to control secretomes from different cell lines. It represents all common pathways upregulated in TIS of 4 ovarian cancer cell lines. The size of the dot is based on protein count enriched in the pathway, and the color of the dot shows the pathway enrichment significance. **I** RT-qPCR analysis of spliceosomal snRNAs in therapy-induced and control secretomes of SKOV3 cells (extracellular vesicles' fractions and supernatants were analyzed; secretomes were collected as indicated in Fig. 3B). Bars represent the level of each snRNA in therapy-induced secretomes compared to control secretomes. All data represent the mean values ± SD ($n = 3$ biologically independent experiments). The p-value was obtained by two-tailed unpaired Student's $t$ test (**B**–**F**, **I**). Source data are provided as a Source Data file.

---

vesicles can substantially increase levels of the spliceosomal proteins in recipient cells.

## Therapy-induced secretomes of ovarian cancer cells activate pathways important for cell response to DNA damage

Having established that spliceosomal components can be transferred from apoptotic cancer cells to surviving ones, we delved into the molecular mechanisms underlying the effects of TIS. Initially, we conducted RNAseq and LC-MS/MS analyses of recipient SKOV3 cells incubated with TIS or control secretomes for 3 days (Fig. 3A; Supplementary Data 5, Sheets 3–4). In alignment with the data obtained from ascitic fluids (Fig. 1F), we observed that TIS activated the expression of genes associated with DNA repair and cell cycle regulation in recipient cells (Fig. 5A). To investigate whether in vitro TIS could recapitulate the differences observed between platinum-sensitive and -resistant isogenic cell lines, we analyzed previously published RNAseq datasets encompassing several isogenic platinum-sensitive and -resistant ovarian cancer cell lines. Principal component analysis demonstrated that SKOV3 cells incubated with TIS clustered with platinum-resistant cancer cell lines. In contrast, cells exposed to control secretomes exhibited gene expression patterns akin to platinum-sensitive cell lines (Fig. 5B; Supplementary Data 5, Sheet 5). Subsequent enrichment analysis demonstrated that recipient cells exposed to TIS reproduced the same transcriptomic changes observed in platinum-resistant cell lines (Fig. 5C).

To further validate the RNAseq results, we conducted a proteomic analysis, revealing that TIS increased the abundance of proteins predominantly involved in DNA repair and cell cycle regulation in recipient cancer cells (Fig. 5A, Supplementary Data 5, Sheet 4). We confirmed these data using western blotting for several DNA repair proteins (Fig. 5D).

Recent research has shown that the knockout or knockdown of at least one splicing factor[42–45], including those identified in TIS, significantly impairs proper DNA repair and enhances the cytotoxic effect of various genotoxic drugs (Fig. 5E; Supplementary Fig. 6A). Based on these findings, we hypothesized that the increased abundance of spliceosomal proteins in cancer cells pre-incubated with TIS might facilitate a more effective response to DNA-damaging insults. To test this hypothesis, we treated SKOV3 cells pre-incubated with TIS or

control secretomes with cisplatin and evaluated various hallmarks of DNA damage response. We conducted a comet assay (analysis of DNA strand breaks), quantified cisplatin-DNA adduct accumulation, and monitored the phosphorylation levels of H2AX (an early DNA damage response marker[46]) and RPA2 (a replicative stress marker[47]). Remarkably, SKOV3 cells pre-incubated with TIS exhibited an increase in the number of γH2AX foci, accompanied by significantly fewer DNA double strand breaks, reduced platinum adducts, and lower RPA2 phosphorylation levels compared to cells incubated with control secretomes (Fig. 5F–I; Supplementary Fig. 6B). Additionally, cell cycle analysis indicated that cancer cells incubated with TIS remained longer in the S phase, which facilitates DNA repair[48–50] (Fig. 5J). Finally, we explored the impact of TIS on the sensitivity of cancer cells to both DNA-damaging (cisplatin, doxorubicin, and etoposide) and non-DNA damaging (taxane and staurosporine) anticancer drugs. The results underscored that TIS heightened the resistance of cancer cells exclusively to DNA-damaging drugs, with no discernible effect on other therapy types (Supplementary Fig. 6C).

In conclusion, these results suggest that TIS, containing various spliceosomal components, promote DNA repair in residual tumor cells, aiding their survival against subsequent therapeutic insults.

## Exogenous spliceosomal components promote malignancy of ovarian cancer cells in vitro and in vivo

To confirm that splicing factors in TIS indeed contribute to the acquisition of a more aggressive phenotype by cancer cells, we examined the effects of both protein and RNA components of the spliceosome on cancer cells in vitro and in vivo.

We selected SYNCRIP and SNU13 as representative exogenous spliceosomal proteins because i) they were detected in TIS secretomes from drug-stressed cancer cells (Supplementary Data 4, Sheets 2–3), ii) their levels increased in recipient cells after incubation with TIS (Fig. 4H), and iii) their depletion impaired DNA damage repair (Fig. 5E). Utilizing a lentiviral overexpression system, we demonstrated that elevated levels of SYNCRIP or SNU13 increased the resistance of ovarian cancer cells to cisplatin (Fig. 6A; Supplementary Fig. 7A, B). Conversely, knockdown of SYNCRIP or SNU13 significantly sensitized cancer cells to cisplatin (Supplementary Fig. 7C, D). RNAseq analysis of SKOV3 cells overexpressing SYNCRIP,

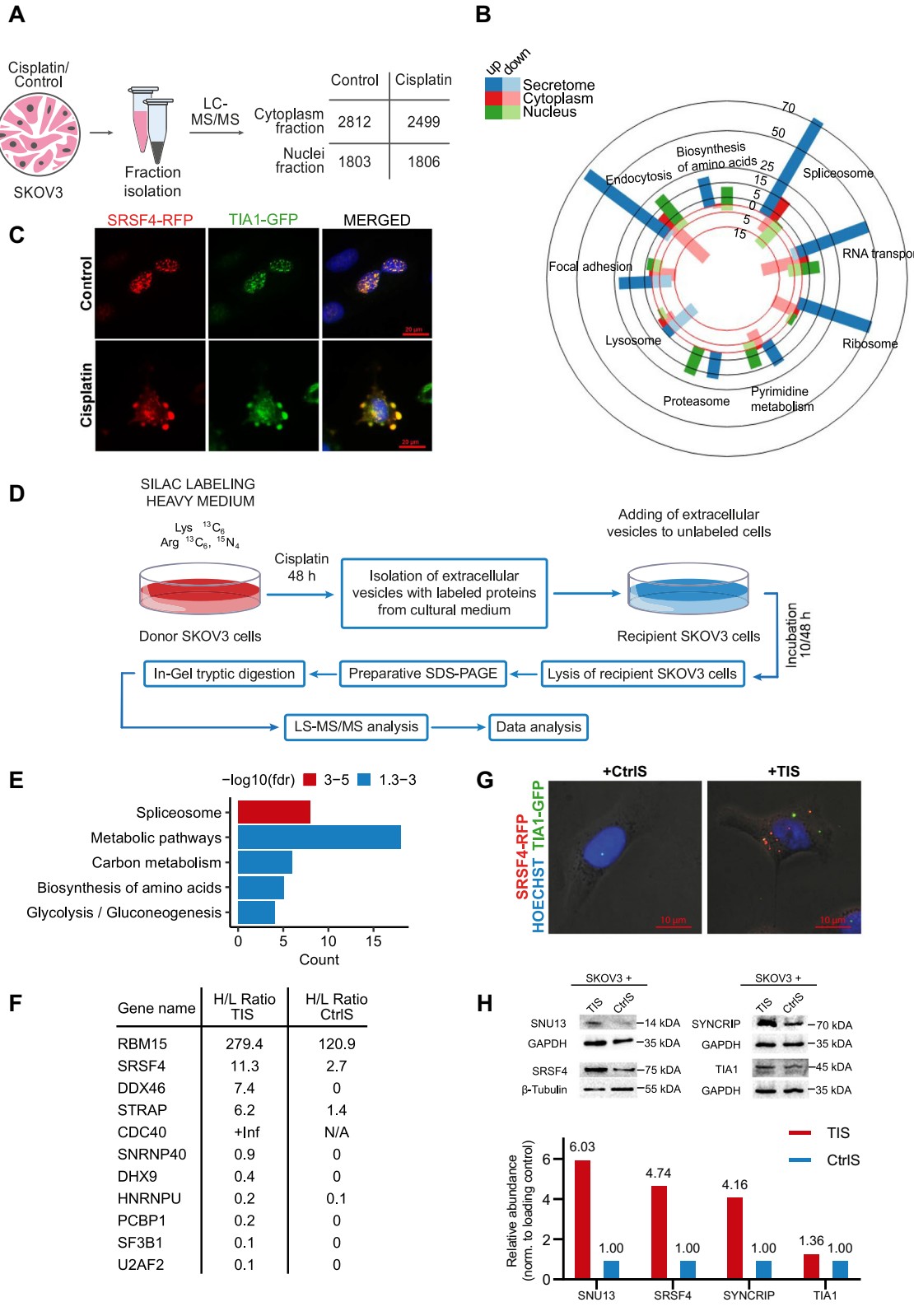

**Nature Communications** | (2024)15:5237

SNU13, or an empty vector before and after cisplatin treatment revealed distinct responses of control and SYNCRIP or SNU13 over-expressing cells to cisplatin (Supplementary Fig. 7E; Supplementrary Data 6). Enrichment analysis demonstrated that after treatment, both SYNCRIP and SNU13 overexpressing cells had increased expression of genes involved in DNA repair compared to control cells

(Fig. 6B, C). These findings suggest that exogenous splicing factors may promote DNA damage response, at least partially, by modulating the expression of DNA repair genes.

Next, we explored the effects of spliceosomal non-coding snRNAs. To select candidates for further investigation, we determined which snRNAs were most actively secreted by cancer cells.

**Fig. 4 | Splicing factors from drug-stressed cells are transferred to recipient cancer cells. A** Experimental workflow used for cell fractionation with subsequent proteomic analysis. **B** Bar diagram showing functional annotation (KEGG) of proteins that changed in abundance at least 2 times upon cisplatin treatment. The bar color indicates cell fraction: the blue represents a secretome fraction; the red represents a cytoplasmic fraction; the green represents a nuclear fraction. The bright colors indicate numbers of upregulated proteins for specific terms; the pale colors represent numbers of down-regulated proteins for corresponding terms. **C** Fluorescence images of SKOV3 cells co-expressing RFP-SRSF4 (red) and GFP-TIA1 (green) before and after treatment with 40 μM cisplatin for 7 h. Nuclei are stained by DAPI (blue). It was repeated with similar results in two independent experiments. **D** Experimental design used for SILAC experiment. **E** Results of the KEGG enrichment analysis of heavy-labeled proteins that were upregulated (at least 2 times) in recipient SKOV3 cells incubated with vesicles from dying cancer cells. STRING was

used for functional enrichment analysis with all genes as background. A hypergeometric test was carried out and all significant categories (false discovery rate < 0.05, after correction for multiple testing using the Benjamini−Hochberg procedure) are displayed. **F** H/L ratio (heavy vs. light isotopes) for spliceosomal proteins 10 h after incubation with extracellular vesicles from secretomes of cisplatin-treated (TIS) or untreated (CtrlS) donor cells. **G** Immunofluorescence images of recipient SKOV3 cells incubated for 3 days with therapy-induced (TIS) or control secretomes (CtrlS) from donor SKOV3 overexpressing SRSF4-RFP and TIA1-GFP proteins (secretomes were collected as indicated in Fig. 3B). It was repeated with similar results in two independent experiments. **H** Western blotting analysis of SKOV3 cells that were incubated for 3 days with therapy-induced (TIS) or control secretomes (CtrlS) from donor SKOV3 (secretomes were collected as indicated in Fig. 3B). It was repeated with similar results in two independent experiments. Source data are provided as a Source Data file.

---

Surprisingly, among all spliceosomal snRNAs, U12 snRNA exhibited the highest relative increase in ascitic fluids after therapy (Fig. 6D). Importantly, U12 snRNA levels increased in samples from most patients (7 out of 11) after chemotherapy, whereas no such trend was observed for 18S rRNA (increased in 4 out of 11 patients) (Fig. 6E). Based on these data, we chose U12 snRNA for further analysis, along with U6atac snRNA which functions in a complex with U12 snRNA (Supplementary Fig. 8A).

To ensure the reliability and reproducibility of our results, we used two complementary approaches to study the effects of exogenous snRNA: i) transfection of cells with artificially synthesized snRNAs; and ii) a lentiviral overexpression system to directly increase intracellular levels of the corresponding U snRNAs.

To prevent a non-specific immune response to the exogenous snRNAs, we incorporated modified nucleotides into the RNA structure[51,52] and confirmed the absence of cytotoxicity of the synthetic RNAs (Supplementary Fig. 8B–D). We used radioactive labels to demonstrate the efficiency of delivery and stability of the synthetic RNA constructs in the cells (Supplementary Fig. 8E).

Next, we investigated the effects of the selected snRNAs by transfecting SKOV3 cells with U12 snRNA, U6atac snRNA, or a GFP mRNA fragment (89 nucleotides) and performed RNAseq and proteomic profiling 48 h after the transfection. Additionally, all samples were assessed for transfection efficiency and absence of interferon response activation (Supplementary Fig. 8F, G). RNAseq analysis revealed that spliceosomal U12 and U6atac snRNAs induced similar changes in gene expression in the recipient cells, while cells transfected with the GFP mRNA fragment remained similar to the non-transfected control (Fig. 6F; Supplementary Data 7, Sheet 1). Functional annotation of upregulated genes after transfection with U12 or U6atac snRNAs unveiled the activation of genes involved in cell cycle regulation as well as for DNA damage checkpoint signaling such as *MSH2, RAD50, DNA2, SMC3, RBBP8* which also facilitates G1/S transition (Fig. 6G, Supplementary Data 7, Sheet 1).

We conducted proteomic profiling of the corresponding samples and identified 3,946 proteins (Supplementary Fig. 8H; Supplementary Data 7, Sheet 2). Consistently with our RNAseq data, among the differentially enriched proteins in cells transfected with both U12 and U6atac snRNAs, significant differences were observed in the cell cycle regulation clusters, including proteins involved in the progression through the G1 phase and DNA repair such as CCNB1, AURKB, CDK4, ANAPC4/5, CHEK1, XPC. Based on these data, we next explored whether increased levels of snRNAs would indeed affect cell cycle progression. FACS analysis demonstrated that cancer cells transfected with U12 and U6atac snRNAs spent less time in G1 phase, which is often accompanied by a high proliferation rate[53] (Supplementary Fig. 8J). To validate these results, we overexpressed U12 and U6atac snRNAs in SKOV3 cells using lentiviral constructs and showed that both U snRNAs substantially increased the proliferation of SKOV3 cells (Fig. 6H, Supplementary Fig. 8I).

Finally, we investigated the effect of exogenous spliceosomal snRNAs on tumor progression in vivo. In this experiment, we first transfected SKOV3 cells with synthetic U12, U6atac snRNA analogs, or a GFP mRNA fragment (control) and subsequently subcutaneously injected the cells into immunodeficient SCID mice. Tumor size was evaluated after 70 days. Our results showed that both U6atac and U12 snRNAs significantly accelerated tumor growth in vivo (Fig. 6I).

In summary, our findings reveal that exogenous spliceosomal proteins and snRNAs, by influencing different pathways such as DNA repair and cell cycle, promote the malignancy of ovarian cancer cells and partially recapitulate the effects of drug-stressed secretomes.

## Discussion

The acquisition of drug resistance in cancer cannot be solely attributed to de novo gene mutations[54]. Extracellular signals play a significant role in the acquisition of therapy resistance by modulating gene expression at the transcriptome level. Previous studies have demonstrated that secretomes from apoptotic tumor cells enhance metabolic activity, cell migration, and therapy resistance of cancer cells[5,7,17,55]. However, current data on this matter remain incomplete and controversial, as some studies indicate that therapy-induced cancer secretomes enhance the cytotoxic activity of natural killer cells[56] and that lethally irradiated cells may transfer death signals to neighboring tumor cells[57]. In the context of ex vivo secretomes, primary tumor ascitic fluids were found to promote cancer progression by enhancing proliferation[58,59], migration[60], invasion[61,62], resistance to apoptosis[63,64] and epithelial-mesenchymal transition[65]. Conversely, some researchers observed that tumor ascites significantly enhance ovarian cancer cell apoptosis[59,66,67]. Additionally, cell-free and concentrated ascites reinfusion therapy was shown to be effective in ovarian cancer patients[68,69]. Thus, malignant ascitic fluids can significantly modulate the behavior of tumor cells, but the role that they play during chemotherapy is unclear. Our study contributes to this understanding by revealing that ovarian cancer ascitic fluids after chemotherapy contain spliceosomal components that promote the acquisition of a more therapy-resistant phenotype in recipient cancer cells.

Despite the high biological importance of therapy-induced tumor secretomes, their composition has not yet undergone systematic assessment. While numerous studies reveal that various stress conditions such as hypoxia, irradiation, and chemotherapy, significantly increase the secretion of extracellular vesicles by dying cancer cells compared to untreated ones[57], proteomic profiling of these secretomes and corresponding extracellular vesicles is often lacking. Existing proteomic data are predominantly confined to descriptions of in vitro secretomes from sensitive and resistant cancer cells[70,71], mesenchymal and epithelial cancer cells[72], and cancer cells with a different metastasis potential[73]. Notably, chemotherapy-induced alterations in cancer cell secretomes ex vivo remain unexplored. In this study, we conducted a proteomic analysis of paired ascitic fluids from ovarian cancer patients as well as in vitro secretomes from several

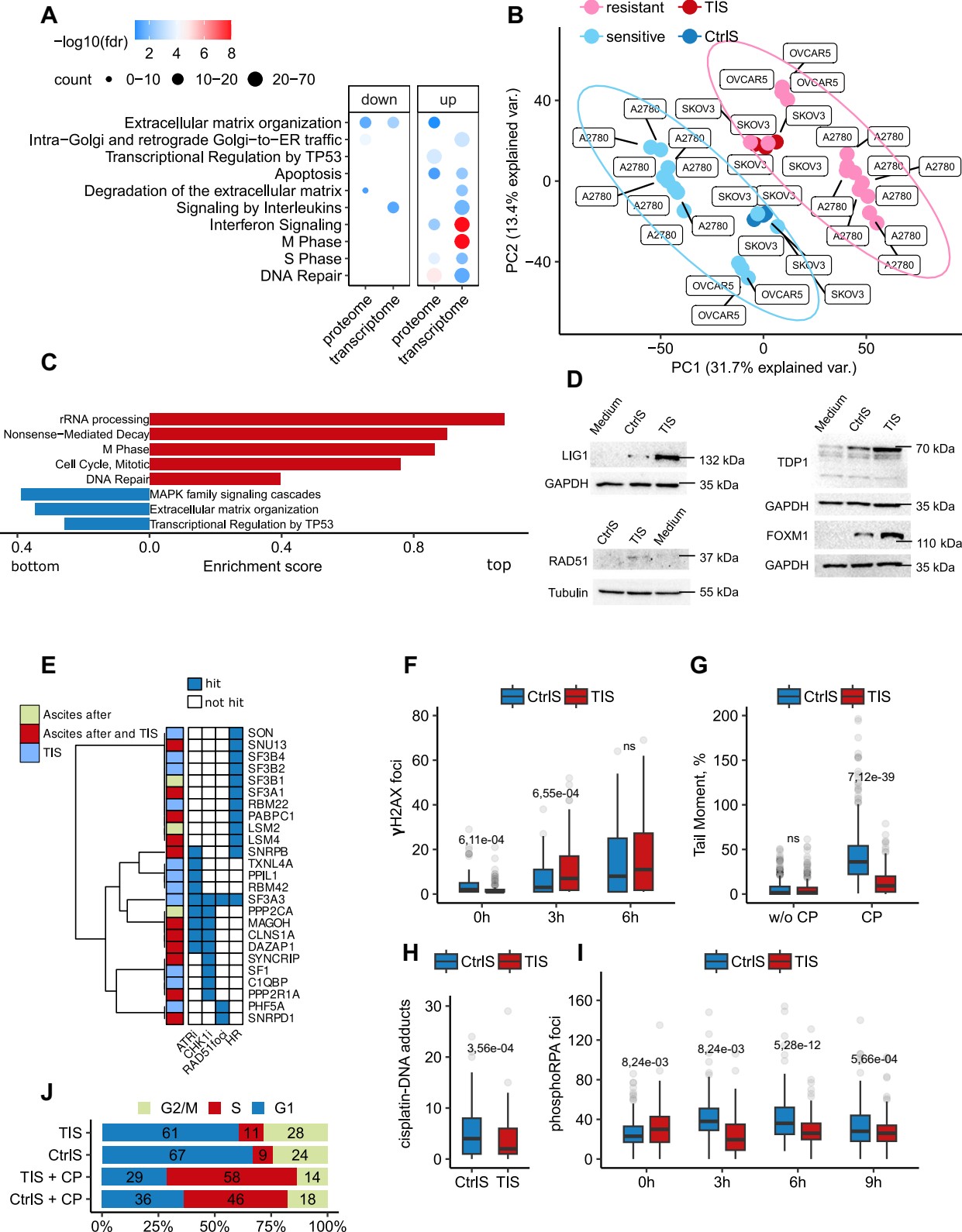

ovarian cancer cell lines before and after chemotherapy. Our findings revealed that both in vitro and ex vivo secretomes after chemotherapy exhibit enrichment with spliceosomal components. This observation was corroborated by analyzing six previously published LC-MS/MS datasets. Consistent with our data, secretomes from dying lymphoma[19], melanoma[20], and glioblastoma[7] cells were enriched with spliceosomal proteins compared to the secreomes from

corresponding untreated cells (Supplementary Data 4, Sheet 4). Interestingly, secretomes from dying normal cells (pigment epithelial cells (RPE)[74], endothelial[75] cells or fibroblasts[76]) did not demonstrate upregulated secretion of spliceosomal proteins. Furthermore, it is noteworthy that senescence-associated secretory phenotype (SASP) inducers did not stimulate the secretion of spliceosomal proteins[77]. Despite identifying multiple SASP-related proteins in all our

**Fig. 5 | Therapy-induced secretomes of ovarian cancer cells activate pathways important for cell response to DNA damage. A** Gene Ontology enrichment analyses of upregulated genes and proteins in SKOV3 cells incubated for 3 days with therapy-induced secretomes (TIS) compared to control secretomes (CtrlS). The color scale refers to −log10 (FDR) values; the number of proteins/genes are represented by the diameter of the circles. **B** The principal component analysis of RNAseq data obtained from platinum-sensitive and -resistant isogenic ovarian cancer cell lines and recipient SKOV3 cells incubated for 3 days with TIS or CtrlS. Pink−platinum-resistant ovarian cancer cell lines, light blue−platinum-sensitive ovarian cancer cell lines, red−recipient SKOV3 cells incubated with TIS, dark blue− recipient SKOV3 cells incubated with CtrlS. **C** GSEA analysis of gene expression in platinum-resistant ovarian cancer cell lines versus platinum-sensitive ovarian cancer cell lines. The *X*-axis represents GSEA enrichment score (*p*-values are indicated by colors). **D** Western blotting analysis of SKOV3 cells that were incubated for 3 days with TIS or CtrlS from donor SKOV3. **E** Results of the intersection between spliceosomal proteins identified in TIS from SKOV3 cells and/or in ovarian cancer ascites after therapy (our data) and the hits from siRNA and CRISPR screenings (derived from data reported in refs. 42–44). ATRi and CHK1i−CRISPR screens with inhibitors targeting ATR and CHK1, respectively. Loss of spliceosomal proteins indicated as "hit" increased the sensitivity of cancer cells to ATR or CHK1 inhibition [42]. RAD51 foci and HR−siRNA screenings indicating that knockdown of spliceosomal protein impair the formation of IR-induced RAD51 foci or decreased homologous recombination (HR) potential in the DR-GFP assay in cancer cells, respectively, [43,44]. **F** Box plots show the number of γH2AX foci per nucleus in SKOV3 cells pre-incubated with TIS or CrlS for 3 days and then treated with cisplatin (25 μM) at different time points (TIS: 0 h *n* = 134 cells, 3 h *n* = 136 cells, 6 h *n* = 129 cells; CtrlS: 0 h *n* = 109 cells; 3 h *n* = 130 cells; 6 h *n* = 144). The number of γH2AX foci

was calculated using ImageJ software with FindFoci plugins. **G** Box plots of tail moments from neutral comet assays of SKOV3 cells pre-incubated with TIS or CrlS for 3 days and then treated with cisplatin (10 μM) for 48 h (TIS: CP *n* = 381 cell, w/o CP *n* = 368 cells; CtrlS: CP *n* = 469 cells, w/o CP *n* = 296). Experiments were performed in triplicate. **H** Box plots show the number of cisplatin-DNA adducts' foci per nucleus of SKOV3 cells pre-incubated with TIS or CrlS for 3 days and then treated with cisplatin (25 μM) for 48 h (TIS *n* = 243 cells; CtrlS *n* = 146 cells). The number of cisplatin-DNA adducts' foci was calculated using ImageJ software. **I** Box plots show the number of phosphorylated RPA2 (phospho S33) foci per nucleus in SKOV3 cells pre-incubated with TIS or CrlS for 3 days and then treated with cisplatin (25 μM) at different time points (TIS: 0 h *n* = 194 cells, 3 h *n* = 216 cells, 6 h *n* = 264 cells, 9 h *n* = 241; CtrlS: 0 h *n* = 228 cells; 3 h *n* = 195 cells, 6 h *n* = 174, 9 h *n* = 193). The number of phosphorylated RPA2 foci was calculated using ImageJ software. **J** Cell cycle analysis with flow cytometry of SKOV3 cells pre-incubated with TIS or CrlS for 3 days and then treated with cisplatin (10 μM) for 24 h. Stacked bar graphs show the percentage of cells in different phases of the cell cycle. Percentage of cells in G1, S, and G2 phases was calculated with NovoExpress software. Secretomes were collected as indicated in Fig. 3B. The line in each box is the median, the up and low of each box are the first and third quartiles. The upper whisker extends from the up of the box to the largest value no further than 1.5*IQR (where IQR is the inter-quartile range). The lower whisker extends from the low of the box to the smallest value at most 1.5*IQR. Data beyond the end of the whiskers are called "outlying" points and are plotted individually. The *p*-value was obtained by two-tailed unpaired Student's *t* test (**F**–**I**). Gene expression signature analysis was performed using the "signatureSearch" packages in "R" against the Reactome database (**C**). Source data are provided as a Source Data file.

---

secretomes (Supplementary Data 4, Sheet 6)[21,76,78], their accumulation did not prominently increase after therapy. Thus, we posit that the biological effect of TIS is unlikely to be associated with the SASP phenotype. Instead, our data underscore the potential significance of splicing regulators as mediators of intercellular communication during therapy.

Under normal conditions, spliceosomal components are mainly localized in the nucleus. Given the results of our subcellular proteomic analysis, we suggest that chemotherapeutic drug-induced stress provokes re-localization of spliceosomal proteins. Previous studies have indicated that splicing factors could be transported to cytoplasmic stress granules upon stress such as hypoxia, heat shock, and chemotherapy[41,79–82]. Stress granules are non-membranous structures with high concentrations of proteins and mRNAs mainly involved in cell-cycle regulation, chromosome organization, and RNA metabolism[41,80–84]. The human stress granules' proteome encompasses around 600 proteins, with approximately half being RNA-binding proteins[85–87]. We speculate that under the stress induced by chemotherapeutic drugs, spliceosomal proteins may re-localize to cytoplasmic stress granules, which potentially can be exported within extracellular vesicles. Intriguingly, our analysis identified proteins known as main stress granule markers (G3BP1, PABPC1, and TIA1) in ascitic fluids and secretomes of cancer cells after chemotherapy. Comparing the proteomic profiles of ascitic fluids/secretomes with the stress granule proteome revealed approximately 300 overlapping proteins, including spliceosomal proteins and translation initiation factors (Supplementary Data 4, Sheet 6). Thus, we propose the intriguing possibility that stress granules and stress granule-associated mRNAs might be exported and subsequently delivered to recipient cells during therapy. Importantly, the potential for intercellular transport of large structures has been previously demonstrated for mitochondria and ribosomes[88–91].

Transcriptomic and proteomic analyses of recipient cancer cells exposed to TIS revealed that TIS, both in vitro and in vivo, upregulated genes and proteins involved in DNA damage response and increased resistance of cancer cells to DNA-damaging drugs. Our modified SILAC-MS approach highlighted spliceosomal proteins as among the most abundant proteins delivered into recipient cells by TIS.

Numerous studies have demonstrated that elevated levels of splicing factors in cancer cells lead to the upregulation of DNA repair proteins[92–95]. Genome-wide siRNA and CRISPR screenings in cancer cells further validated that knockdown/knockout of spliceosomal genes disrupts DNA repair processes[42–45], resulting in increased levels of H2AX phosphorylation[96], and synthetic lethality for cancer cells in combination with DNA-damaging agents[95]. Our data specifically highlight the significant contribution of splicing factors SYNCRIP and SNU13 to TIS-induced enhanced DNA repair. These findings align with a growing body of evidence demonstrating that various spliceosomal proteins regulate the expression of DNA repair genes, and are involved in the R-loop resolving, signaling of DNA breaks, establishment of liquid-liquid phase separation, or remodeling of chromatin at DNA damage sites[93]. However, the precise molecular mechanisms by which spliceosomal proteins participate in DNA repair warrant further investigation.

In addition to impacting DNA repair pathways, TIS also affected cell cycle regulation in recipient cancer cells. Similarly, the overexpression of spliceosomal proteins SYNCRIP and SNU13, as well as spliceosomal snRNAs U12 and U6atac, led to the upregulation of genes associated with cell cycle progression. It is noteworthy that multiple splicing factors were shown to participate in cell cycle regulation[97,98]. They can regulate cell division through the alternative splicing of mitotic-related pre-mRNAs[99,100] or play a direct role in mitosis progression[101–103]. For example, several spliceosomal proteins were shown to be associated with the cohesin complex which regulates the segregation of sister chromatids. Depletion of spliceosomal proteins disrupts this process, leading to aberrant mitosis[97,103–109]. It is reasonable to speculate that exogenous spliceosomal components can influence cell cycle regulation in recipient cells. Importantly, we demonstrated that not only spliceosomal proteins but also spliceosomal snRNAs (U12 and U6atac) can be involved in this process. Cells with higher levels of these snRNAs had higher proliferation rate and spent a shorter time in the G1 phase. This observation aligns with data showing that the knockdown of U6atac snRNA, as well as several other components of the minor spliceosome, resulted in the downregulation of genes related to the cell cycle and DNA repair and blocked the proliferation of cancer cells but not non-cancer cell lines[110].

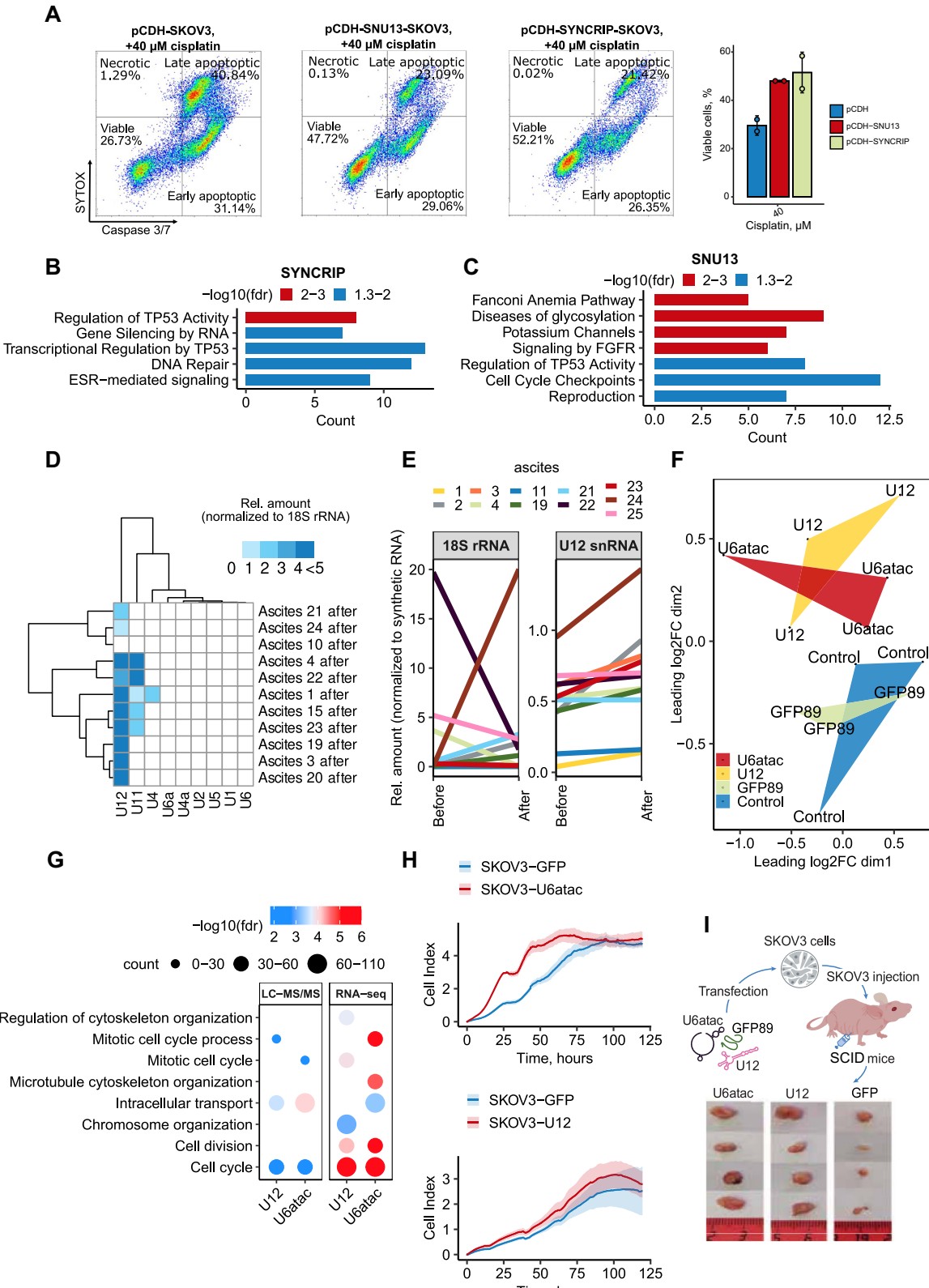

In this study, our primary focus was on exploring the communication between cancer cells. However, it is essential to acknowledge the potential significance of interactions between the tumor and its microenvironment. Our findings indicate a substantial negative impact of tumor cell secretomes on fibroblast proliferation. While this observation could be attributed to in vitro experimental conditions, it also raises the possibility of an important biological mechanism. The

inhibition of fibroblast proliferation by cancer cells might signify a strategic maneuver. Cancer cells could potentially impede the formation of a fibrous capsule around the tumor, thereby facilitating metastasis and access of the blood vessels to the tumor[111]. On the other hand, the tightly regulated and stable phenotype of normal cells, compared to cancer cells, may render them less responsive to signals from TIS, possibly resulting in a limited pro-proliferative effect on non-

**Fig. 6 | Extracellular spliceosomal components contribute to the aggressiveness of ovarian cancer. A** Flow cytometry analysis of caspase 3/7 activity and SYTOX staining of SKOV3 cells stably expressing SYNCRIP, SNU13 or an empty vector (pCDH), treated with 40 µM of cisplatin for 24 h. The bar graph (on the right) shows the percentage of viable cells for each cell line 24 h after treatment with 40 µM of cisplatin (n = 2 biologically independent experiments). **B** Reactome enrichment analysis of upregulated genes in SKOV3 cells overexpressing SYNCRIP versus control pCDH cells 24 h after cisplatin (40 µM) addition (p-values are indicated by colors). **C** Reactome enrichment analysis of upregulated genes in SKOV3 cells overexpressing SNU13 versus control pCDH cells 24 h after cisplatin (40 µM) addition (p-values are indicated by colors). **D** Heat map displaying the relative amount of 9 snRNAs in ovarian cancer cells and malignant ascitic fluids after chemotherapy from 11 patients. The data represent the mean values ± SD (n = 3 biologically independent samples). **E** Spaghetti plots of U12 snRNA and 18S rRNA levels in paired ascitic fluids from 11 patients before and after chemotherapy. The data represent the mean values ± SD (n = 3 biologically independent samples). **F** Multidimensional scaling (MDS) plot of all expressed genes reflects changes between control cells transfected with empty lipofectamine (blue) or GFP mRNA fragment (green) and cells transfected with U12 snRNA (yellow) or U6atac snRNA (red). Each point represents 1 sample, and the distance between 2 points reflects the logFC of the corresponding RNA-seq samples. **G** Gene Ontology enrichment analysis of upregulated proteins (left panel) or genes (right panel) in SKOV3 cells transfected with synthetic U12 snRNA or U6atac snRNA compared to control cells transfected with empty lipofectamine. The color scale refers to −log10 (FDR) values; the number of proteins/genes are represented by the diameter of the circles. **H** xCELLigence proliferation assay of SKOV3 cells overexpressing U12 snRNA (top panel), U6atac snRNA (bottom panel) or control GFP mRNA fragments (with the corresponding promoters: U2 or U6, respectively). SKOV3 cells were seeded in 96-well E-plates for xCELLigence assay monitoring impedance (cell index, CI) (n = 3 biologically independent samples). Mean values of the CI were plotted ± standard deviation. **I** Scheme of the experiment (top panel) and the representative tumor images (bottom panel) obtained from SCID mice injected with 4 × 10⁶ SKOV3 cells which were transfected with synthetic U12 snRNA, U6atac snRNA, or a GFP mRNA fragment (control). ClusterProfiler was used for functional enrichment analysis with all genes as background (**B**, **C**). A hypergeometric test was carried out, and all significant categories (false discovery rate < 0.05, after correction for multiple testing using the Benjamini−Hochberg procedure) are displayed. Source data are provided as a Source Data file.

cancerous cell types. A compelling avenue for further investigation would be to explore whether secretomes after therapy also have the capacity to influence the proliferation of immune cells. This exploration could uncover yet another mechanism through which TIS might exert pro-oncogenic effects.

In conclusion, our study underscores the significance of extracellular signals in acquisition of therapy resistance in ovarian cancer. It highlights the role of spliceosomal components in intercellular communication during therapy, elucidating their contributions into the DNA damage response and cell cycle regulation in recipient cells. This sheds light on a mechanism underlying chemotherapy resistance. These findings pave the way for potential therapeutic strategies. Targeting spliceosomal components could enhance the efficacy of DNA-damaging drugs, curtail the acquisition of therapy resistance, and ultimately enhance the outcomes for patients with ovarian cancer.

## Methods
This research complies all relevant ethical regulations. All patients provided written informed consent for participation. Ethical approval was obtained from the Ethics Committees of the Russian Research Center of Roentgenology and Radiology (agreement and protocol no. 30-2018/E from 13 November 2018) and National Medical Research Center for Obstetrics, Gynecology, and Perinatology named after Academician V.I. Kulakov of the Ministry of Healthcare of the Russian Federation (protocol no. 10 of 5 December 2019). All animal experiments were carried out in compliance with the protocols and recommendations for the proper use and care of laboratory animals (EEC Directive 86/609/EEC). The study protocol was approved by the Committee on the Ethics of Animal Experiments of the Administration of SB RAS (Permit #40 from April 4, 2018).

### Patients and specimens
Paired ascitic samples were collected from 25 ovarian cancer patients before and after neoadjuvant chemotherapy. The samples were obtained from the National Medical Research Center for Obstetrics, Gynecology, and Perinatology named after Academician V.I. Kulakov of the Ministry of Healthcare of the Russian Federation (Moscow, Russia) and from the Russian Research Center of Roentgenology and Radiology of the Ministry of Healthcare of the Russian Federation (Moscow, Russia). Ascitic fluids obtained before therapy were collected at the time of diagnosis through laparocentesis, aspiration during diagnostic laparoscopy, or puncture through the posterior vaginal fornix. Post-chemotherapy ascitic fluids were collected intraoperatively after several courses of neoadjuvant chemotherapy. The clinical samples were carefully characterized, including histological and cytological findings, information on the number of

neoadjuvant chemotherapy courses, and mutations in the BRCA1/2 genes. RECIST criteria were assessed for each patient. Patients, from whom we got paired ascitic fluids before and after therapy, had Partial Response, Stable Disease, or Progressive Disease. Information on the current state of the patients and recurrences was also collected (Supplementary Data 1, Sheet 1).

The ascitic fluids were used for the following experiments: establishment of primary cell cultures; RNAseq analysis; MTT, and wound healing assaysis to study the effect of ascitic fluids on cancer cells; proteomic analysis; exosome counting using nanoparticle tracking analysis (NTA); immunoblotting; RNA isolation followed by RT-qPCR of selected molecules.

### Processing of patient samples
The ascitic fluids from ovarian cancer patients before and after chemotherapy were collected in sterile 50-ml tubes (Eppendorf). Within 1 h of collection, fresh ascitic fluids were centrifuged at 200 g for 15 min at 4 °C to precipitate the cells. The supernatant fraction of the ascitic fluids was aliquoted and stored at −70 °C.

Contaminating red blood cells in the cell pellets of ascites were removed by hypotonic lysis using sterile MilliQ H2O. One part of the ascites cells was cryopreserved in 90% fetal bovine serum (FBS; Gibco, #A3382001) and 10% dimethyl sulfoxide (DMSO; Paneco, Φ135), while the other part was seeded on plastic plates (Corning) and cultured for no longer than 8 passages in RPMI growth medium (Paneco, C330п) supplemented with 10% FBS, 2 mM glutamine (Gibco, #35050061), 1% penicillin/streptomycin (Paneco, A065), and 1% non-essential amino acids (Paneco, Φ115/100п). The cells were maintained at 37 °C in a 5% $CO_2$ atmosphere.

### Cell cultures
Human ovarian cancer cell lines SKOV3 (ATCC, HTB-77), MESOV (ATCC, CLR-3272), and human keratinocyte cells HaCaT (CLS, 300493) were grown as adherent monolayers in DMEM medium (Paneco, C420п) supplemented with 10% FBS, 2 mM L-glutamine, and 1% penicillin/streptomycin. Human ovarian cancer cell line OVCAR3 (ATCC, HTB-161) was grown in RPMI medium supplemented with 10% FBS, 2 mM L-glutamine, and 1% penicillin/streptomycin. hTERT FT282 cells (ATCC, CRL-3449) were grown in DMEM/F12 medium (Paneco, C470п) supplemented with 10% FBS, 2 mM L-glutamine, and 1% penicillin/streptomycin. A primary culture of human dermal fibroblasts was grown in DMEM medium supplemented with 10% FBS, 1% non-essential amino acids, 2 mM L-glutamine, and 1% penicillin/streptomycin. Ethical approval was obtained for the fibroblast cell line from the Research and Clinical Center of Physical-Chemical Medicine. The Phoenix-GP packaging cell line (ATCC, CRL-3215) was cultured in DMEM/F12

medium (Paneco, C470п) containing 10% FBS, 2 mM L-glutamine, 1 mM sodium pyruvate (Paneco, Φ023), and 1% penicillin/streptomycin. All cell lines were incubated at 37 °C in a humidified atmosphere containing 5% CO2.

## Secretome generation

To generate conditioned media (CM), cells (at approximately 80% confluence) were incubated with FBS-free standard medium with or without cisplatin (Teva Pharmaceutical Industries Ltd, N011590/02): 40 μM for SKOV3 cells, 60 μM for MESOV cells, 25 μM for OVCAR3, HaCaT cells, and primary culture of 26 cells, and 80 μM for Fibroblasts and hTERT FT282 cells. After 7 h, cisplatin-containing and control media were removed, and the adherent cells were washed thoroughly four times with phosphate-buffered saline (PBS). Then, cells were incubated for 17 (for experiment with inhibitors) or 41 h in a fresh FBS-free medium (Fig. 3A). The cisplatin treatment regimen was chosen to ensure that approximately 50% of the donor cells were dead at the time of CM collection, allowing for a high level of apoptosis in tumor cells and simultaneous collection of drug-free medium.

CMs were collected 24 or 48 h after cisplatin adding, followed by centrifugation for 10 min at $500\,g$ at 4 °C in the F-45-24-11 rotor (Eppendorf, Germany) to remove cells. The supernatants were concentrated using a 100-kDa molecular weight cut-off spin concentrators (Amicon, Merck, UFC910024) to enrich with extracellular vesicles. Finally, a fraction of extracellular vesicles (>100 kDa fraction) was diluted in fresh media containing 2% FBS.

For proteomic analysis of secretomes, cells were treated or untreated with cisplatin as indicated above. After 7 h, cisplatin-containing or control media were removed, and the adherent cells were washed thoroughly four times with PBS. Then, cells were incubated for 41 h in an FBS-free and phenol-red-free medium (Paneco, C420п-1).

For experiments with low molecule inhibitors, effective concentrations of Brefeldin A (6 μg/ml; Sigma-Aldrich, B6542) and Leptomycin B (37 nM; Cayman Chemical, #10004976) were selected using western blot analyses (Supplementary Fig. 2E, F), an effective concentration of Z-VAD-FMK (50 μM; Sigma-Aldrich, V-116) was selected using CellEvent Caspase-3/7 Green Detection Reagent (Invitrogen, #C10427) followed by flow cytometry (Supplementary Fig. 2G).

## Animal studies

All animal experiments were carried out in compliance with the protocols and recommendations for the proper use and care of laboratory animals (EEC Directive 86/609/EEC). The study protocol was approved by the Committee on the Ethics of Animal Experiments of the Administration of the Siberian Branch of the Russian Academy of Science (Permit #40 from April, 4 2018). The mice were housed in groups of four animals per cage and had access to autoclaved water and pelleted feed. The cage environment was enriched with a mouse house. The mice were kept at a standard temperature of 22 C ± 2 °C and relative humidity of 55% (45–70%) in a 12:12-h light:dark cycle (lights on, 6 am to 6 pm). SKOV3 cells were transfected with 10 μM of synthetic analogs U6atac snRNA, U12 snRNA, or a synthetic mRNA fragment of the GFP in complex with a lipid transfection agent RNAiMAX (Invitrogen, #13778150). Control cells were incubated in a medium with an empty transfection agent. 48 h after transfection, cell suspensions ($4 \times 10^6$ cells in 500 μl of PBS) were injected subcutaneously in female SCID mice (6–8 weeks old, SHO-PRKDC SCID HR/HR1EW 43375; 4 mice in each group, SPF vivarium of the Institute of Cytology and Genetics SB RAS, Novosibirsk, Russia) as previously described[112]. We used female mice in this study, because females mice are more stable and less aggressive in terms of behavior despite the fact that they have the estrous cycle. Moreover, our object of our study is ovarian cancer, which represents gynecological disease. 70 days after tumor cell injections, mice were sacrificed, and tumor development was analyzed.

Tumor volumes were calculated using the formula V= ½ (length × width²). For subcutaneous tumors, the maximum size allowed for a mouse is 20 mm in diameter. If an animal has more than one tumor, these sizes are the maximum allowable sizes for all tumors combined.

## Survival assay

Cancer cells were plated on 96-well plates (5000 or 10,000 cells per well) and allowed to adhere overnight. Cells were then incubated with or without cisplatin, etoposide (Cell Signaling Technologies, 2200S), doxorubicin (Sigma-Aldrich, D1515), staurosporin (Cell Signaling Technologies, 9953) or paclitaxel (Cell Signaling Technologies, 9807S) for an indicated period. Cell viability was determined with MTT reagent (Sigma-Aldrich, M5655) using the iMark Microplate Reader (Bio-Rad, USA). All experiments were performed at least 3 times.

For caspase 3/7 activity assay, cells were stained with CellEvent Caspase-3/7 Green Flow Cytometry Assay Kit (Thermo Fisher Scientific, #C10427) according to the manufacturer's protocol and analyzed by NovoCyte Flow Cytometer (ACEA Biosciences, USA).

## Wound healing assay

Wound healing (scratch) assay was performed as described previously[5,113]. Briefly, ovarian cancer cells were seeded on 6-well plates, and conditioned media/ascitic fluids (25% v/v) were added for 3 days. A gap was generated using a tip, cells were washed, and a growth medium was added. Images were acquired using microscope «IX53» (Olympus, Japan) after an indicated period. All experiments were performed independently at least twice.

## Comet assay

SKOV3 cells were incubated with control or therapy-induced secretomes for 3 days and then treated with 10 μM of cisplatin. After 48 h, 3000 cells were processed for single-cell gel electrophoresis using the the guidelines provided by Trevigen (catalog # 4250-050-K). The images were captured using Nikon Eclipse Ts2 (Japan). Tail moment was defined as the product of the tail length and the fraction of total DNA in the tail (Tail moment = tail length x % of DNA in the tail) and was quantified using the CometScore pro (2.0.0.38) software (TriTek Corp.).

## Cell cycle analysis

SKOV3 cells were incubated with secretomes for 3 days and then treated with 10 μM of cisplatin for 24 h. Cells were fixed with ice-cold 70% ethanol at −20 °C overnight and incubated with DAPI solution and 0.1% Triton X-100 in PBS for 30 min. Flow cytometry was performed on NovoCyte Flow Cytometer with NovoExpress software (ACEA Biosciences, USA).

## Cell proliferation assay

Cell proliferation analysis was conducted using the xCELLigence RTCA DP instrument (ACEA Biosciences, USA), following the manufacturer's recommended protocol. SKOV3 cells overexpressing U6atac snRNA, U12 snRNA or GFP mRNA fragment were seeded on xCELLigence E-plate 16 (5000 cells per well) in standard growth medium. The electrical impedance values, represented as the cell index, were then recorded at 15-min intervals over a 5-day period.

## Ascites fractionation

For ascitic fluid fractionation by molecular weight, ascitic fluids were diluted twice with PBS and then centrifuged sequentially through 100, 30, 10, and 3 kDa molecular weight cut-off spin concentrators (Sartorius, #Z614297, #Z614246, #Z614211, #Z629405).

## RNA isolation

For RNA sequencing, total RNA from primary ovarian cancer cell cultures, SKOV3 cells, and fibroblasts was isolated using the RNeasy Mini Kit (Qiagen, #74104).

a) DNAse treatment of all samples (except samples of SKOV3 cells transfected with snRNAs) was carried out with TURBO DNA-free kit (Thermo Fisher Scientific, #AM2238), in volumes of 50 μl. Following DNAse treatment, RNA cleanup was performed with the Agencourt RNA Clean XP kit (Beckman Coulter, A66514).

b) Samples of SKOV3 cells transfected with snRNA were treated with DNAse I (Thermo Fisher Scientific, #EN0525). Subsequently, purification was carried out using the PureLink RNA Mini Kit (Thermo Fisher Scientific, #12183020).

The concentration and quality of the total RNA were assessed using the Quant-it RiboGreen RNA assay (Thermo Fisher Scientific, #R11490) and the RNA 6000 Pico chip (Agilent Technologies), respectively.

## RNA sequencing
a) Samples of primary ovarian cancer cell cultures and SKOV3 cells incubated with therapy-induced or control secretomes were analyzed using HiSeq 2500 Sequencing System (Illumina). Enrichment of polyadenylated RNA and library preparation was performed with NEBNext Poly(A) mRNA Magnetic Isolation Module and NEBNext Ultra II Directional RNA Library Prep Kit (NEB, #E7490S, #E7760S), respectively, according to the manufacturer's protocol. The library underwent a final cleanup using the Agencourt AMPure XP system (Beckman Coulter, #A63882) after which the libraries' size distribution and quality were assessed using a high-sensitivity DNA chip (Agilent Technologies). Libraries were subsequently quantified by Quant-iT DNA Assay Kit, High Sensitivity (ThermoFisher, #Q33120). Finally, equimolar quantities of all libraries (12 pM) were sequenced by a high throughput run on the Illumina HiSeq 2500 using 2 × 100 bp paired-end reads and a 5% Phix spike-in control.

b) Samples of SKOV3 cells overexpressing spliceosomal proteins or transfected with snRNAs, were analyzed on the NextSeq 550 Sequencing System (Illumina). Enrichment of polyadenylated RNA and library preparation was performed with NEBNext Poly(A) mRNA Magnetic Isolation Module and NEBNext Ultra Directional RNA Library Prep Kit (NEB, #E7490S, #E7420S), respectively, according to the manufacturer's protocol. Complementary DNA libraries for paired-end sequencing were prepared using NEBNext® Ultra Directional according to the manufacturer's protocol. Samples were sequenced with a NextSeq® 500/550 High Output Kit v2 (Illumina, #FC-404-2005): 300 Cycles for SKOV3 cells overexpressing spliceosomal proteins; 75 Cycles for SKOV3 cells transfected with snRNAs.

The raw sequence data and processed data have been deposited in the Gene Expression Omnibus the following accession numbers: GSE241908; GSE241909; GSE241910; GSE241912; GSE241913; GSE241914.

## Analysis of RNAseq data
a) For all samples (except samples of SKOV3 cells transfected with snRNAs) raw sequence data were initially converted to the FASTQ format using "bcl2fastq" software (Illumina). Subsequently, sequenced reads underwent a preprocessing step, which included trimming for adaptor sequences and quality control using "Trim Galore" (v.0.5.0). Trimmed RNAseq reads were then quantified against the *Homo sapiens* GRCh38.13 genome annotation at the transcript level employing the "Salmon" software (v. 1.4)[114]. The results were aggregated to the gene level using the "R" package "tximport"[115]. Datasets were filtered to remove rows with only a single count across all samples and differentially expressed genes were identified using the "R" package "DESeq2"[116]. Principal Component Analysis (PCA) was carried out using the "FactoMineR" package[117], and data visualization was achieved using the "ggplot2" package[118].

b) For samples of SKOV3 cells transfected with snRNAs, the raw sequence data were converted to the FASTQ format using "bcl2fastq" software (Illumina). For enhanced read quality before mapping, paired

reads underwent trimming using "Trimmomatic" (v.0.35)[119]. Trimmed RNAseq reads were quantified against the *Homo Sapiens* GRCh37.p13 genome annotation employing the "STAR" (v.2.5.2b) software. Gene read quantification was conducted using "HTSeq-count" (v.0.6.0) package. Differentially expressed genes were identified through the application of the "R" package "edgeR".

## Analysis of public RNA-Seq data
Unprocessed RNA-Seq samples from both cisplatin-resistant and cisplatin-sensitive cancer cell lines were obtained from the NCBI GEO data repository with the following accession numbers: GSE148003, GSE98559, GSE98230, GSE173201. Gene expression levels and differentially expressed genes were calculated for each dataset using the same methodology as employed for the RNA-Seq data we generated. To mitigate batch effects associated with different studies, we applied the ComBat algorithm from the "sva" package in R. Additionally, the relationships between samples were assessed through principal component analysis of DESeq2 rlog-transformed counts, utilizing the "FactoMineR" package in R.

## Protein concentration determination
Protein concentrations were determined using Quick Start Bradford Protein Assay (Bio-Rad, #5000201) or BCA Protein Assay Kit (Thermo Scientific, #23225) following the standard protocol provided by the respective manufacturers. Bovine Serum Albumin (BSA) served as the reference standard.

## LC-MS/MS analysis
**a) Sample preparation for cancer cells.** Cancer cells, incubated for three days with secretomes or primary cell cultures, incubated with autologous ascites were collected, washed three times with PBS, and then lysed using 4% SDS, and 50 mM TRIS-HCl (pH = 8) with protease inhibitors as described previously[120]. The cell lysates were subjected to sonication on ice (3 cycles: 10 s on/off pulses with a 30% amplitude). Disulfide bonds of each sample were reduced with DTT (final concentration 5 mM) for 30 min at room temperature (RT). Afterwards, iodoacetamide was added to a final concentration of 10 mM. The samples were incubated in the dark at RT for 20 min, with the reaction being stopped by the addition of DTT up to the final concentration of 5 mM. After precipitation of proteins using methanol/chloroform, the semi-dry protein pellet was dissolved in 50 μL of 8 M urea, 2 M thiourea, 10 mM TRIS-HCl (pH = 8). Protein concentration was then measured for each sample using Bradford Kit (Bio-Rad, #5000201), and then samples were diluted by the addition of 50 mM ammonium bicarbonate solution to reduce urea concentration to 2 M Trypsin (Promega, #V511A) was added at the ratio of 1:100 w/w and the samples were incubated for 14 h at 37 °C. After that, the reaction was stopped by the addition of formic acid up to a final concentration of 5%. Finally, the tryptic peptides were desalted using SDB-RPS membrane (Sigma-Aldrich, 66886-U), vacuum-dried, and stored at −80 °C before LC-MS/MS analysis. Prior to LC-MS/MS analysis, samples were redissolved in 5% ACN with 0.1% TFA solution and sonicated.

**b) Sample preparation for ascitic fluids.** Ten pairs of ascitic fluids from ovarian cancer patients before and after chemotherapy were treated with Combinatorial Peptide Ligand Library (CPLL) (Bio-Rad, #1633007) as described previously[22]. Briefly, reagent and column preparation, sample binding, and sample washes were carried out according to the manufacturer's protocol (ProteoMiner Protein Enrichment Kit, Bio-Rad, #1633007). The bound proteins were eluted three times with 100 μl of 4 different elution solutions: (i) 1 M NaCl, 20 mM HEPES, pH = 7.5; (ii) 200 mM glycine-HCl, pH = 2.4; (iii) 60% ethylene glycol; and (iv) 13.3% isopropyl alcohol, 7% ACN, 0.1% TFA. The acidic protein fraction was neutralized by 3 M TRIS-HCl immediately after elution.

All fractions were pooled together and added to a 5-kDa molecular weight cut-off spin concentrator (Agilent Technologies, #5185-5991) for buffer exchange. The concentration of each sample was measured, and then solubilization buffer (8 M urea, 2 M thiourea, 10 mM TRIS-HCl, (pH = 8)) was added to 250 μg of each sample at a 1:3 ratio. The samples were incubated at RT for 30 min and then subjected to in-solution protein digestion (as indicated in the section "Sample preparation for cancer cells").

**c) Sample preparation for secretome preparation.** For LC-MS/MS analysis, cell secretomes (FBS-free and phenol-red-free conditioned media) were collected by aspiration (from T175 flasks), a protease inhibitor cocktail (Sigma-Aldrich, GE80-6501-23) was added to each sample, and then the samples were centrifuged at 500 $g$ for 10 min to remove detached cells. Subsequently, the supernatants were immediately frozen and subjected to lyophilization to reduce the volume of the secretome. The lyophilizates were reconstituted in a buffer containing 6 M Gd-HCl, 10 mM TRIS-HCl (pH = 8), and 2 mM DTT for 30 min. To precipitate the insoluble fraction, the solutions were centrifuged at 16,000 $g$ for 10 min at 4 °C. Each supernatant was transferred to a 5-kDa molecular weight cutoff spin concentrator (Corning Spin-X UF6, Sigma-Aldrich, #431482) for buffer exchange. The protein concentration of each sample was determined, and all material of each sample was subjected to in-solution protein digestion (as indicated in the section "Sample preparation for cancer cells").

**d) Cell fractionation for proteomic analysis.** Cytoplasm and nuclei were prepared from SKOV3 cells as described previously[121] with slight modifications. Briefly, cells were treated or untreated with cisplatin (40 μM) for 7 h, and then cells were washed three times with PBS and incubated for an additional 41 h. Cells were washed three times with PBS and subsequently resuspended in 5 ml of buffer A, which contained 10 mM HEPES (pH = 7.9), 1.5 mM MgCl2, 10 mM NaCl, 0.5 mM DTT, and protease inhibitor cocktail (Sigma-Aldrich, GE80-6501-23). After that, cells were incubated with Triton X-100, bringing it to a final concentration of 0.5%, and 0.25 M sucrose on ice for 10 min. During this step, the integrity of the cytoplasmic membrane was disrupted, and the release of the nuclei was monitored under a microscope. Following incubation, the solution was centrifuged at 228 $g$ and 4 °C for 5 min. The supernatant represents the cytoplasmic fraction. Laemmli buffer was added to the cytoplasmic fraction for protein denaturation. The nuclear pellet was resuspended in 0.5 ml buffer A and washed twice to obtain clean, pelleted nuclei. These nuclei were then resuspended in Laemmli buffer and subjected to sonication for 3 rounds of 15 s each. Protein concentrations for both the cytoplasmic and nuclear fractions were determined using the BCA Protein Assay Kit.

Equal amounts of biological samples (250 μg each) were separated by 9% (w/v) SDS-PAGE (20 cm × 20 cm). Electrophoresis was stopped once the dye front reached a point 5 cm below the stacking gel, as described previously[22]. Briefly, gel bands were cut into small pieces (1 mm × 1 mm) and transferred to sample tubes. Protein disulfide bonds were reduced with 10 mM DTT in 100 mM ammonium bicarbonate buffer at 50 °C for 30 min and, afterwards, alkylated with 55 mM iodoacetamide in 100 mM ammonium bicarbonate buffer at RT for 20 min in the dark. After alkylation, gel samples were de-stained with 50% ACN in 50 mM ammonium bicarbonate buffer and then dehydrated by adding 100% ACN. Following the removal of the 100% ACN, the samples were subjected to in-gel trypsin digestion. The digestion buffer containing 13 ng/μl trypsin in a 50 mM ammonium bicarbonate buffer was added to the gel pieces. Trypsin digestion was carried out overnight at 37 °C. The resulting tryptic peptides were then extracted from the gel by adding 2 volumes of 0.5% TFA to the samples (incubation for 1 h) and then 2 volumes of 50% ACN (incubation for 1 h). Finally, the extracted peptides were vacuum-dried and re-dissolved in 3% ACN with 0.1% TFA prior to LC-MS/MS analysis.

**e) SILAC experiment.** For SKOV3 labeling, the DMEM medium deficient in l-lysine and l-arginine (Thermo Fisher Scientific, #88364) was prepared, supplemented with either l-lysine-2hcl (82 mg/l, thermo fisher scientific, #1860968) and l-arginine-hcl (50 mg/l, thermo fisher scientific, #1860970) or 13c6 l-lysine-2HCl (82 mg/L, Thermo Fisher Scientific, #1860969) and 13C6,15N4 l-arginine-HCl (50 mg/L, Thermo Fisher Scientific, #89990) to make light or heavy labeling media, respectively. Cells were cultured as described in the "Cell culture" section with 10% dialyzed FBS (Invitrogen, #A3382001). At passage 10, cell lysates were taken to determine > 95% of SILAC incorporation. Heavy-labeled donor cells were subjected to treatment with 40 μM cisplatin (to generate apoptotic vesicles) or left untreated (to generate control vesicles). After 7 h, cisplatin-containing and control media were removed, and the adherent cells were washed thoroughly four times with PBS). Then, heavy-labeled donor cells were incubated for 41 h in a fresh FBS-free medium. Then extracellular vesicles were isolated from conditioned media by centrifugation for 120 min at 100,000 $g$ in a Ti60 rotor (Beckman) at 4 °C. The vesicles were washed twice with PBS and then added to light-labeled SKOV3 recipient cells. After 10 or 48 h, recipient cells were washed 3 times with PBS, lysed in Laemmli buffer, sonicated, separated by preparative SDS-PAGE, and subjected to in-gel trypsin digestion as indacated in the "Cell Fractionation for Proteomic Analysis" section. The analysis of heavy-labeled proteins revealed which proteins from extracellular vesicles entered the recipient cells.

**f) Mass-spectrometry analysis.** Proteomic analysis of all samples (expect SKOV3 cells and fibroblasts secretomes from and cell fractionation samples) was performed using Q Exactive HF mass-spectrometer. Samples were loaded onto 50-cm columns packed in-house with C18 3 μM Acclaim PepMap 100 (Thermo Fisher Scientific), using Ultimate 3000 Nano LC System (Thermo Fisher Scientific) coupled to a mass-spectrometer (Q Exactive HF, Thermo Fisher Scientific). Peptides were loaded onto the column thermostatically controlled at 40 °C in buffer A (0.2% Formic acid) and then eluted with a linear gradient of 4–55% buffer B (0.1% formic acid, 80% ACN) in buffer A over 120 min at a flow rate of 350 nl/min. Mass-spectrometry data was stored during the automatic switching between MS1 scans and up to 15 MS/MS scans (topN method). The target value for MS1 scanning was set to $3 \times 10^6$ in the range 300–1200 $m/z$ with a maximum ion injection time of 60 ms and resolution of 60,000. Precursor ions were isolated at a window width of 1.4 $m/z$ and fixed the first mass of 100.0 $m/z$. Precursor ions were fragmented by high-energy dissociation in a C-trap with a normalized collision energy of 28 eV. MS/MS scans were saved with a resolution of 15,000 at 400 $m/z$ and at a value of $1 \times 10^5$ for target ions in the range of 200–2000 $m/z$ with a maximum ion injection time of 30 ms.

Proteomic analysis of SKOV3 cells and fibroblasts secretomes from and cell fractionation samples was carried out on TripleTOF 5600+ mass spectrometer with NanoSpray III ion source (AB Sciex, USA) coupled with NanoLC Ultra 2D+ nano-HPLC system (Eksigent, USA). The HPLC system was configured in a trap-elute mode. Sample loading buffer and buffer A consisted of a mixture of 98.9% water, 1% methanol, and 0.1% formic acid (v/v). Buffer B consisted of 99.9% ACN and 0.1% formic acid (v/v). Samples were loaded onto a Chrom XP C18 trap column (3 μm, 120 Å, 350 μm × 0.5 mm; Eksigent) at a flow rate of 3 μl/min for 10 min and eluted through a 3C18-CL-120 separation column (3 μm, 120 Å, 75 μm × 150 mm; Eksigent, USA) at a flow rate of 300 nl/min. The gradient included 5–40% buffer B in 90 min, followed by 10 min at 95% buffer B and 20 min of re-equilibration with 5% buffer B. Two blank 45-mine runs with 5–8 min waves (5% buffer B, 95%, 95%, 5%) were conducted between different samples to wash the system and prevent carryover. The information-dependent mass spectrometry experiment included 1 survey MS1 scan followed by 50 dependent MS2 scans. MS1 acquisition parameters were as follows: mass range for

MS2 analysis 300–1250 $m/z$; signal accumulation time 250 ms. Ions for MS2 analysis were selected based on intensity intensity with a threshold of 200 counts per second and a charge state from 2 to 5. MS2 acquisition parameters were as follows: the resolution of the quadrupole set to UNIT (0.7 Da); measurement mass range 200–1800 $m/z$; signal accumulation time 50 ms for each parent ion. Collision-activated dissociation was performed with nitrogen gas with the collision energy ramped from 25 to 55 V within the signal accumulation time of 50 ms. Analyzed parent ions were sent to the dynamic exclusion list for 15 s in order to get MS2 spectra at the chromatographic peak apex. To calibrate the mass spectrometer and ensure system performance, stability, and reproducibility, a β-Galactosidase tryptic solution (20 fmol) was run with a 15-min gradient of 5–25% buffer B between every 2 samples and between sets of samples.

**g) Protein identification and spectral counting.** Raw LC-MS/MS data from Q Exactive HF mass spectrometer were converted to.mgf peaklists with MSConvert from the ProteoWizard Software Foundation. The following parameters were used: "--mgf --filter peakPicking true [1,2]". Raw LC-MS/MS data from TripleTOF 5600+ mass spectrometer were converted to.mgf peaklists with ProteinPilot (version 4.5). The identification mode in ProteinPilot was used with parameters including Cys alkylation by iodoacetamide, trypsin digestion, TripleTOF 5600 instrument, and a thorough I.D. search with a detected protein threshold of 95.0% against the UniProt human protein knowledgebase.

For thorough protein identification, the generated peaklists were searched with MASCOT (version 2.5.1) and X! Tandem (ALANINE, 2017.02.01) search engines. The search was conducted against the UniProt human protein knowledgebase with the concatenated reverse decoy dataset. The precursor and fragment mass tolerance were set at 20 ppm and 0.04 Da, respectively. Database search parameters included the following: tryptic digestion with 1 possible missed cleavage; static modification for carbamidomethyl (C); dynamic/flexible modifications for oxidation (M). For X! Tandem, additional parameters were selected to check for protein N-terminal residue acetylation, peptide N-terminal glutamine ammonia loss, or peptide N-terminal glutamic acid water loss. Result files from the search engines were then submitted to Scaffold 4 software (version 4.0.7) for validation and meta-analysis. We used the local false discovery rate scoring algorithm with standard experiment-wide protein grouping. For the evaluation of peptide and protein hits, a false discovery rate of 5% was selected for both. False positive identifications were based on reverse database analysis. In Scaffold, protein annotation preferences were set to highlight Swiss-Prot accessions among others in protein groups.

For protein quantification, raw LC-MS/MS data were analyzed with MaxQuant (version 1.6.10.43) against the UniProt human protein knowledgebase. For this procedure, we used the following parameters: Orbitrap instrument type; tryptic digestion with 2 possible missed cleavages; fixed modification for carbamidomethyl (C); variable modifications for oxidation (M) and acetyl (protein N-term); LFQ label-free quantification. For SILAC analysis, we also included the following parameters: Arg10 and Lys6 as heavy labels; maximum 3 labeled amino acids.

**Quantitative statistical analysis of ascites proteome data**
We employed the statistical method "Limma"[122] to perform differential analysis of protein abundance, which was previously used for quantitative proteomic data. PCA was performed using the "R" package "FactoMineR". Hierarchical clustering was computed using Euclidean distance with the "R" function "hclust."

**Pathway analysis**
For functional annotation of differentially expressed genes/proteins, we used KEGG, Gene Ontology, and Reactome databases together with the "clusterprofiler"[123] and "ReactomePA"[124] packages in "R/

Bioconductor". A p-value correction for multiple testing was applied using the False Discovery Rate (FDR) method, with a cut-off threshold of 0.05. Additionally, we conducted gene expression signature analysis using the "signatureSearch" packages in "R" against the Reactome and Gene Ontology (GO) databases.

**Determination of cisplatin in secretomes by LC-MS/MS**
To check the absence of cisplatin in collected therapy-induced secretomes (TIS) we collected secretomes from SKOV3 donor cells treated or untreated with cisplatin as described in the "Secretome Generation" section in two biological replicates. SKOV3 cells were incubated with FBS-free standard medium with or without 40 µM cisplatin. After 7 h, cisplatin-containing and control media were removed, and the adherent cells were washed thoroughly four times with PBS. Then, cells were incubated for 41 h in a fresh FBS-free medium. Then, conditioned media were collected, centrifuged for 10 min at 500 $g$ at 4 °C in the F-45–24-11 rotor (Eppendorf, Germany) to remove cells. The supernatants were concentrated up to 100 µL using a 100-kDa molecular weight cut-off spin concentrators (Amicon, Merck, UFC910024) to enrich with extracellular vesicles.

Then, 50 µL of 50 mM sodium diethyldithiocarbamate (DDTC) solution in 0.1 N NaOH and 10 µL of ACN containing internal standard were added to 100 µL of the sample. Samples were incubated for 35 min at 40 °C and 1200 rpm. 700 µL of cold ACN (−20 °C) was added to each sample followed by incubation for 2 min at RT and 1200 rpm. Then, the samples were subjected to centrifugation (16,000 $g$) for 5 min at 15 °C. 700 µL of the supernatant was taken from the sample and lyophilized. The samples were reconstituted in 50 µL of water:ACN (1:1) solution acidified with 0.1% formic acid. The resulting samples were subjected to the centrifugation (16,000 $g$) for 5 min at 15 °C. Then 40 µL of the supernatant was transferred to a marked vial. Verapamil (25 ng/mL) was added to the samples as an internal standard; palladium acetate (10 ng/mL) was added to the samples as an internal response standard. The studied samples were prepared in 2 biological repeats, each of which was analyzed in two technical repeats.

Calibration samples and quality control samples were obtained by adding calculated amounts of cisplatin standard directly to the cell culture media. The cell culture media without the addition of cisplatin was used as a blank sample. The sample preparation for calibration, quality control and blank samples was like the samples under study. The calibration curve for cisplatin was constructed in the range 12.5–312 nM/L ($R^2 = 0.9$).

A tandem time-of-flight mass spectrometer (SCIEX 6600) combined with a liquid chromatograph (ExionLC AD) was used for the analysis. Source settings were: GS1 = 55; GS2 = 55; CUR = 35; TEM = 450 °C; ISFV = 5500 V; an ESI probe was used for sample submission. Ion detection of the studied ions was carried out in the TOF MS mode and in the ProductIon mode for the following target masses: 492.0, 640.0, 403.0.

Chromatographic separation of the test sample's components was carried out in RPLC mode using a Phenomenex PhenylHexyl 4.6 × 50 mm chromatographic column with 2.6 µm particles: phase A (water; 5 mm ammonium formate); phase B (MeOH; 0.1% FA); flow rate 0.45 mL/min; sample input volume 20 µL. The chromatographic gradient conditions are shown in Supplementary Data 1, sheet 4. According to published data[125,126], the predominant signal was obtained for a three-substituted complex [Pt(DDTC)3]+ with $m/z = 641.0477$, therefore observations and calculations were carried out for [Pt(DDTC)3]+.

**Quantitative real-time PCR**
For quantitative RT-PCR experiments, total RNA was extracted from various sources, including cancer cells, acellular fractions from ascites, or conditioned media. The RNA extraction was performed using Trizol reagent (Invitrogen, #15596018), followed by DNase I treatment

(Thermo Fisher Scientific, #EN0525) and purification using the RNeasy Mini Kit (Qiagen, #74104). RNA concentration was determined using either a Nanodrop 2000 Spectrophotometer (Thermo Fisher Scientific) or a Qubit Fluorometer (Thermo Fisher Scientific). Subsequently, cDNA was synthesized using the iScript reverse transcription supermix (Bio-Rad, #1708841) following the manufacturer's protocol. For evaluating snRNA levels in ascites and conditioned media samples, RT-qPCR was conducted in triplicate with BioMaster RT-PCR SYBR Blue (Biolabmix, RM04-200) on the LightCycler 96 System (Roche, Switzerland). Data analysis was performed using the qbase+ software, version 3.1 (Biogazelle), which includes the selection of stable reference genes with geNORM, quality control, and relative quantification of mRNA levels of genes[127,128]. The results represent mean values (± standard deviation) from three independent experiments, and PCR product specificity was confirmed by visualizing DNA on 1.5% agarose gel following PCR. GAPDH, 18S rRNA, or a synthetic fragment of GFP mRNA (89 bases) was used as an internal control for data normalization.

To verify the overexpression of SNU13 or SYNCRIP in SKOV3 cells, RT-qPCR reactions were carried out using qPCRmix-HS SYBR (Evrogen, #PK147L) on the CFX96 Touch Real-Time PCR Detection System (Bio-Rad, USA). All samples were tested in triplicate, and GAPDH was used as a reference gene for data normalization. Primer sequences are provided in Supplementary Data 1, Sheet 3.

### Proteins digestion with Proteinase K

Extracellular vesicles were isolated as described in the "SILAC Experiment." Subsequently, each sample was divided equally by volume into 4 parts, including untreated control, 5 ng Proteinase K treatment, 10 ng Proteinase K treatment, and 5 ng Proteinase K treatment with 4% SDS. Samples were incubated with Proteinase K (Invitrogen, 25530049) and 4% SDS (if needed) for 20 min at 37 °C. The reaction was stopped by adding PMSF (Roche, 10837091001) up to 5 mM, and samples were immediately subjected to western blot analysis.

### Western blotting

a) Extracellular fluids (ovarian cancer ascites and SKOV3 secretomes) were treated at 50 °C for 30 min with RIPA buffer containing 1% protease inhibitor cocktail. After treatment, samples were pre-cleaned by centrifugation at 16,000 g for 15 min at 4 °C. Protein concentrations were determined using the BCA Protein Assay Kit. Equal volumes of each sample were then analyzed by 10% SDS-PAGE and transferred to a PVDF membrane (Millipore, IPVH20200). The membrane was blocked with either 5% Blotting Grade Blocker Nonfat Dry Milk (Bio-Rad) for SKOV3 secretomes or Immobilon Block-CH (Millipore, WBAVDCH01) for ascites samples, both for 1 h. Subsequently, the membranes were incubated overnight with primary antibodies against various proteins: SRSF1 (Invitrogen, #32-4500, 1:250), HNRNPD (Cell Signaling Technology, #12382, 1:1000), TIA1 (Invitrogen, MA5-26474, 1:2000), U2AF65 (GeneTex, GTX55828, 1:500), SRSF2 (Abcam, ab204916, 1:1000), HNRNPM (Novus Biologicals, NB200-314, 1:500), SNU13 (Abcam, ab181982, 1:1000), SRSF3 (Abcam, ab198291, 1:1000), or U2AF1 (Abcam, ab172614, 1:1000). The following incubation was with peroxidase-conjugated secondary antibodies (Invitrogen, G-21234, G-21040, 1:40,000) for 1 h. Immunolabeled proteins were detected using the ChemiDoc MP Gel Imaging System (Bio-Rad, USA).

b) Lysates were pre-cleaned by centrifugation at 16,000 g for 15 min at 4 °C. Protein concentrations were determined using the Bradford assay. Equal amounts of protein samples (15 μg/lane) were analyzed by 12% SDS-PAGE and transferred to a PVDF membrane (Millipore, IPVH20200). The membrane was blocked with 5% Blotting Grade Blocker Nonfat Dry Milk (Bio-Rad, #1706404) for 1 h and then incubated overnight with primary antibodies against CD63 (Abcam, ab134045, 1:1000), LIG1 (Abcam, ab177946, 1:1000), TDP1(Cell Signaling, #59710, 1:1000), FOXM1 (Cell Signaling Technology, #5436,

1:1000), RAD51 (Abcam, ab213, 1:200), p53 (Cell Signaling Technology, #2524, 1:1000), TIA1 (Invitrogen, MA5-26474, 1:2000), SNU13 (Abcam, ab181982, 1:1000), SRSF4 (Novus Biologicals, NBP2-04144, 1:2000) or SYNCRIP (HNRNPQ/R, Cell Signaling Technology, #8588, 1:500). After the primary antibody incubation, peroxidase-conjugated secondary antibodies (Invitrogen) were used for a 1-h incubation. Antibodies against GAPDH (HyTest, 5G4/5G4cc, 1:1000), Lamin B (Cloud-Clone Corp., PAF548Hu01, 1:1000) or β-Tubulin (Cloud-Clone Corp., PAB870Hu01, 1:500) were employed as loading controls. Immunolabeled proteins were detected by ChemiDoc MP Gel Imaging System (Bio-Rad, USA).

c) Secretomes after Proteinase K digestion were resuspended in Laemmli buffer and heated to 95 °C for 5 min. The samples were then analyzed by 12% SDS-PAGE and transferred to a PVDF membrane (Millipore, IPVH20200). Following a 1-h blocking step with 5% Blotting Grade Blocker Nonfat Dry Milk (Bio-Rad, #1706404), the membrane was incubated overnight with primary antibodies against DHX9 (Abcam, ab183731, 1:10000), SYNCRIP (HNRNPQ/R Cell Signaling Technology, #8588, 1:500), SRSF2 (Abcam, ab204916, 1:1000), or SRSF3 (Abcam, ab198291, 1:1000). Subsequently, peroxidase-conjugated secondary antibodies (Invitrogen) were used for a 1-h incubation. Immunolabeled proteins were detected by ChemiDoc MP Gel Imaging System (Bio-Rad, USA).

### Immunodepletion of ascitic fluids

Pool of ascitic fluids (100 μL) from three patients after therapy (Supplementary Data 1, Sheet 2) were diluted three times with PBS. To deplete spliceosomal snRNPs from ascites, a mix of antibodies to several spliceosomal proteins (U2AF1 (Abcam, ab172614, 1/10), SF3B1 (Abcam, ab170854, 1/100), and PRPF8 (Abcam, ab79237, 5 μg/ml) or an antibody to the CD63 (Santa Cruz Biotechnology; Cat. #sc-5275, 1/100) protein were immobilized on 50 μl of Protein A/G Magnetic Beads (Thermo Fisher Scientific, #88802) according to the manufacturer's protocol. The beads were washed 3 times with PBS and incubated for 1 h with 300 μL of pre-cleaned, diluted ascitic fluids under constant agitation. Next, supernatants were collected. To assess the presence of spliceosomal snRNAs in depleted ascitic fluids, total RNA from supernatants was isolated using Trizol reagent and samples subjected to quantitative Real-Time PCR.

### Extracellular vesicle isolation

Extracellular vesicles from ascitic fluid samples were purified by differential centrifugation as described previously[129] with some modifications. For each sample, 1 ml of ascitic fluid was centrifuged for 15 min at 500 g, then for 30 min at 10,000 g in an F-45-24-11 rotor (Eppendorf, Germany) at 4 °C. To isolate extracellular vesicles, 1 mL of supernatant was diluted with 1 mL of PBS, concentrated on a 1,000-kDa molecular weight cut-off spin concentrator (Sartorius, #VS0161), and washed twice with PBS using the same concentrator.

Additional vesicle purification was performed via 3-step sucrose gradient (48%, 40%, and 20% w/v) centrifugation in an MLS-50 rotor (Beckman, USA). The vesicles from the spin concentrators were diluted in 62% sucrose up to a final concentration of 48% (w/v) sucrose in 600 μl. Then 500 ml of 40% sucrose were overloaded on the first layer. The third low-density layer was formed by 4 ml of 20% sucrose. The samples were centrifuged at 50,000 rpm (205,000 g) for 3 h in an MLS-50 rotor (Beckman) at 4 °C. Prior to use, 62% sucrose was purified on a 100-kDa molecular weight cut-off spin concentrator (Amicon, Merck, UFC910024) to remove vesicle-like particle contamination of sucrose. The extracellular vesicles floated on 40% sucrose, and the vesicles concentrated on the border between the 40% and 48% sucrose layers were collected, washed twice with PBS, and concentrated using 100-kDa filters (Amicon, Merck).

For isolation of extracellular vesicles from in vitro secretomes, 10 × 10⁶ SKOV3 cells were treated or not with cisplatin as indicated in

the section "Secretome Generation". After 24 h, conditioned media were centrifuged for 10 min at 500 g in A-4-44 swinging-bucket rotor (Eppendorf, Germany). Next, the supernatants were concentrated on a 100-kDa molecular weight cut-off spin concentrator (Amicon, Merck) and washed twice with PBS. Finally, vesicles diluted in 100 μl PBS were analyzed by NTA.

## Nanoparticle tracking analysis

Particle size distribution and concentration were measured with nanoparticle tracking analysis (NTA) using Nanosight LM10 HS-BF instrument (Nanosight Ltd, UK) equipped with the following options: 405 nm, 65 mW LM12 laser unit with no temperature control; high sensitivity camera of an EMCCD type. Measurements were performed according to ASTM E2834-12 (ISO 19430:2016) using camera and video processing setups optimized for extracellular vesicles. We used a previously described instrument configuration[130] with minor adjustments: higher threshold and detection threshold (10,920 and 8 instead of 11,180 and 9, correspondingly) for better detection of small particles in the measured sample set. Briefly, samples were diluted with particle-free PBS (pH 7.4) to reach the optimal concentration for nanoparticle tracking analysis. Measurements were done in several repeats (12 to 14) to get at least 5,000 tracks in total. Data from all repeats were joined to get the particle size distribution and total particle concentration; both were corrected for dilution factor used for each sample.

## Flow cytometry analysis of extracellular vesicles

SKOV3 cells were transfected with pTagGFP2-SNU13 or pTagGFP2-SYNCRIP on 25 cm² cell culture flasks (two flasks per subline). After 2 days, cells were treated with 40 μM cisplatin for 7 h, then washed three times with PBS and incubated additional 41 h in fresh FBS free and phenol-red-free DMEM medium. To isolate extracellular vesicles (EVs), cell culture supernatants were centrifuged at 500 g for 10 min to pellet dead cells and bulky debris in A-4-44 swinging-bucket rotor (Eppendorf, Germany). EVs were collected by ultrafiltration of supernatants with 100-kDa molecular weight cut-off spin concentrator (Amicon, Merck, UFC910024) according to manufacturer protocol and washed twice with PBS.

CD9-positive EVs were immune-selected with Exosome-Human CD9 Flow Detection Reagent (ThermoFisher Scientific; #10620D) according to the manufacturer protocol. CD9 Deanabeads (15 μl per probe) were incubated with collected EVs overnight at 4 °C. After incubation, EVs-coated magnetic beads were washed with 0.05% Tween 20 / PBS buffer. Some of the samples were stained against a specific protein marker of exosome antiCD81-APC (BD Biosciences, #551112, 10 μl per probe) for 2 h on a rotator at RT followed by flow analysis with BD LSRFortessa™ Cell Analyzer (BD, USA). Population of immunoisolated EVs are CD81 positive (Source Data file for Supplementary Fig. 5E). Red (PE channel) and green (FITC channel) fluorescence of the EVs-coated beads were registered for protein SRCF4-RFP and SYNCRIP-GFP or SNU13-GFP detection, correspondingly. The data has been processed with Kaluza software (Beckman Coulter, USA).

## In vitro synthesis of small nuclear RNAs

In vitro synthesis of U12 and U6atac small nuclear RNAs was performed as described previously[52]. Briefly, DNA templates for U12 and U6atac snRNA analogs were obtained by RT-PCR of total RNA from human cells with specific primers (Supplementary Data 1, Sheet 3) using one-step BioMaster RT-PCR-Color (Biolabmix, RMC02-200). A DNA template for the control fragment of GFP (89 bases) was a plasmid pSPT19 (Roche, 10 999 644 001) with a target fragment GFP, linearized at the site EcoR1. The amplification products were analyzed by electrophoresis in a 2.5% agarose gel, and DNA sequences were verified by Sanger sequencing. Artificial snRNAs were synthesized by in vitro transcription with "T7 RNA Synthesis Kit" (Biolabmix, T7-tr-20) according to the manufacturer's protocol. To obtain the modified analogs, we used

pseudouridine-5′-triphosphate (Ψ-UTP) instead of UTP; m⁷G[5′]ppp[5′]G monomethylated cap analog (for U6atac snRNA analog); m₃^{2,2,7}G[5′]ppp[5′]G trimethylated cap analog (for U12 snRNA analog) (Jena Bioscience, #NU-1139, #NU-852, #NU-853).

In vitro transcription products were purified from low molecular weight components using Illustra MicroSpin G-25 columns (GE Healthcare, 27-5325-01). The synthetic RNA solutions were treated with DNase I (Thermo Fisher Scientific) and alkaline phosphatase (Thermo Fisher Scientific, #EF0651). The ncRNA analogs were isolated by phenol/chloroform extraction using Lira Reagent (Biolabmix, LR-100). Finally, RNA transcripts were purified by ion-pair reverse-phase high-performance liquid chromatography (IPRP HPLC) and precipitated by ethanol with 0.6 M Sodium acetate. Synthetic RNAs were stored in a DEPC-treated water solution at −70 °C. RNA sequences were verified by reverse transcription and subsequent Sanger cDNA sequencing. The presence of modified monomers and their amounts were verified by HPLC-MS/MS and HPLC-UV of nucleosides after enzymatic digestion of artificial RNAs. Additionally, 5′- or 3′-terminally ³²P-labeled snRNA analogs were synthesized. To obtain ³²P-labeled analogs, we used T4 polynucleotide kinase (Thermo Fisher Scientific, #1351304) and ³²P-labeled ATP or ³²P-labeled pCp according to RNA ligase protocols (Laboratory of Biotechnology, ICBFM SB RAS, Novosibirsk, Russia).

RNA integrity was verified on 2.5% agarose or 8% denaturing polyacrylamide gels using visualization by SYBR Green I (Thermo Fisher Scientific, S7563) or 5′-[³²P]-label (Laboratory of Biotechnology, ICBFM SB RAS, Novosibirsk, Russia).

## Transient transfection

a) For RT-qPCR, RNAseq, and proteomic analyses, SKOV3 cells were transfected with analogs of U6atac snRNA, U12 snRNA, or GFP89 (10 nM each) using lipofectamine RNAiMax (Invitrogen, #13778075) according to the manufacturer's protocol. At 48 h, total RNA or proteins were isolated from the cells as indicated above.

b) For spliceosomal protein localization assessment, SKOV3 cells, seeded on 4-well chamber slides, were transfected with plasmids pTagRFP-C-SRSF4 and pEGFP-C1-TIA1 using Lipofectamine 3000 (Invitrogen, #L3000015) according to the manufacturer's protocol. After 24 h, cells were treated with cisplatin at a concentration of 40 μM. Re-localization of these proteins in response to cisplatin was monitored 10 h after cisplatin addition by fluorescence microscopy.

c) To confirm SILAC data, SKOV3 cells, seeded on T25 flasks, were transfected with plasmids pTagRFP-C-SRSF4 and pEGFP-C1-TIA1[131] using Lipofectamine 3000 (Invitrogen) according to the manufacturer's protocol. 24 h after transfection, cells were treated with cisplatin in concentration 40 μM; a conditioned medium was collected and processed as described in the "SILAC experiment" section. Therapy-induced or control secretomes were resuspended in complete DMEM and added to recipient SKOV3 cells for 10 h followed by fluorescence microscopy.

## Fluorescence microscopy

a) Cells, transfected with pTagRFP-C-SRSF4 and pEGFP-C1-TIA1, on chamber slides were fixed with 4% PFA for 15 min, washed twice with PBS, and then incubated with DAPI solution (Sigma-Aldrich, D9542) for 10 min. Slides were mounted with the Fluoroshield mounting medium (Sigma-Aldrich, F6937) and covered with cover glasses. Images were taken with the Nikon Eclipse Ni-E microscope in DAPI, FITC, and TRITC channels.

b) To confirm SILAC data, the nuclei of recipient SKOV3 cells were stained with a nuclear Hoechst dye (Bio-Rad, #1351304) according to the manufacturer's protocol. Images of cells without fixation were taken with an Olympus IX53F fluorescence microscope.

c) Cells, transfected with pTagGFP2-SNU13, pTagGFP2-SYNCRIP, and pTagGFP on chamber slides were fixed with 4% PFA for 15 min, washed thrice with PBS, to increase permeability added with 0.1%

Triton X-100 in water for 15 min, washed thrice with PBS, then incubated with Phalloidin conjugated with Alexa 555 (Thermo Fisher Scientific, A-34055; 1:400) for 90 min, washed thrice with PBS, then incubated with DAPI solution for 20 min. Slides were mounted with the Fluoroshield mounting medium (Sigma-Aldrich, #F6937) and covered with cover glasses. Images were taken with inverted Nikon A1 confocal microscope and via a Nikon Plan Apo À 40x (NA 0.95) objective using Nikon NIS-Elements software. Fluorescent signal of GFP was excited by 488 nm laser line, and its fluorescence was acquired within the spectral range 500-550 nm. Alexa 555 was excited by 561 nm laser line, and its fluorescence was acquired within the spectral range 570-620 nm. DAPI was excited by 450 nm and its fluorescence was acquired in range 425-475 nm.

d) Cells were incubated with control or therapy-induced secretomes for 3 days, followed by treatment with or without 25 μM of cisplatin for 3, 6, 9, or 24 h and fixed overnight with ice-cold 70% ethanol for cisplatin-DNA adducts staining or with 4% PFA (Sigma-Aldrich, 47608) for 15 min for $\gamma$H2AX and RPA32/RPA2 (phospho S33) staining with subsequent permeabilization with 0,1% Triton X-100 for 15 min. Next, cells were incubated with primary antibodies against $\gamma$H2AX (Sigma-Aldrich, 05-636, 1:200), RPA32/RPA2 (phospho S33) (Abcam, ab211877, 1:500) or against cisplatin modified DNA (Abcam, ab103261, 1:200) overnight. Then, cells were incubated with secondary antibodies Alexa Fluor 555 (Thermo Fisher Scientific, A-21428, 1:500) or Alexa Fluor 488 (Thermo Fisher Scientific, A-11001, 1:500) for 2 h, and then were stained with DAPI solution for 15 min. Slides were mounted with the Fluoroshield mounting medium (Sigma-Aldrich, F6937) and covered with cover glasses. Images were taken with the Nikon Eclipse Ni-E microscope in DAPI, FITC, and TRITC channels.

e) Cells, incubated with cisplatin (10 μM, 24 h) or control cells, on chamber slides were fixed with 4% PFA for 15 min, washed thrice with PBS, to increase permeability added with 0.1% Triton X-100 in water for 15 min, washed thrice with PBS, then incubated with primary antibodies SRSF1 (Thermo Fisher Scientific, A-34055) and anti-RPA32/RPA2 (phospho S33) (Abcam, ab21187) overnight. Then, cells were incubated with secondary antibodies Alexa Fluor 555 (Thermo Fisher Scientific, A-28180, 1:500) or Alexa Fluor 488 (Thermo Fisher Scientific, A-11008, 1:500) for 2 h, and then were stained with DAPI solution for 15 min. Slides were mounted with the Fluoroshield mounting medium (Sigma-Aldrich, F6937) and covered with cover glasses. Images were taken with the Nikon Eclipse Ni-E microscope in DAPI, FITC, and TRITC channels.

f) Primary cell cultures were seeded on chamber slides and fixed with 4% PFA for 15 min, then washed thrice with PBS. Next, cell were incubated with primary antibodies Anti-EpCam (Abcam, ab223582, GR3367015-9, 1:100), Anti-CA125 (Novus Bio, NBP2-59023, MAB-02920, 1:50) and Anti-CD44 (Sony, 2294010, 1:100) overnight. Then, cells were incubated with secondary antibodies Alexa Fluor 555 (Thermo Fisher Scientific, A-28180, 500) or Alexa Fluor 488 (Thermo Fisher Scientific, A-11008, 1:500) for 2 h, and then were stained with DAPI solution for 15 min. Slides were mounted with the Fluoroshield mounting medium (Sigma-Aldrich, F6937) and covered with cover glasses. Images were taken with the Nikon Eclipse Ni-E microscope in DAPI, FITC, and TRITC channels.

## Plasmid construction

The DNA fragment encoding SNU13 was amplified from SKOV3 cDNA by PCR technique using primer pair EcoRI-SNU13/SNU13-BamHI and cloned into EcoRI/BamHI sites of lentiviral plasmid pCDH-EF1-MCS-IRES-Puro (System Biosciences, CD532A-2) to generate pCDH-SNU13 plasmid or into EcoRI/BamHI sites of pTagGFP2-C plasmid (Evrogen, #FP191) to generate pTagGFP2-SNU13 plasmid. The DNA fragment encoding hnRNP Q1 (a.k.a. SYNCRIP) was amplified from SKOV3 cDNA by PCR technique using primer pair NheI-SYNCRIP/SYNCRIP-BstBI and cloned into NheI/BstBI sites of lentiviral plasmid pCDH-EF1-MCS-IRES-

Puro (System Biosciences, CD532A-2) to generate pCDH-SYNCRIP plasmid. The DNA fragment encoding SYNCRIP was amplified from SKOV3 cDNA by PCR technique using primer pair SalI-SYNCRIP/SYNCRIP-KpnI and cloned into SalI/KpnI sites of pTagGFP2-C plasmid (Evrogen, #FP191) to generate pTagGFP2-SYNCRIP plasmid. The absence of unwanted mutations in the inserts and vector-insert boundaries was verified by sequencing. Plasmid DNA was purified with Plasmid Midi Kit (Qiagen, #12145).

The DNA fragment encoding U6atac snRNA was made from synthetic oligos (U6atac 1/2/3F & 1/2/3R), with the following amplification using primer pair BsmBI-U6F/U6R-BamHI and cloned into BsmBI/BamHI sites of lentiviral plasmid lenti sgRNA(MS2)_puro backbone (Addgene, #73795) to generate lenti-U6atac plasmid. The control DNA fragment was amplified from plasmid pEGFP-N1 (Clontech) using primer pair BsmBI-gfpF/gfpR-BamHI and cloned into the same lentiviral backbone to generate lento-U6-GFP plasmid.

The DNA fragments encoding U2 promoter, U2 3'box and U12 snRNA were amplified from SKOV3 DNA by PCR technique using primer pairs NheI-U2F/U2 R2, U2 3box F2/ U2 3box R1, and U12F/U12R, respectively. DNA fragments were then assembled during PCR using primer pair NheI U2F/U2 3'box R2. The obtained PCR product included U2 promoter, a fragment encoding snRNA U12, and U2 3'box and was cloned into NheI/BamHI sites of lentiviral plasmid lenti sgRNA(MS2)_puro backbone (Addgene, #73795) to generate lenti-U12 plasmid. The control DNA fragment was amplified from plasmid pEGFP-N1 (Clontech) using primer pair gfp U2F/gfp U2R and cloned into the same lentiviral backbone to generate lenti-U2-GFP plasmid.

## Lentivirus production and transduction

To establish overexpression cell lines, Phoenix-GP packaging cells (ATCC, CRL-3215) were co-transfected with pCDH-EF1-MCS-IRES-Puro, pCDH-SYNCRIP, or pCDH-SNU13 lentiviral vectors, and two packaging plasmids psPAX2 and pMD2.G (Addgene, 12260, 12259). To establish knockdown cell lines, Phoenix-GP packaging cells (ATCC, CRL-3215) were co-transfected with the same packaging plasmids psPAX2 and pMD2.G and pLKO.1-SYNCRIP, or pLKO.1-SNU13 lentiviral vectors. SNU13 Mission® shRNA plasmid DNA in a pLKO.1 vector backbone (clone ID: TRCN0000074801), SYNCRIP Mission® shRNA plasmid DNA in a pLKO.1 vector backbone (clone ID: TRCN0000017270), and pLKO.1-puro control DNA were purchased from Sigma. Growth media were changed the following day and lentivirus-containing supernatants were harvested 72 h later. One day before transduction, SKOV3 were seeded on 25 cm² cell culture flasks so they will be 30% confluent at the time of transduction. The next day, SKOV3 cells were incubated with viral supernatants for 24 h in the presence of 10 μg/ml polybrene (Sigma-Aldrich, TR-1003-G). Two days after infection, transduced cells were selected with 1 μg/ml puromycin (Sigma-Aldrich, P7255) for 4 days.

## Materials availability

Plasmids generated in this study are available from the lead contact upon request.

## Statistics and reproducibility

All data are presented as mean ± SD. The number of replicates for each experiment was stated in the Figure legends and refers to independent biological replicates. Statistical differences between the two groups were evaluated by unpaired two-tailed $t$ test: $p$-value < 0.05 was considered statistically significant. No statistical method was used to predetermine the sample size. The sample size was determined on the basis of our previous studies and the experience of the authors[132]. No blinding or randomization was used. The Investigators were not blinded to allocation during experiments and outcome assessment. No data were excluded from the analyses.

**Reporting summary**

Further information on research design is available in the Nature Portfolio Reporting Summary linked to this article.

## Data availability

Publicly available datasets used in this study were obtained from the Gene Expression Omnibus (https://www.ncbi.nlm.nih.gov/geo/) database: GSE148003, GSE98559, GSE98230, GSE173201. All RNAseq data generated in this study have been deposited in the Gene Expression Omnibus database (GEO) under the accession numbers: GSE241908 - Transcriptomic profiles of ovarian cancer cells isolated from paired cancer ascites before and after chemotherapy); GSE241909 - Transcriptomic profiles of ovarian cancer cells incubated with autologous ascitic fluids from patients before and after chemotherapy; GSE241910 - Effect of therapy-induced secretomes on the transcriptome of ovarian cancer cells; GSE241912 - Effect of extracellular spliceosomal snRNAs on the transcriptome of ovarian cancer cells; GSE241913 - Transcriptome of ovarian cancer cells overexpressing SYNCRIP or SNU13); GSE241914 - This SuperSeries is composed of the SubSeries listed upper. All proteomic datasets have been deposited to the ProteomeXchange Consortium via the PRIDE partner repository with the dataset identifiers – PXD019327 - Proteome profiles of paired ovarian cancer ascites before and after chemotherapy; PXD019096 - Cisplatin-Induced Changes in Proteomic Profiles of Ovarian Cancer Cell and Fibroblast Secretomes; PXD019642 - Subcellular Relocalization of Proteins in Response to Cisplatin-Induced DNA Damage; PXD027948 - SILAC Strategy for Analysis of Proteins Released To the Extracellular Medium and Penetrated Into Recipient Cells; PXD027950 - The Role of Extracellular Spliceosomal snRNAs in Communication of Ovarian Cancer Cells; PXD027794 - Effect of therapy-induced secretomes on the proteome of ovarian cancer cells; PXD045647 - Effect of malignant ascites before and after therapy on the proteome of ovarian cancer cells; PXD045663 - Cisplatin-Induced Changes in Proteomic Profiles of Ovarian Cancer Cell and HaCaT Secretomes. All processed data are available in the article, Supplementary files, and Source Data. Source data are provided with this paper.

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

## Acknowledgements

This work was supported by grant 075-15-2019-1669 from the Ministry of Science and Higher Education of the Russian Federation (V.O.S., P.V.S., G.P.A., O.M.I., M.A.L., V.N.L., K.S.A., K.M.K. for RNAseq analyses); by the Russian Science Foundation project nos. 22-15-00462 (V.O.S., G.P.A, P.V.S., M.M.L., O.M.I. for proteomic analysis and experiments with cell cultures), 22-14-00234 (M.S.P. for splicing analysis), and 22-75-10153 (G.A.S. for RNA synthesis); by the State funding project 124031200004-7 (G.P.A. for bioinformatics analysis). We thank Dr. Roman Trikin, Dr. Eugen N. Imyanitov, and Dr. Sergey V. Razin for the critical reading and editing of the manuscript.

## Author contributions

Conceptualization: V.O.S., K.S.A., G.P.A., M.S.P., V.M.G., and M.A.L.; Data curation: V.O.S.; Formal analysis: V.O.S., G.P.A., K.S.A., and P.V.S.; Funding acquisition: V.O.S., G.A.S., M.S.P., V.M.G., and M.A.L.; Investigation: V.O.S., P.V.S., K.S.A., G.P.A., M.S.P., O.M.I., I.K.M., G.A.S., E.Zh., I.O.B., O.N.B., R.H.Z., K.M.K., V.A.V., T.V.G., S.Y.M., E.Kh., E.G.E., O.I.A., A.V.S., N.A.B., L.A.A., A.A.N., D.D.Kh., V.N.L., O.S.L., and A.N.B.; Methodology: V.O.S., K.S.A., G.P.A., G.A.S., E.Zh., A.A.N., P.V.S., I.O.B., O.N.B., R.H.Z., K.M.K., V.A.V., T.V.G., S.Y.M., E.Kh., I.Yu.P., A.A.M., E.G.E., M.M.L., Z.W., A.S.S., A.I.L., L.K.A.; Project administration: V.O.S. and V.M.G.; Resources: V.O.S., V.M.G., and M.A.L.; Software: K.S.A. and G.P.A.; Supervision: V.O.S., V.M.G., and M.A.L.; Validation: M.S.P., K.S.A., and G.A.S.; Visualization: K.S.A., V.O.S., and P.V.S.; Writing—original draft: V.O.S, M.S.P., K.S.A., P.V.S., and O.M.I.; Writing—review & editing: M.A.L and V.M.G. All authors have read and agreed to the published version of the manuscript.

## Competing interests

The authors declare no competing interests.

## Additional information

**Victoria O. Shender** ®[1,2,3] ✉, **Ksenia S. Anufrieva**[1,2], **Polina V. Shnaider**[1,2,4], **Georgij P. Arapidi** ®[1,2,3,5], **Marat S. Pavlyukov** ®[3], **Olga M. Ivanova** ®[1,2], **Irina K. Malyants**[2,6], **Grigory A. Stepanov** ®[7,8], **Evgenii Zhuravlev** ®[7], **Rustam H. Ziganshin** ®[3], **Ivan O. Butenko** ®[2], **Olga N. Bukato**[2], **Ksenia M. Klimina** ®[1,2], **Vladimir A. Veselovsky** ®[2],

Tatiana V. Grigorieva [9], Sergey Y. Malanin[9], Olga I. Aleshikova[10,11], Andrey V. Slonov [2], Nataliya A. Babaeva [10,11], Lev A. Ashrafyan[10,11], Elena Khomyakova[12], Evgeniy G. Evtushenko [13], Maria M. Lukina[1,2], Zixiang Wang [14], Artemiy S. Silantiev[2], Anna A. Nushtaeva [7], Daria D. Kharlampieva [2], Vassili N. Lazarev[1,2], Arseniy I. Lashkin[2], Lorine K. Arzumanyan[2], Irina Yu. Petrushanko [15], Alexander A. Makarov[15], Olga S. Lebedeva[1,2], Alexandra N. Bogomazova[1,2], Maria A. Lagarkova [2] & Vadim M. Govorun [16]

[1]Center for Precision Genome Editing and Genetic Technologies for Biomedicine, Lopukhin Federal Research and Clinical Center of Physical-Chemical Medicine of Federal Medical Biological Agency, Moscow 119435, Russian Federation. [2]Lopukhin Federal Research and Clinical Center of Physical-Chemical Medicine of the Federal Medical and Biological Agency, Moscow 119435, Russian Federation. [3]Shemyakin-Ovchinnikov Institute of Bioorganic Chemistry of the Russian Academy of Sciences, Moscow 117997, Russian Federation. [4]Faculty of Biology; Lomonosov Moscow State University, Moscow 119991, Russian Federation. [5]Moscow Institute of Physics and Technology (State University), Dolgoprudny 141701, Russian Federation. [6]Faculty of Chemical-Pharmaceutical Technologies and Biomedical Drugs, Mendeleev University of Chemical Technology of Russia, Moscow 125047, Russian Federation. [7]Institute of Chemical Biology and Fundamental Medicine, Siberian Branch, Russian Academy of Sciences, Novosibirsk 630090, Russian Federation. [8]Department of Natural Sciences, Novosibirsk State University, Novosibirsk 630090, Russia. [9]Kazan Federal University, Kazan 420008, Russian Federation. [10]National Medical Scientific Centre of Obstetrics, Gynaecology and Perinatal Medicine named after V.I. Kulakov, Moscow 117198, Russian Federation. [11]Russian Research Center of Roentgenology and Radiology, Moscow 117997, Russian Federation. [12]Exosome Analytics, Evry 91058, France. [13]Faculty of Chemistry; Lomonosov Moscow State University, Moscow 119991, Russian Federation. [14]Department of Obstetrics and Gynecology, Qilu Hospital, Cheeloo College of Medicine, Shandong University; Jinan, 250012 Shandong, China. [15]Engelhardt Institute of Molecular Biology, Russian Academy of Sciences, Moscow 119991, Russian Federation. [16]Research Institute for Systems Biology and Medicine, Moscow 117246, Russian Federation. ✉e-mail: victoria.shender@gmail.com

