## [Peer Review File · Nature Communications]

REVIEWER COMMENTS

Reviewer #1 (Remarks to the Author): Expert in RNA splicing, non-coding RNAs, and ovarian cancer

In this manuscript, the authors compare ascites before and after therapy and provide evidence that chemotherapy induces changes in the secretion which stimulates chemoresistance and tumor growth of ovarian cancer. The authors further reveal that extracellular vesicles secreted by apoptotic cells deliver spliceosomal proteins to recipient cells, hence promoting aggressive behavior. The manuscript is well written and the findings are novel and interesting. Overall, the data support the conclusions. However, the following concerns need to be addressed.

1. The authors claim that cisplatin treatment provokes active re-localization of spliceosomal proteins (Fig. 5C). I am not convinced it is an active or specific process. On Fig. 5C, SRSF4 and TIA1 are mostly observed in apoptotic vesicles. Apoptosis is characterized by the destruction of essential cell organelles, including the cell nucleus. Both nuclear pore complex (NPC) and the nuclear lamina are not intact which lead to nuclear proteins release from nucleus to the cytosol passively. Immunofluorescence analysis of nuclear envelope markers might be helpful to make is clear.
2. The authors chose SYNCRIP and SNU13 as representative secretory spliceosomal proteins for study. Additional evidence would be required to address their localization such as immunofluorescence or fluorescent fusion protein experiments secretory vesicles.
3. The expression level of SYNCRIP and SNU13 in ovarian cancer should be analyzed using TCGA or other data sets.
4. Did the authors investigated the functional roles of SRSF4 in recipient ovarian cancer cells?
5. The statistical result is missing in Fig.2E, Fig.4D, 4E, 4F etc.
6. In Fig.3B, what is the meaning of the vertical coordinates "heigh(t)" in hierarchical clustering dendrogram?
7. In Fig.5 legend, line 9, the concentration of cisplatin is "40 M", μ is missed.
8. Analysis on Fig.6B is poor, GraphPad Prism is more powerful.

Reviewer #2 (Remarks to the Author): Expert in ovarian cancer, ascites and proteomics

Manuscript

“Therapy-induced Secretion of Spliceosomal Components Mediates Pro-survival Crosstalk Between Ovarian Cancer Cells”

Corresponding author: Victoria O. Shender

PLEASE SEE COMMENTS AND OBSERVATIONS IN ATTACHED WORD FILE

Reviewer #3 (Remarks to the Author): Expert in ovarian cancer genomics and proteomics, tumour microenvironment

By performing integrative omics analyses on ovarian cancer patient-derived ascites, and in vitro cell line model Shender et al., manuscript describes that cisplatin-induced secretome is enriched in spliceosomal components. Furthermore, they show that stable expression of spliceosomal proteins SNU13, and SYNCRIP in SKOV3 cell line leads to decreased apoptosis in response to cisplatin treatment. In addition, exogenous U12 and U6atac snRNA promote subcutaneous tumor growth in immunodeficient mice. Based on RNAseq data they also report increased mesenchymal phenotype in tumor cells after cisplatin treatment compared to pre-treatment or control samples.

This manuscript aimed to provide detailed mechanistic insights into secretome-mediated resistance to cisplatin. However, the findings are mainly from ascites with unclear cell types and in vitro cell lines which represent only a subset of tumor cell populations that are more aggressive or have adapted to in vitro growth conditions and thus may not fully represent biology of in situ tumor. Furthermore, as indicated in the comments below, there are major concerns about the experimental design used in the study which confound the results and conclusions.

Major comments:

1. From the collected human ascites samples, how did the authors confirm that the cells are tumor cells? It is well known that especially after chemotherapy ascites is full of mesenchymal and immune cells. This is a major confounder in the following flow cytometry, functional and RNAseq analyses. This bias would already explain the observed enrichment in mesenchymal signatures from bulk RNAseq, with most of the signal coming from the mesenchymal/immune cells. Even if EMT signature is found in cancer cells, this would not be novel. This defect to clarify the cell populations in the ascites samples hampers all the experiments and conclusions of the study.
2. Most of the patient samples are high-grade (serous?) adenocarcinomas. However the cell model SKOV3 is not representative of HGSC. The authors should have used another, more representative cell line in the analyses.
3. The authors use skin fibroblasts as “normal cells” to show that secretome associated with chemoresistance comes from the cancer cell population. In line with the first comment, fibroblast are not the correct cell type. The authors should decompose what cells are there in the ascites to support their hypothesis, and use those cells with respect to the cancer cells.
4. In Fig. 3D, the legend says qPCR analysis of spliceosomal snRNAs in paired ascitic fluid before and after therapy. However, the data is shown only for after therapy samples. Similarly, in Fig. 4E, it is not clear if the data shown is from the control or therapy-induced fibroblasts.
5. The authors should specify the data in Fig 3E is from ascites pooled from different patients?
6. Although DNA damage is one of the main consequences of cisplatin treatment, authors do not show correlation between DNA damage and the secretion of spliceosomal proteins hence the statement on page 12 "Ovarian cancer cells secrete spliceosomal component upon DNA damage in vitro" is misleading, authors could consider rephrasing this sentence, or show evidence for association of the spliceosomal

proteins and DNA damage.

7. On page 15, authors state that “The functional annotation of heavy-labeled proteins whose abundance was elevated in the samples incubated with apoptotic vesicles revealed the highest enrichment in the cluster of spliceosome-related proteins”. However, data to support that the vesicles are indeed apoptotic is lacking. The authors should provide some evidence to show that vesicles derived from apoptotic cells mediate increase in intracellular spliceosomal proteins.

8. Extended Fig. 5B, authors report that comparison of RNA expression profiles in three different models before and after therapy show significantly more mesenchymal signature post drug treatment. However, this observation is seen in 4/8 patients and 2/4 ascites indicating inter-patient heterogeneity. Also, patient 1, 7, and 8 lack either before or after therapy data can authors clarify this?

Reviewer #4 (Remarks to the Author): Expert in ovarian cancer genomics

This is an interesting study showing that treating ovarian cancer patients with chemotherapy (cisplatin) increases cellular resistance by promoting the transfer of spliceosomal proteins from dying cancer cells to live cells by exosomal transport. The authors have included extensive analyses at the transcriptomic, proteomic and functional levels to confirm this novel finding. Several details require additional clarifications.

Comments:

1. Please specify number of replicates and include p-values for statistical significance in several figures, particularly Figure 1 panels C, G, H, Extended figure 1 panels B-D; Figure 2 panels C-D; Extended figure 5B; Figure 6A
2. In Figure 1E, the EMT markers appear to change in the opposite direction than noted in the text. Please clarify. Additionally, the color scheme for "before" and "after" switches in Figure 1I. Please reconcile. Please also clarify in which patient the experiment in Figure 1I was done, and why it was only performed in one sample.
3. in Figure 1H, please clarify what is quantified in the bar graph, since this does not appear to reflect what is shown in the images.
4. Extended figure 3D is poor quality. Please replace.
5. Figure 3F the y-axis appears incorrectly proportioned.
6. Lines 408-409 and Extended figures 5B and 6A-D need additional explanation, particularly as to which method was used and why the negative EMT score indicates increased mesenchymal phenotype.
7. Extended figures 7A-B need more explanation regarding the meaning of the numbers in the table.
8. Line 480 and extended figure 9E - please explain the relevance of the changes in cell cycle.

Reviewer #5 (Remarks to the Author): Expert in cancer proteomics

Summary :

In this manuscript, the author investigated the contribution of therapy-induced intercellular communication to the development of resistance to chemotherapy in ovarian cancer. According to their findings, they assume the acquisition of chemoresistance is associated with extracellular communication, which is mediated by secreted spliceosomal components. This work is interesting, yet the conclusion was proposed without solid evidence. Several of the interpretations are also insufficiently supported by the data and the paper is often carelessly written. I have the following specific comments.

Major Points:

1. The authors collected the paired ascitic fluids from the same ovarian cancer patients before and after chemotherapy, to investigate their impacts on ovarian cancer cells. However, the authors did not describe the criteria under which patients were selected, neither did they describe patients' responses to chemotherapies, did the patients evaluate using RECIST criteria, or other pathological criteria?
2. The impact and clarity of this paper is compromised by the fact that only one type of cell line was used in this paper. Since the patients contained serous, small-cell carcinoma of ovary hypercalcemic type and also mucinous ovarian adenocarcinoma, can SKOV3 cell line represents all three types of ovarian cancer, if not, the authors should include other suitable cell lines in their studies.
3. Also, since the serious ovarian cancers are thought to arise from fallopian tube and mucinous from bowel, it is confusing why the author chose the fibroblast cells to represent the normal control cells for ovarian cancer?
4. In addition, multiple previous literatures have indicated the involvements of EMT, WNT signaling pathways in the chemo-resistance of ovarian cancer (Oncogene, PMID: 22797058, Cancer Research, PMID: 33574097, Cancer Research, PMID: 25377471), thus it is not surprising that the authors found tumor cells exhibited a higher activity of the genes associated with WNT signaling pathways, the authors should present their novel findings.
5. In line 189-193, the authors indicated that inhibition of apoptosis as well as vesicular transport in donor cells diminished the protective effect of secretome. However, the authors ignore to describe how do the inhibitors been selected?
6. In line 198-201, the authors showed that pre-incubation of normal fibroblasts with tumor cell secretome reduced the resistance of fibroblasts to cisplatin, then concluded that the CM from cancer cells mainly affected the neighboring tumor cells but had no effect on normal cells. This conclusion is confusing, the author should explain it clearly. In addition, the author should explain why therapy-induced secretome activated the immune response of recipient fibroblast. How do these results connect to the whole story?
7. In figure 3A, why does the cirrhosis ascites in particularly been collected for secretome analysis, is

there any association between cirrhosis and ovarian cancer?

8. The author indicated that one of their major findings is detecting spliceosomes in the extracellular space. However, other groups have already reported that apoptotic cancer cells can secrete apoptotic extracellular vesicles enriched with various spliceosomes, to promote survival cell proliferation and therapy resistance (Cancer Cell, PMID: 29937354). It is immediately clear how this study provides novel insights.

9. The authors utilized the spectral count values to quantify the proteins identified from the ascitic fluids. However, in the majority of proteomic studies, the intensity-based absolute quantification (iBAQ) algorithm or label-free quantification algorithm (LFQ) were conducted for label-free protein quantification (Nature Communications, PMID: 30258067, Cell Metabolism, PMID: 29320704). I would recommend the authors to use iBAQ or LFQ values instead of spectral count values for protein quantification, to ensure the utility of these data by public.

10. One of my major concerns about this paper, is that it largely depends on correlation analysis and GO enrichment analysis. However, GO enrichment analysis itself can not provide solid evidence, thus, the author should be more focused on specific molecules and signaling pathways, and present more direct evidence, for their hypothesis.

11. The authors identified 63 common spliceosomal proteins that may contribute to intercellular communication. It is an interesting finding, yet need to be verified. Given that so many proteomics studies have been published over the past few years, the author should compare and contrast the data collected here with public database.

12. The authors utilized overexpression systems to verify the effects of SNU13 and SYNCRIP. However, the author should also include knockdown experiments too.

13. The authors indicated that U12 snRNA, U6atac snRNA caused tumor cell proliferation. It is an important finding, yet, the authors should conduct some functional experiments to elucidate how these snRNAs impact the cell proliferation process, and finally lead to aggressive prognosis.

Minor points:

1. In figure 1E, the colors that presented cytometry results before and after therapy are marked incorrectly.
2. In line 125-127, the authors indicated the activation of EMT as well as the MYC signaling pathway, but do not describe in which group do these processes and pathways been activated?
3. The extended figure 5H does not exist.
4. The figure 4F does not contain IL6.
5. The writing of the paper should be improved. The authors do not clearly describe how specific analyses were performed. Specific methods should be cited in the main results section.
6. The figure legends should be written with more details.

Reviewer #6 (Remarks to the Author): Expert in extracellular vesicles

In this paper, the authors show that spliceosomal components encapsulated in EV emanating from cisplatin treated cancer cells favors cell survival of recipient cells treated with aforementioned donor EVs.

Experimentally, the authors used multi-omics, in vitro and in vivo biology to support their conclusion. Experimental approach is adequate and well described. Discussion is well balanced and data do not seem to be overinterpreted.

Data would be of interest for cell biology and cancer biology, and reinforce the pro-tumoral role of EVs. However, I believe that the data could be even stronger if the proof that the signal carried by EVs is responsible for the observed phenotypes. A significant improvement would be to isolate EVs through sucrose flotation, directly demonstrate that spliceosome candidates are within the EV using well established protease protection assay et clearly assess the quality of the EV prep. Such isolated EVs are expected to trigger similar phenotype, and may even enhance it if the preparation is purer. More direct demonstration would better support direct role of EVs.

Manuscript

“Therapy-induced Secretion of Splicesomal Components Mediates Pro-survival Crosstalk Between Ovarian Cancer Cells”

Corresponding author: Victoria O. Shender

The authors describe the problem of resistance in ovarian cancer as an important limiting factor in the success of treatment, focusing on a different way of acquiring resistance through the extracellular pathway instead of the classic pathway through mutations in the genome.

They highlight the impact of malignant ascites generated by ovarian cancer and its participation in cross-communication for the development of a more aggressive disease. Particularly, they demonstrate the participation of extracellular vesicles in the transport of relevant molecules for the acquisition of resistance to cisplatin treatment. In the same vein, they present their findings on the transfer by extracellular vesicle secretion of the spliceosome elements U12 and U6atac snRNAs as "representative" end-elements of the components involved in mediating mechanisms of malignant cells survival.

The results presented in this manuscript are of relevance to the field of cancer in general and in particular for ovarian cancer because the participation of RNA splicing in cancer development is understudied.

In general terms, the results presented in the manuscript support the conclusions.

However, some major and minor issues will be pointed out.

MAJOR ISSUES

The research work is comprehensive and includes a sound methodological strategy, the findings demonstrate that there is a transfer of an important mechanism for the acquisition of resistance to treatment via the extracellular pathway, through the transfer of spliceosome elements; however, the following points are not discussed:

1). What are the specific target(s) of this "spliceosome" machinery, which would be directly responsible in the mechanism of an increased resistance to carboplatin? The authors should perform a bioinformatic analysis of their sequencing data of their SKOV-3 model incubated with conditioned medium (CM) to evaluate which were the modified genes and proteins affected by the arrival and utilization of these splicing elements which function as key elements for cell survival.

2). Knowing which elements are transferred by cells in apoptosis or "donor culture" the authors should perform a bioinformatics analysis of their sequencing data of representative spliceosome elements such as U12 and U6atac snRNAs, and proteins pre- and post-treatment with chemotherapeutic agents to determine what the differences are between these elements.

3). If "donor" cells possess elements that help "acceptor" cells resist cisplatin, how do the authors explain that these elements are not used by the donor cells themselves to survive treatment?

METHODOLOGY:

Regarding the ascitic fluids used, authors provide information about 23 paired samples, before and after the administration of a routine course of neoadjuvant chemotherapy. Following the Methods section, it is evident that the ascitic fluids were used for:

1. Establishment of primary cell cultures
2. RNAseq analysis
3. MTT viability assays
4. Wound healing assays
5. Flow cytometry analysis to study the effect of ascitic fluids on cancer cells
6. Proteomic analysis
7. Exosomes counting using nanoparticles tracking analysis (NTA)
8. Immunoblotting
9. RNA isolation followed by qRT-PCR of selected molecules

In addition to the use of cells collected from the different samples of ascitic fluids, authors also used commercial cell lines such as SKOV-3 cells and OVCAR3 both from ATCC. In this section authors provide a detailed description of the conditions in which the cells were maintained and cultured. Also they mention that as a control cell line they used human dermal fibroblasts with the ethical approval of the Research and Clinical Center of Physical-Chemical Medicine. They also included in this study, human cells from embryo kidney for Lentivirus production and transduction.

At this point, it would be of help to include catalog number for all the cell lines used in the study.

Regarding the secretome generation, SKOV3, OVCAR3 and Fibroblast were exposed to different concentrations of Cisplatin (**the catalog number is missing**), to produce approximately 50% of dead cells; they mention that the reported method provides a high level of apoptosis in tumor cells (**not shown**) and also facilitates the recovery of a drug-free medium (**which was not tested**).

In general, in any of the reagents used the catalog number is provided, only the company name from which they were obtained, except for the antibodies used for flow cytometry assays.

For biological samples processing, cells present in ascitic fluids were recovered after centrifugation; these cells were used for survival assays (MTT and Caspase 3/7 activity assay), and flow cytometry assays to evaluate the abundance of surface markers present in cells recovered before and after chemotherapy and from tumor cell lines before incubation with ascitic fluids and also before and after treatment with chemotherapy

For wound healing assasy by scratch, authors refer to a previous publication for details (*Nature* **520**, 368–372 (2015)). However, **there is no detailed description of the "growth medium"**.

In supplementary Table 1, there is the description of the use of the different ascitic samples in different methodological strategies, i.e., authors did choose ten pairs of ascitic fluids from ovarian

cancer patients before and after chemotherapy to be processed by the Combinatorial peptide ligand library in order to enrich low concentration proteins to improve their identification by LC-MS/MS, instead of eliminating high concentration proteins. **What was the reason to select 10/23 pairs of ascitic fluids and which were the criteria followed to select those 10 that were processed? Please provide an explanation how each sample was selected for particular experiments.**

SECRETOME PREPARATION FOR PROTEOMIC ANALYSIS

Page 36, Line 886:

Heavy labeled donor cells were treated (control vesicles) or untreated (apoptotic vesicles) with cisplatin according to the previously described scheme

Page 36, line 886: There seems thatlabeled donor cells were treated (control vesicles) or untreated (apoptotic vesicles) should be written the other way around:treated (apoptotic vesicles) or untreated (control vesicles)...

Could it be possible to include in one chapter all types of processing and all types of samples for LC-MS/MS Analysis?

Ascitic fluids

Secretomes

Cell fractions: cytoplasm and nuclei

In-gel samples

Subtitle in line 938 could go at the beginning of the whole section.

The way all these methodologies are presented in the manuscript made me go back and forth to re-read and understand the differences.

Please explain the differences for the processing for proteomic analysis in a) line 939 and b) line 955

Preparative SDS-PAGE and In-gel trypsin digestion

Which samples were treated in this way?

PRESENTATION OF DATA:

The information and data presented in the manuscript is considerable, given that many conditions and types of samples were analyzed by LC-MS/MS generating a huge amount of information.

All results from transcriptomic, proteomic and sequencing data, as well as bioinformatics analyses are properly presented in supplementary tables. From those, selected information was used to prepare the corresponding figures and extended figures.

From my point of view, In the way the manuscript is organized, results presented in Figures 1 and 2 are not strictly relevant to the main results of the manuscript. Figure 1 shows that ovarian cancer cells recovered from patients after therapy display higher chemoresistance and activation of the EMT pathway, whereas Figure 2 depicts that the cells secretome induced after therapy is able to promote cell chemoresistance *in vitro*. These two figures together with extended figures 1 and 2, point out that chemotherapy induce a more aggressive behavior of tumor cells both *in vivo* and *in*

vitro, which is not really a new finding. Please see lines 88 to 91. Moreover, results from extended figures 5 and 6 further confirm the induction of the EMT process by incubation with ascitic fluids recovered after chemotherapy. The transfer of vesicles with the "spliceosome" component, contained in the conditioned medium occurs in the context of ascites-induced malignant mesenchymal cells and that this process occurs in transition cells or in intermediate phenotype. Probably a literature review of this phenomenon in the context of malignant cells under TEM can be considered to contribute to the characterization of the generation of an aggressive phenotype by ascites. Interestingly, this part of the work remains unconnected at the end; authors did not take into consideration the epithelial, mesenchymal or intermediate phenotype of the cells when collecting the conditioned medium and did not discuss this further. **Could this information be obtained and incorporated into the manuscript?**

Then in figure 3, authors begin to work with paired ascitic fluids obtained before and after patients were treated with chemotherapy; with these samples they found that malignant ascitic fluids (after chemotherapy) were enriched with spliceosomal components. To me this very well could be the first figure in the manuscript.

Additional questions:

It is unclear whether the transfer of spliceosome components occurs with the 23 pairs of ascites (pre and post).

It is also not clear whether 23 primary cell cultures were established

How frequent is the formation of ascites following neoadjuvant chemotherapy? Please comment on this issue in the text.

Regarding the study sample, consisting of 23 pairs of ascites fluids from ovarian cancer patients, before and after treatment with cisplatin, what was the volume obtained in all cases before and after chemotherapy?

How was the procedure to obtain the ascites fluids? it is not clear, if during the paracentesis performed before the chemotherapy the whole volume of ascites was removed, in order to subsequently give the neoadjuvant chemotherapy?

How was it detected if the patients regenerated ascites, considering that post-treatment ascites were recovered after only 21 days?

"Minor Issues"

Some inconsistencies:

Neuroendocrine tumor of the ovary. This name appears only three times along the manuscript. There is no description of it in Supplementary Table 1, sheet 1.

In ST3 there is information about only four samples: 3, 9, 6, and 11. However in the text they mention 5 primary ovarian cancer cell cultures, but ahead in the text they mention three were

Epithelial ovarian cancer (EOC) and one a more aggressive neuroendocrine tumor of the ovary = 4 samples.

Figure 2: During this experiment, SKOV3 donor cells were incubated with a cisplatin-containing medium for 7 h, then the cells were washed and incubated with a cisplatin-free medium for 17 or 41 h (Fig. 2A). Lanes 181-183, with the Fig. 2 scheme showing that SKOV-3 cells were incubated with cisplatin for 24 or 48 h, and then incubated for 7 h

Lines 278 to 281: We analyzed the proteomes of conditioned media from ovarian adenocarcinoma SKOV3 cells and primary cultures of normal skin fibroblasts before and 24 h after cisplatin treatment (Fig. 4A). To reduce the likelihood of serum contamination, after 7 h of incubation with cisplatin, the cells were washed with PBS and the medium was replaced with the serum-free one. It is not clear whether cisplatin was added again when the medium w/serum was changed by serum-free medium.

Figure 4F. Again, the text does not fit with the results shown in Fig 4F.

In the manuscript, lines 323-325: In contrast, in the fibroblast secretomes, most of these proteins including TGF β 1 and IL6 were detected exclusively after chemotherapy (Fig. 4F). In the figure the proteins that were increased in fibroblasts secretomes after cisplatin treatment were GDF15 (26), CSF1 (54), and FSTL1 (66); Il-6 does not appear in these protein lists.

The comparison of these data with the biological effects of secretomes suggests that despite the reduced content of soluble growth factors, tumor therapy-induced secretomes have a higher effect on recipient cells than the secretomes obtained before therapy (Fig. 2B). This effect may be explained by the presence of spliceosomal proteins.

How do authors reach this conclusion??? Authors did know in advance to this work, that cells under chemotherapy responded with changes in splicing that were among the most conservative mechanisms of therapy response in tumor cells.

Line 376:proteins not involved in RNA splicing (Extended Fig. 5H). Should be (Extended Fig. 4H).

Line 383- 385: We confirmed the SILAC data by detecting SRSF4-RFP and TIA1-GFP secreted by drug stressed cancer cells in the recipient cells using fluorescent microscopy and by Western Blot (Fig. 4G,H). Should be (Fig. 5G, H)

Line 393: Comparing results from figure 5I with results in figure 2F, how come that they agree with each other? Figure 2F does not show any spliceosome-related results. It shows four hallmarks: E2F targets, EMT, G2M checkpoint, and MYC targets

Lines 472-474:

In addition, these snRNAs cause enhanced expression of genes that promote cell migration — *MATN2*, *ITGB1*, and *FGFR1OP*. Where are these data shown?

Lines 475-476:

Next, we performed proteomic profiling of the corresponding samples. A total of 4,813 proteins were identified. The data sheet regarding the proteome (supplementary Table 8, sheet 2) shows a list of 3962 proteins

Lines 119 and 161:

it is not clear whether they are cell lines or primary cultures, because it says "Our results showed that all tumor cell lines obtained after chemotherapy were considered more resistant".

Line 180:

the authors mention the treatment of SKOV-3 acceptor cells with the conditioned medium, but do not mention **the time and conditions of treatment**. It is also not found in the methods section. This should be included between the sections on "cell cultures" (line 676) and "secretome generation" (line 688).

The authors should clarify how the inclusion of a cirrhosis ascites control is not biasing the study by ruling out proteins inherent to a pathologic process that may be shared with malignant disease but that equally contribute to disease development and progression?

What is the 76GS method? Please provide a brief description of it. In the cited reference this does not appear in the methods section, instead is part of the results.

DETAILS IN FIGURES:

Extended figure 2A:

Western blot results showing the presence of CD63 are really poor quality.

Extended figure 5B:

The third graph on the right: It could seem irrelevant for the results, but in the left to the right reading of the results, I should be better to graph the "before" conditions on the left and the "after" conditions on the right of the graph.

R1, R2, and R3: which samples do they come from? Are they related to patient 1, patient 2 and patient 3?

Point-by-point Response to Reviewer's Comments

Reviewer #1 (Remarks to the Author): Expert in RNA splicing, non-coding RNAs, and ovarian cancer

In this manuscript, the authors compare ascites before and after therapy and provide evidence that chemotherapy induces changes in the secretion which stimulates chemoresistance and tumor growth of ovarian cancer. The authors further reveal that extracellular vesicles secreted by apoptotic cells deliver spliceosomal proteins to recipient cells, hence promoting aggressive behavior. The manuscript is well written and the findings are novel and interesting. Overall, the data support the conclusions. However, the following concerns need to be addressed.

We highly appreciate the Reviewer for taking time to carefully review the manuscript and give thoughtful and constructive comments, which has greatly helped to improve this paper. Below is our point-by-point response to each comment.

1. The authors claim that cisplatin treatment provokes active re-localization of spliceosomal proteins (Fig. 5C). I am not convinced it is an active or specific process. On Fig. 5C, SRSF4 and TIAI are mostly observed in apoptotic vesicles. Apoptosis is characterized by the destruction of essential cell organelles, including the cell nucleus. Both nuclear pore complex (NPC) and the nuclear lamina are not intact which lead to nuclear proteins release from nucleus to the cytosol passively. Immunofluorescence analysis of nuclear envelope markers might be helpful to make is clear.

Response: We performed immunocytochemistry analysis of SKOV3 cells treated or untreated with cisplatin using antibodies against spliceosomal protein SRSF1 and nuclear protein RPA2. We showed that only SRSF1 was detected in cytoplasm after cisplatin treatment while RPA2 maintained its presence in the nuclei (Figure 1). This observation suggests that cisplatin may induce the translocation of spliceosomal proteins from the nucleus to the cytoplasm, occurring prior to nuclear fragmentation. Additionally, we have revised the text, replacing the phrase "stress provokes active re-localization of spliceosomal proteins" with "stress provokes re-localization of spliceosomal proteins".

Figure 1 - Fluorescence images of SKOV3 cells stained for SRSF1 (Alexa 555, red), phRPA2 (Ser33, green) and with DAPI (blue) before and after treatment with 40 μ M cisplatin for 7 h. Scale bar is 100 μ m.

Revised manuscript (Page 10, Lines 330-337, Extended Figure 5B).

2. The authors chose SYNCRIP and SNU13 as representative secretory spliceosomal proteins for study. Additional evidence would be required to address their localization such as immunofluorescence or fluorescent fusion protein experiments secretory vesicles.

Response: We verified that SYNCRIP and SNU13 can be secreted from dying cancer cells and transferred into recipient cells using GFP- and RFP- fusion constructs and subsequent confocal microscopy and flow cytometry analysis (Figure 2A). Furthermore, we also isolated CD9 positive extracellular vesicles from secretomes using Exosome-Human CD9 Flow Detection Reagent and confirmed the presence of SRSF4-RFP, SYNCRIP-GFP, and SNU13-GFP in CD9 positive extracellular vesicles secreted by dying cancer cells (Figure 2B).

Figure 2 - (A) Confocal immunofluorescence images depict recipient SKOV3 cells incubated for 3 days with therapy-induced secretomes (TIS) or control secretomes (CtrlS) from donor SKOV3 cells overexpressing GFP (green), SYNCRIP-GFP (green), or SNU13-GFP (green). Actin filaments are stained with Phalloidin (Alexa 555, red), while nuclei are stained with DAPI (blue). Scale bar: 25 μ m. (B) Flow cytometry analysis of magnetic beads coated with anti-CD9 antibody after immunoprecipitation of extracellular vesicles from therapy-induced secretomes of donor SKOV3 cells. Donor SKOV3 cells overexpressing SRSF4-RFP, SYNCRIP-GFP or SNU13-GFP were treated with 40

μM cisplatin for 7 h and then medium were replaced with fresh medium not containing cisplatin and incubated additional 41 h. High-fluorescent events are highlighted in blue.

Revised manuscript (Page 11, Lines 357-363, Extended Fig. 5D-E).

3. The expression level of *SYNCRIP* and *SNU13* in ovarian cancer should be analyzed using TCGA or other data sets.

Response: We appreciate your observation, and we are actively conducting independent research on the roles of *SYNCRIP* and *SNU13* in ovarian cancer progression. With your permission, we propose to incorporate this valuable information into our other manuscripts.

We assessed the expression of *SYNCRIP* and *SNU13* in ovarian cancer and normal tissues, leveraging data from the TCGA and GTEX databases, respectively. Notably, the expression of *SYNCRIP*, but not *SNU13*, was elevated in tumor tissues compared to normal tissues (Figure 3A). Additionally, we evaluated the prognostic significance of *SYNCRIP* and *SNU13* in ovarian cancer. Utilizing proteome data from the TCGA database, we observed a significant correlation between high levels of *SNU13* and poor overall survival and progression-free survival in patients with ovarian cancer (Figure 3B).

Figure 3 - (A) *SYNCRIP* and *SNU13* mRNA expression was analyzed in the TCGA-OV cohort (n = 105 - samples from TCGA with a tumor cell content of more than 90% were selected) and normal tissue in GTEx datasets (n = 88). **(B)** Kaplan-Meier survival curves illustrate the influence of *SNU13* expression on survival and progression-free survival of ovarian cancer patients. The p-value was obtained by log-rank test. (not included into the final manuscript).

4. Did the authors investigate the functional roles of SRSF4 in recipient ovarian cancer cells?

Response: Thank you for the question. We have intentions to further investigate the role of SRSF4, especially given that our SILAC and western blotting data indicate a significant increase in its level in recipient cells upon exposure to therapy-induced secretome. However, the establishment of cell lines with stable overexpression or knockdown of SRSF4 is intricate due to its autoregulatory feedback loop [PMID: 22436691]. Notably, recent findings have demonstrated that SRSF4 plays a role in promoting temozolomide resistance in glioblastoma cells by modulating double-strand break repair [PMID: 37014544].

5. The statistical result is missing in Fig.2E, Fig.4D, 4E, 4F etc.

Response: Thank you for pointing this out. We added statistical information to the corresponding figures.

6. In Fig.3B, what is the meaning of the vertical coordinates "height(t)" in hierarchical clustering dendrogram?

Response: The vertical coordinate "height" in the hierarchical clustering dendrogram serves as a representation of the dissimilarity or distance between clusters. Height values signify the level of dissimilarity at which clusters are merged or joined. Lower height values indicate a higher degree of similarity or proximity between clusters, whereas higher height values signify greater dissimilarity or distance. The utilization of height values aids in the visualization of the hierarchical structure of the clustering process, providing insights into the relationships and similarities among the clusters.

7. In Fig.5 legend, line 9, the concentration of cisplatin is "40 M", uM is missed.

Response: Thank you for pointing this out. We corrected it.

8. Analysis on Fig.6B is poor, GraphPad Prism is more powerful.

Response: Thank you for pointing this out. We corrected it (Figure 6D).

Reviewer #2 (Remarks to the Author): Expert in ovarian cancer, ascites and proteomics
The authors describe the problem of resistance in ovarian cancer as an important limiting factor in the success of treatment, focusing on a different way of acquiring resistance through the extracellular pathway instead of the classic pathway through mutations in the genome. They highlight the impact of malignant ascites generated by ovarian cancer and its participation in cross-communication for the development of a more aggressive disease. Particularly, they demonstrate the participation of extracellular vesicles in the transport of relevant molecules for the acquisition of resistance to cisplatin treatment. In the same vein, they present their findings on the transfer by extracellular vesicle secretion of the spliceosome elements U12 and U6atac snRNAs as "representative" end-elements of the components involved in mediating mechanisms of malignant cells survival. The results presented in this manuscript are of relevance to the field of cancer in general and in particular for ovarian cancer because the participation of RNA splicing in cancer

development is understudied. In general terms, the results presented in the manuscript support the conclusions. However, some major and minor issues will be pointed out.

We thank the Reviewer for taking the time to review our manuscript and the helpful comments which helped us in improving the quality of the manuscript. Please find below a detailed point-by-point response to all comments:

MAJOR ISSUES

The research work is comprehensive and includes a sound methodological strategy, the findings demonstrate that there is a transfer of an important mechanism for the acquisition of resistance to treatment via the extracellular pathway, through the transfer of spliceosome elements; however, the following points are not discussed:

9. What are the specific target(s) of this "spliceosome" machinery, which would be directly responsible in the mechanism of an increased resistance to carboplatin? The authors should perform a bioinformatic analysis of their sequencing data of their SKOV-3 model incubated with conditioned medium (CM) to evaluate which were the modified genes and proteins affected by the arrival and utilization of these splicing elements which function as key elements for cell survival.

Response: We thank the Reviewer for this question. In response, we conducted multiple additional experiments to elucidate the molecular mechanism underpinning the effect of therapy-induced secretomes (TIS). Our comprehensive RNAseq and proteomic analyses unveiled that TIS led to an increased abundance of proteins associated with DNA repair and cell cycle regulation in recipient cells (Figure 4A). These findings were further corroborated through western blotting and cell cycle analysis (Figure 4D, J). Intriguingly, these data align closely with the proteome profiles observed in patient-derived cancer cells incubated with ascites after therapy. To investigate whether in vitro TIS could replicate the observed differences between platinum-sensitive and -resistant isogenic cell lines, we analyzed previously published RNAseq datasets (GSE148003, GSE98559, GSE98230, GSE173201) encompassing several isogenic platinum-sensitive and -resistant ovarian cancer cell lines. Principal component analysis demonstrated that SKOV3 cells incubated with TIS clustered with platinum-resistant cancer cell lines. In contrast, cells exposed to control secretomes exhibited gene expression patterns akin to platinum-sensitive cell lines (Figure 4B-C). Recent research has shown that the knockout or knockdown of at least one splicing factor, including those identified in TIS, significantly impairs DNA repair and enhances the cytotoxic effect of various genotoxic drugs (Figure 4E). Based on these findings, we hypothesized that the increased abundance of spliceosomal proteins in cancer cells pre-incubated with TIS might facilitate a more effective response to DNA-damaging insults. To test this hypothesis, we treated SKOV3 cells pre-incubated with TIS or control secretomes with cisplatin and evaluated various hallmarks of DNA damage response. We conducted a comet assay (analysis of DNA strand breaks), quantified cisplatin-DNA adduct accumulation, and monitored the phosphorylation levels of H2AX (an early DNA damage response marker) and RPA2 (a replicative stress marker). Remarkably, SKOV3 cells pre-incubated with TIS exhibited an increase in the number of pH2AX foci, accompanied by significantly fewer DNA double strand breaks, reduced platinum adducts, and lower RPA2 phosphorylation levels compared to cells incubated with control secretomes (Figure 4F-I). Additionally, cell cycle analysis indicated that

cancer cells incubated with TIS remained longer in the S phase, which facilitates DNA repair (Figure 4J). Finally, we explored the impact of TIS on the sensitivity of cancer cells to both DNA-damaging (cisplatin, doxorubicin, and etoposide) and non-DNA damaging (taxane and staurosporine) anticancer drugs. The results underscored that TIS heightened the resistance of cancer cells exclusively to DNA-damaging drugs, with no discernible effect on other therapy types (Figure 4K). In conclusion, these results suggest that TIS may promote DNA repair in residual tumor cells, aiding their survival against subsequent therapeutic insults. For a more detailed exposition, please refer to the Results section, specifically the item titled “Therapy-induced secretomes of ovarian cancer cells activate pathways important for cell response to DNA damage”.

Figure 4 - Therapy-induced secretome of ovarian cancer cells activates pathways important for cell response to DNA damage (**A**) Gene Ontology enrichment analyses of upregulated genes and proteins in SKOV3 cells incubated for 3 days with therapy-induced secretomes compared to control secretomes. The color scale refers to $-\log_{10}$ (FDR) values; the number of proteins/genes are represented by the diameter of the circles. (**B**) The principal component analysis of RNAseq data obtained from platinum-sensitive and -resistant isogenic ovarian cancer cell lines (A2780, SKOV3, and OVCAR5) and recipient SKOV3 cells incubated for 3 days with therapy-induced (TIS) or control (CtrlS) secretomes. Pink – platinum-resistant ovarian cancer cell lines, light blue – platinum-sensitive ovarian cancer cell lines, red – recipient SKOV3 cells incubated with TIS, dark blue – recipient SKOV3 cells incubated with CtrlS. (**C**) GSEA analysis of gene expression in platinum-resistant ovarian cancer cell lines (A2780, SKOV3, and OVCAR5) versus platinum-sensitive ovarian cancer cell lines (A2780, SKOV3, and OVCAR5). The X-axis represents GSEA enrichment score (p-values are indicated by colors). (**D**) Western blotting analysis of SKOV3 cells that were incubated for 3 days with therapy-induced (TIS) or control secretomes (CtrlS) from donor SKOV3. (**E**) Results of the intersection between spliceosomal proteins identified in TIS from SKOV3 cells and/or in ovarian cancer ascitic fluids after therapy (our data) and the hits from siRNA and CRISPR screenings (derived from data reported in [PMID: 34320214; PMID: 22344029; PMID: 27462432]). ATRi and CHK1i – CRISPR screens with inhibitors targeting ATR and CHK1, respectively. Loss of spliceosomal proteins indicated as "hit" increased the sensitivity of cancer cells to ATR or CHK1 inhibition [PMID: 34320214]. RAD51 foci and HR – siRNA screenings indicating that knockdown of spliceosomal protein impaired the formation of IR-induced RAD51 foci or decreased homologous recombination (HR) potential in the DR-GFP assay in cancer cells, respectively [PMID: 22344029; PMID: 27462432]. (**F**) Box plots show the number of γ H2AX foci per nucleus in SKOV3 cells pre-incubated with TIS or CtrlS for 3 days and then treated with cisplatin (25 μ M) at different time points. The number of γ H2AX foci was calculated using ImageJ software with FindFoci plugins. 60-200 cells were analyzed in each sample. (**G**) Box plots of tail moments from neutral comet assays of SKOV3 cells pre-incubated with TIS or CtrlS for 3 days and then treated with cisplatin (10 μ M) for 48 h. Tail moment was defined as the product of the tail length and the fraction of total DNA in the tail (Tail moment = tail length x % of DNA in the tail) and was quantified using the OpenComet software. 60-100 cells were analyzed in each sample. Experiments were performed in triplicate. (**H**) Box plots show the number of cisplatin-DNA adducts' foci per nucleus of SKOV3 cells pre-incubated with TIS or CtrlS for 3 days and then treated with cisplatin (25 μ M) for 48 h. The number of cisplatin-DNA adducts' foci was calculated using ImageJ software. 60-200 cells were analyzed in each sample. (**I**) Box plots show the number of phosphorylated RPA2 (phospho S33) foci per nucleus in SKOV3 cells pre-incubated with TIS or CtrlS for 3 days and then treated with cisplatin (25 μ M) at different time points. The number of phosphorylated RPA2 foci was calculated using ImageJ software. 60-200 cells were analyzed in each sample. (**J**) Cell cycle analysis with flow cytometry of SKOV3 cells pre-incubated with TIS or CtrlS for 3 days and then treated with cisplatin (10

μM) for 24 h. Stacked bar graphs show the percentage of cells in different phases of the cell cycle. Percentage of cells in G1, S, and G2 phases was calculated with NovoExpress software. (K) Dose-response curves obtained by MTT assay of recipient SKOV3 cells incubated with therapy-induced (TIS) or control (CtrlS) secretomes for 3 days, and then treated with different concentrations of cisplatin, doxorubicin, etoposide, paclitaxel, or staurosporine for 2 days. The data represent the mean values \pm SD ($n=3$). Asterisks denote statistically significant differences determined using unpaired, two-tailed Student's t-test: $*0.05 < p\text{-value} > 0.005$; $**0.005 < p\text{-value} > 0.00005$; $***p\text{-value} < 0.00005$.

Revised manuscript (Pages 11-13, Lines 369-412, Figure 5 and Extended Fig. 6C).

10. Knowing which elements are transferred by cells in apoptosis or "donor culture" the authors should perform a bioinformatics analysis of their sequencing data of representative spliceosome elements such as U12 and U6atac snRNAs, and proteins pre- and post-treatment with chemotherapeutic agents to determine what the differences are between these elements. If "donor" cells possess elements that help "acceptor" cells resist cisplatin, how do the authors explain that these elements are not used by the donor cells themselves to survive treatment?

Response: Initially, we examined potential post-transcriptional modifications (ubiquitination, phosphorylation) of proteins secreted by dying cancer cells compared to those from untreated cells using LC-MS/MS data, but no differences were observed. Additionally, we assessed the expression levels of spliceosomal components in cells after cisplatin treatment and found no upregulation of these endogenous spliceosomal proteins (Figure 5). We speculate that the treatment's impact on dying cells is too intense, preventing the upregulation of endogenous spliceosomal proteins that could facilitate cell survival. Nonetheless, we propose that dying cells generate a pool of "exogenous" spliceosomal components, potentially augmenting their abundance in neighboring tumor cells and promoting their survival without necessitating the upregulation of endogenous proteins.

Figure 5 - Real-time RT-PCR analysis of *SNU13*, *SRSF4*, *SYNCRIP*, U12 snRNA and U6atac snRNA expression levels in SKOV3 cells treated with indicated cisplatin concentrations (0, 1, 5, 10, or 40 μM)

for 6, 12, 24, 48, or 72 hours. The data were normalized to *GAPDH* (*SNU13*, *SRSF4*, *SYNCRIP*) or 18S rRNA (U12 snRNA and U6atac snRNA) expression. The data are reported as means \pm SEM with significant differences between groups (untreated or treated with different concentrations of cisplatin) labeled with asterisks (*, $p < 0.01$), (not included into final manuscript).

METHODOLOGY:

Regarding the ascitic fluids used, authors provide information about 23 paired samples, before and after the administration of a routine course of neoadjuvant chemotherapy. Following the Methods section, it is evident that the ascitic fluids were used for:

- 1) *Establishment of primary cell cultures*
- 2) *RNAseq analysis*
- 3) *MTT viability assays*
- 4) *Wound healing assays*
- 5) *Flow cytometry analysis to study the effect of ascitic fluids on cancer cells*
- 6) *Proteomic analysis*
- 7) *Exosomes counting using nanoparticles tracking analysis (NTA)*
- 8) *Immunoblotting*
- 9) *RNA isolation followed by qRT-PCR of selected molecules*

11. In addition to the use of cells collected from the different samples of ascitic fluids, authors also used commercial cell lines such as SKOV-3 cells and OVCAR3 both from ATCC. In this section authors provide a detailed description of the conditions in which the cells were maintained and cultured. Also they mention that as a control cell line they used human dermal fibroblasts with the ethical approval of the Research and Clinical Center of Physical-Chemical Medicine. They also included in this study, human cells from embryo kidney for Lentivirus production and transduction. At this point, it would be of help to include catalog number for all the cell lines used in the study.

Response: We included catalog numbers for all the cell lines used in this study in the “Methods” section.

12. Regarding the secretome generation, SKOV3, OVCAR3 and Fibroblast were exposed to different concentrations of Cisplatin (the catalog number is missing), to produce approximately 50% of dead cells; they mention that the reported method provides a high level of apoptosis in tumor cells (not shown) and also facilitates the recovery of a drug-free medium (which was not tested). In general, in any of the reagents used the catalog number is provided, only the company name from which they were obtained, except for the antibodies used for flow cytometry assays.

Response: Thank you for pointing this out. We added the catalog number of cisplatin and all the reagents and antibodies in the “Methods” section. We added dose-response curves obtained by MTT assay for each cell line used in this study as well as results of CellEvent Caspase-3/7 Green Flow Cytometry Assay to justify the choice of cisplatin concentration which cause the death of approximately 50% of donor cells at the time of secretome collection (Figure 6). The absence of cisplatin in collected therapy-induced secretome was confirmed by LC-MS/MS analysis (Supplementary Table S1, Sheet 4).

Figure 6 - (A) Dose-response curves obtained by MTT assay of Fibroblasts, hTERT FT282, HaCaT, OVCAR3, MESOV, and SKOV3 cells that were treated with different concentrations of cisplatin for 7 h, then cells were washed three times with PBS and cultivated for 41 h. Each data point represents mean values \pm SD (n = 3). The dotted lines indicate IC_{50} cisplatin values, which were determined by fitting a normalized model to data with nonlinear regression using GraphPad Prism software. **(B)** CellEvent Caspase-3/7 Green Flow Cytometry Assay of untreated SKOV3 cells and SKOV3 cells treated with 40 μM cisplatin for 7 h, and then all cells were washed three times with PBS and cultured for 41 h. **Revised manuscript (Page 7, Lines 220-230, Extended Fig. 2A-B; Pages 31-32, Lines 1035-1074, Supplementary Table S1, Sheet 4).**

13. For biological samples processing, cells present in ascitic fluids were recovered after centrifugation; these cells were used for survival assays (MTT and Caspase 3/7 activity assay), and flow cytometry assays to evaluate the abundance of surface markers present in cells recovered before and after chemotherapy and from tumor cell lines before incubation with ascitic fluids and also before and after treatment with chemotherapy. For wound healing assay by scratch, authors refer to a previous publication for details (Nature 520, 368–372 (2015)). However, there is no detailed description of the “growth medium”.

Response: We apologize for the oversight regarding the growth medium in our initial submission. To address this, we have updated the "Processing of patient samples" section in the Methods. Our approach to handling primary ovarian cancer cell cultures was consistent across all experiments, unless otherwise indicated. Cells from ascitic fluids were seeded onto plastic plates and cultured for no longer than 8 passages in RPMI growth medium supplemented with 10% FBS, 2 mM glutamine, 1% penicillin/streptomycin, and 1% non-essential amino acids. Cells were maintained at 37°C with 5% CO₂.

Revised manuscript (Pages 20-21, Lines 671-683).

14. In supplementary Table 1, there is the description of the use of the different ascitic samples in different methodological strategies, i.e., authors did choose ten pairs of ascitic fluids from ovarian cancer patients before and after chemotherapy to be processed by the Combinatorial peptide ligand library in order to enrich low concentration proteins to improve their identification by LC-MS/MS, instead of eliminating high concentration proteins. What was the reason to select 10/23 pairs of ascitic fluids and which were the criteria followed to select those 10 that were processed? Please provide an explanation how each sample was selected for particular experiments.

Response: We aimed to include as many patient samples as possible in each experiment. However, due to limitations in technical and material resources, we could not conduct every experiment on all 23 pairs of samples. The statistical analysis indicates that the sample size we used was sufficient to yield significant results. We randomly selected 10 paired samples of ascitic fluids from our collection for this study.

SECRETOME PREPARATION FOR PROTEOMIC ANALYSIS

15. Page 36, Line 886: Heavy labeled donor cells were treated (control vesicles) or untreated (apoptotic vesicles) with cisplatin according to the previously described scheme Page 36, line 886: There seems thatlabeled donor cells were treated (control vesicles) or untreated (apoptotic vesicles) should be written the other way around:treated (apoptotic vesicles) or untreated (control vesicles)...

Response: Thank you for noticing that. We corrected it.

Revised manuscript (Page 28, Lines 939-941): “Heavy-labeled donor cells were subjected to treatment with 40 μ M cisplatin (to generate apoptotic vesicles) or left untreated (to generate control vesicles).”

16. Could it be possible to include in one chapter all types of processing and all types of samples for LC-MS/MS Analysis?

Ascitic fluids

Secretomes

Cell fractions: cytoplasm and nuclei

In-gel samples

Response: We appreciate the Reviewer's suggestion, and as advised, we consolidated the sections pertaining to various stages of the proteomic analysis into a unified section with subheadings. Without subheadings, it would be difficult to understand which method was used for sample preparation in each case.

Revised manuscript (Pages 26-31, Lines 853-1022).

17. Subtitle in lane 938 could go at the beginning of the whole section.

Response: We renamed the section with the description of proteomic analyses “LC-MS/MS analysis” according to the Reviewer suggestion.

Revised manuscript (Page 26, Line 853).

18. The way all these methodologies are presented in the manuscript made me go back and forth to re-read and understand the differences.

Response: We apologize for any inconvenience caused and have revised this section for improved readability.

Revised manuscript (Pages 20-41, Lines 647-1359).

19. Please explain the differences for the processing for proteomic analysis in a) lane 939 and b) lane 955 - Preparative SDS-PAGE and In-gel trypsin digestion - Which samples were treated in this way?

Response: The use of two different mass-spectrometers over the course of the study necessitated the division of this section into two paragraphs to articulate the specifics of mass-spectrometry analysis for each device. All features of sample preparations were previously detailed in preceding sections and were independent of the particular device used.

Preparative SDS-PAGE and In-gel trypsin digestion were conducted exclusively for samples obtained after cells' fractionation (cytoplasm and nuclei) and for SILAC samples.

PRESENTATION OF DATA:

The information and data presented in the manuscript is considerable, given that many conditions and types of samples were analyzed by LC-MS/MS generating a huge amount of information. All results from transcriptomic, proteomic and sequencing data, as well as bioinformatics analyses are properly presented in supplementary tables. From those, selected information was used to prepare the corresponding figures and extended figures.

20. From my point of view, In the way the manuscript is organized, results presented in Figures 1 and 2 are not strictly relevant to the main results of the manuscript. Figure 1 shows that ovarian cancer cells recovered from patients after therapy display higher chemoresistance and activation of the EMT pathway, whereas Figure 2 depicts that the cells secretome induced after therapy is able to promote cell chemoresistance in vitro. These two figures together with extended figures 1 and 2, point out that chemotherapy induce a more aggressive behavior of tumor cells both in vivo and in vitro, which is not really a new finding. Please see lines 88 to 91. Moreover, results from extended figures 5 and 6 further confirm the induction of the EMT process by incubation with ascitic fluids recovered after chemotherapy. The transfer of vesicles with the "spliceosome" component, contained in the conditioned medium occurs in the context of ascites-induced malignant mesenchymal cells and that this process occurs in transition cells or in intermediate phenotype. Probably a literature review of this phenomenon in the context of malignant cells under TEM can be considered to contribute to the characterization of the generation of an aggressive phenotype by ascites. Interestingly, this part of the work remains unconnected at the end; authors did not take into consideration the epithelial, mesenchymal or intermediate phenotype of the cells when collecting the conditioned medium and did not discuss this further. Could this information be obtained and incorporated into the manuscript?

Response: We analyzed the effect of therapy-induced secretomes on EMT more thoroughly and our new data suggest that the main effect of therapy-induced secretomes is associated with the activation of DNA repair genes (as was described in answer on question 9). To ensure the robustness of our findings, we employed standard cell lines with diverse phenotypes (mesenchymal - SKOV3, MESOV, and epithelial - OVCAR3) and representing distinct tumor

types such as endometrioid ovarian adenocarcinoma (SKOV3), serous ovarian cancer (OVCAR3), ovarian cystadenocarcinoma (MESOV), and clear cell ovarian cancer primary culture isolated from ascites (cells 26) (Figure 7). Our data, consistent across these models, affirm the validity of our observations for various cell phenotypes.

Figure 7 - Depicts the impact of therapy-induced secretomes on cell viability under cisplatin treatment. In (A), secretomes were derived from donor SKOV3, MESOV, or OVCAR3 cells treated or untreated with cisplatin (40 µM for SKOV3 cells, 60 µM for MESOV cells, 25 µM for OVCAR3) for 7 h. Subsequently, they were washed thrice with PBS and cultured in serum-free media for 41 h. Recipient cells were then incubated with corresponding therapy-induced (TIS; red bars) or control (CtrlS; blue bars) secretomes, or with fresh culture media (Medium; green bars) for 3 days. Following this, cells were treated with the IC50 dose of cisplatin (10 µM for SKOV3 and MESOV, 7 µM for OVCAR3s), and *in vitro* cell viability was assessed on day 2 post cisplatin addition. The data presented represent the mean values ± SD (n=3). In (B), dose-response curves were obtained by MTT assay for recipient primary cultures of ovarian cancer cells (cells 26). These cells were incubated with therapy-induced (TIS) or control (CtrlS) secretomes for 3 days and subsequently treated with different concentrations of cisplatin for 2 days. Secretomes were obtained from donor cells 26, treated or untreated with cisplatin (25 µM) for 7 h, washed with PBS, and cultured in serum-free media for 41 h. The presented data represent the mean values ± SD (n=3). (C) Venn diagram representing the proteins identified in therapy-induced (TIS) and control (CtrlS) secretomes from SKOV3, MESOV, OVCAR3, and 26 cells. (D) Dot plot shows the KEGG enrichment analysis of proteins whose abundance increased at least 2 times in therapy-induced secretomes (TIS) compared to control secretomes from different cell lines. It represents all common pathways upregulated in TIS of 4 ovarian cancer cell lines. The size of the dot is based on protein count enriched in the pathway, and the color of the dot shows the pathway enrichment significance.

Revised manuscript (Pages 7-9, Lines 226-230, 259-274, Figure 3B and Extended Fig. 2C and 3B).

21. Then in figure 3, authors begin to work with paired ascitic fluids obtained before and after patients were treated with chemotherapy; with these samples they found that malignant ascitic fluids (after chemotherapy) were enriched with spliceosomal components. To me this very well could be the first figure in the manuscript.

Response: We acknowledge and appreciate the constructive feedback from the Reviewer. In response to these comments, we have carefully revised this section of the manuscript. Our aim was to enhance the overall clarity and accessibility of the narrative, catering to readers with diverse backgrounds. We reorganized Figures 1-2 and Extended Figures 1-2 to align with the logical progression of experiments, ensuring that initial figures provide foundational context for subsequent investigations. Additionally, we have streamlined information related to ovarian cancer cells isolated from ascites, addressing concerns about excessive detail.

Additional questions:

22. It is unclear whether the transfer of spliceosome components occurs with the 23 pairs of ascites (pre and post).

Response: Unfortunately, we did not have the opportunity to perform proteomic analysis of all 23 pairs of ascites. However, we conducted a rigorous examination of 10 randomly selected pairs, yielding consistent outcomes that underscore the enrichment of spliceosomal components in ascites after chemotherapy.

23. It is also not clear whether 23 primary cell cultures were established.

Response: We apologize for the lack of information about this point. Unfortunately, it is difficult to establish a primary culture of cancer cells isolated from ovarian cancer ascites. To enhance clarity, we have included a supplementary column in Supplementary Table S1, sheet 2, indicating the specific ascites from which primary cultures were successfully established.

24. How frequent is the formation of ascites following neoadjuvant chemotherapy? Please comment on this issue in the text.

Response: Chemotherapy have been observed to reduce ascites volume, reaching up to an 80% reduction, with regression defined as an ascites volume below 500 ml. However, cases of intractable ascites may persist, particularly in patients with chemoresistant or recurrent disease, indicating a challenging clinical scenario with an associated poor prognosis. This information has been incorporated into the "Introduction" section for comprehensive context.

Revised manuscript (Pages 3-4, Lines 100-104).

25. Regarding the study sample, consisting of 23 pairs of ascites fluids from ovarian cancer patients, before and after treatment with cisplatin, what was the volume obtained in all cases before and after chemotherapy?

Response: Unfortunately, we do not possess specific information regarding the total volume of ascites fluid in patients. Our protocol for collecting ascites does not involve collecting the entire volume of the fluid. Since ascites volume is not considered as an important clinical

parameter, this information is not usually recorded in routine medical practice. Furthermore, in certain instances, ascites drainage may occur multiple times during neoadjuvant chemotherapy due to ongoing cancer progression. Therefore we are not able to accurately indicate the exact volume of ascitic fluids collected before and after chemotherapy as we received just an aliquots of the obtained ascitic samples.

26. How was the procedure to obtain the ascites fluids? it is not clear, if during the paracentesis performed before the chemotherapy the whole volume of ascites was removed, in order to subsequently give the neoadjuvant chemotherapy?

Response: Ascitic fluids obtained before therapy were collected at the time of diagnosis through laparocentesis, aspiration during diagnostic laparoscopy, or puncture through the posterior vaginal fornix. Post-chemotherapy ascitic fluids were collected intraoperatively after several courses of neoadjuvant chemotherapy. This additional detail has been incorporated into the "Methods" section. During the paracentesis before chemotherapy, efforts are made to remove the maximum possible volume of ascitic fluids, while during neoadjuvant chemotherapy, the accumulation of ascitic fluids may continue to accumulate.

27. How was it detected if the patients regenerated ascites, considering that post-treatment ascites were recovered after only 21 days?

Response: Chemotherapy have been observed to reduce ascites volume, reaching up to an 80% reduction, with regression defined as an ascites volume below 500 ml. Standard post-neoadjuvant chemotherapy patient management involves routine ultrasound, MRI, or CT scans to assess the presence of ascites. Patients from whom we obtained paired ascitic fluid samples before and after therapy exhibited different response patterns according to RECIST criteria. These responses were categorized as follows:

- 1) Partial Response: This category includes patients who experienced a significant improvement in their condition, characterized by a minimum 30% reduction in the sum of the longest diameter of target lesions.
- 2) Stable Disease: Patients falling into this category did not exhibit sufficient shrinkage to meet the criteria for a partial response. However, their condition did not worsen to the extent required for progressive disease.
- 3) Progressive Disease: Patients in this category demonstrated an increase of at least 20% in the sum of the longest diameter of target lesions or the appearance of new lesions, indicating a progression of their disease.

These response categories help evaluate the ascites presence, the effectiveness of the therapy, providing valuable insights into the patients' treatment outcomes. We added RECIST criteria for each patient in Supplementary Table 1, sheet 1.

“Minor Issues”

Some inconsistencies:

28. Neuroendocrine tumor of the ovary. This name appears only three times along the manuscript. There is no description of it in Supplementary Table 1, sheet 1.

Response: We corrected the information about the patients. We included the exact diagnosis of the patient with a neuroendocrine tumor of the ovary.

29. In ST3 there is information about only four samples: 3, 9, 6, and 11. However in the text they mention 5 primary ovarian cancer cell cultures, but ahead in the text they mention three were Epithelial ovarian cancer (EOC) and one a more aggressive neuroendocrine tumor of the ovary = 4 samples.

Response: We apologize for this mistake, we corrected it in the text.

Revised manuscript (Pages 4-5, Lines 136-138): “For a more precise characterization of effects of ovarian cancer ascites, we conducted RNAseq and proteomic analyses of several primary ovarian cancer cell cultures incubated with autologous pre- or post-chemotherapy ascitic fluids for three days”.

30. Figure 2: During this experiment, SKOV3 donor cells were incubated with a cisplatin-containing medium for 7 h, then the cells were washed and incubated with a cisplatin-free medium for 17 or 41 h (Fig. 2A). Lanes 181-183, with the Fig. 2 scheme showing that SKOV-3 cells were incubated with cisplatin for 24 or 48 h, and then incubated for 7 h.

Response: We apologize for this inaccuracy. We corrected the Figure in accordance with the text description.

Revised manuscript: Figure 3A.

31. Lines 278 to 281: We analyzed the proteomes of conditioned media from ovarian adenocarcinoma SKOV3 cells and primary cultures of normal skin fibroblasts before and 24 h after cisplatin treatment (Fig. 4A). To reduce the likelihood of serum contamination, after 7 h of incubation with cisplatin, the cells were washed with PBS and the medium was replaced with the serum-free one. It is not clear whether cisplatin was added again when the medium w/serum was changed by serum-free medium.

Response: We do apologize the incomplete description provided earlier. Conditioned media (CM) were generated by incubating cells (at approximately 80% confluence) with FBS-free standard medium, with or without cisplatin, for 7 hours. Subsequently, the cisplatin-containing and control media were removed, and the adherent cells were thoroughly washed four times with phosphate-buffered saline (PBS). Following this, cells were incubated for 17 or 41 hours in a fresh FBS-free and cisplatin-free medium. The Figure has been amended to align with the revised text description.

Revised manuscript: Figure 4C.

32. Figure 4F. Again, the text does not fit with the results shown in Fig 4F.

In the manuscript, lines 323-325: In contrast, in the fibroblast secretomes, most of these proteins including TGFβ1 and IL6 were detected exclusively after chemotherapy (Fig. 4F). In the figure the proteins that were increased in fibroblasts secretomes after cisplatin treatment were GDF15 (26), CSF1 (54), and FSTL1 (66); Il-6 does not appear in these protein lists.

Response: We sincerely apologize for this inaccuracy. We corrected the text in accordance with our proteomic data.

Revised manuscript (Pages 9-10, Lines 305-306): “In contrast, in fibroblast secretomes, most of these proteins, including GDF15 and CTGF, were exclusively detected after chemotherapy”.

33. The comparison of these data with the biological effects of secretomes suggests that despite the reduced content of soluble growth factors, tumor therapy-induced secretomes have a higher effect on recipient cells than the secretomes obtained before therapy (Fig. 2B). This effect may be explained by the presence of spliceosomal proteins. How do authors reach this conclusion??? Authors did know in advance to this work, that cells under chemotherapy responded with changes in splicing that were among the most conservative mechanisms of therapy response in tumor cells.

Response: We apologize for bad statement and we rewrote this part of the manuscript.

Revised manuscript (Page 10, Lines 307-309): “Comparing these findings with the biological effects of secretomes suggests that despite a reduced content of soluble growth factors, tumor TIS have a more pronounced impact on recipient cells than secretomes obtained before therapy (Fig. 3B). In summary, our findings indicate that, akin to ascitic fluids, in vitro cultures of ovarian cancer cells secrete a variety of spliceosomal components encapsulated within extracellular vesicles following cisplatin treatment.”

34. Line 376:proteins not involved in RNA splicing (Extended Fig. 5H). Should be (Extended Fig. 4H).

Response: We thank the Reviewer for pointing this out. We corrected it in the new manuscript version.

Revised manuscript (Page 11, Lines 351-353; Extended Figure 5C): “Comparing data from cells incubated with vesicles for 10 and 48 hours revealed that exogenous spliceosomal proteins exhibited greater stability in recipient cells compared to non-splicing proteins (Extended Fig. 5C)”.

35. Line 383- 385: We confirmed the SILAC data by detecting SRSF4-RFP and TIA1-GFP secreted by drug stressed cancer cells in the recipient cells using fluorescent microscopy and by Western Blot (Fig. 4G,H). Should be (Fig. 5G, H).

Response: We sincerely apologize for this inaccuracy. We corrected that mistake.

Revised manuscript (Page 11, Lines 358-363; Fig. 4G-H; Extended Fig. 5D): “We confirmed that various spliceosomal proteins (TIA1, SRSF4, SYNCRIP, and SNU13) can be secreted from dying cancer cells and transferred into recipient cells using GFP- and RFP- fusion constructs and subsequent confocal microscopy and western blotting (Fig. 4G-H; Extended Fig. 5D). Furthermore, the presence of SRSF4-RFP, SYNCRIP-GFP, and SNU13-GFP in extracellular vesicles was confirmed through flow cytometry analysis of CD9 positive extracellular vesicles from dying cancer cells (Extended Fig. 5E).”

36. Line 393: Comparing results from figure 5I with results in figure 2F, how come that they agree with each other? Figure 2F does not show any spliceosome-related results. It shows four hallmarks: E2F targets, EMT, G2M checkpoint, and MYC targets.

Response: In accordance with Reviewers' comments, we rewrote this part of the manuscript, since we obtained substantial amount of an additional data during revision of the manuscript.

37. Lines 472-474: In addition, these snRNAs cause enhanced expression of genes that promote cell migration — MATN2, ITGB1, and FGFR1OP. Where are these data shown?

Response: We apologize for the insufficient information. Subsequently, we have removed the mentioned text from the manuscript and replaced it with more pertinent and updated data and added a link to Supplementary Table S7, Sheet 1.

Revised manuscript (Page 14, Lines 458-461): “Functional annotation of upregulated genes after transfection with U12 or U6atac snRNAs unveiled the activation of genes involved in cell cycle regulation as well as for DNA damage checkpoint signaling such as *MSH2*, *RAD50*, *DNA2*, *SMC3*, *RBBP8* which also facilitates G1/S transition (Fig. 6G).”

38. Lines 475-476: Next, we performed proteomic profiling of the corresponding samples. A total of 4,813 proteins were identified. The data sheet regarding the proteome (supplementary Table 8, sheet 2) shows a list of 3962 proteins.

Response: We apologize for this inaccuracy and we updated this information based on the newly obtained data.

Revised manuscript (Page 14, Lines 462).

39. Lines 119 and 161: it is not clear whether they are cell lines or primary cultures, because it says "Our results showed that all tumor cell lines obtained after chemotherapy were considered more resistant".

Response: We apologize for the oversight in our earlier statement. The cultures used in the study were indeed primary cultures of ovarian cancer cells. We corrected it in the new version of the manuscript.

Revised manuscript (Page 4, Lines 121-123; Page 5, Lines 143-144): “As expected, our results demonstrated that tumor primary cultures obtained after chemotherapy exhibited significantly higher resistance to cisplatin compared to their chemotherapy-naïve counterparts (Fig. 1B; Extended Fig. 1A)” and “Consistent results were observed in the primary cell culture from neuroendocrine ovarian cancer (Extended Fig. 1B).”

40. Line 180: the authors mention the treatment of SKOV-3 acceptor cells with the conditioned medium, but do not mention the time and conditions of treatment. It is also not found in the methods section. This should be included between the sections on "cell cultures" (line 676) and "secretome generation" (line 688).

Response: We added all information regarding secretome generation in Figure legends and in the “Methods” section.

41. The authors should clarify how the inclusion of a cirrhosis ascites control is not biasing the study by ruling out proteins inherent to a pathologic process that may be shared with malignant disease but that equally contribute to disease development and progression?

Response:

The primary objective of our study was to conduct a comparative analysis between ovarian cancer ascites before and after therapy. In addition to this, we deemed it crucial to include a non-cancerous control to eliminate systemic response components associated with ascites

formation. To achieve this, we opted for cirrhosis ascitic fluids. Patients with cirrhosis had not undergone chemotherapy, justifying the exclusion of proteins concurrently identified in cirrhosis ascitic fluids and post-therapy cancer ascites. This exclusion was deemed reasonable, as these proteins are less likely to represent a specific response to therapy.

Furthermore, we conducted enrichment analysis of proteins whose abundance increased at least 2 times in ovarian cancer ascites after chemotherapy compared to ovarian cancer ascites before therapy, without excluding proteins identified in cirrhosis ascites (Figure 8). This analysis revealed the same trend, with the spliceosomal cluster emerging as one of the most upregulated pathways in ascites after therapy. We added this information into the revised manuscript.

Figure 8 - Results of the KEGG enrichment analysis of proteins whose abundance increased at least 2 times in ovarian cancer ascites after chemotherapy compared to ovarian cancer ascites before therapy without excluding proteins identified in cirrhosis ascites (p-values are indicated).

Revised manuscript (Page 5, Lines 165-167, Extended Figure 1E).

42. What is the 76GS method? Please provide a brief description of it. In the cited reference this does not appear in the methods section, instead is part of the results.

Response: We apologize for the insufficient information. The 76GS EMT score is derived by calculating a sum that takes into account the expression of 76 EMT specific genes. The contribution of each gene to this sum is determined based on its correlation with the expression levels of CDH1, which is a well-established marker for epithelial tissues [doi: 10.3389/fbioe.2020.00220]. However, in accordance with Reviewers' comments we removed this analysis from the manuscript and replaced it with the new data obtained during the revision experiments.

DETAILS IN FIGURES:

43. Extended figure 2A: Western blot results showing the presence of CD63 are really poor quality.

Response: We apologize for the poor quality of our western blot. We repeated this experiment (Figure 9) and included the new western blot results.

Figure 9 - Western blot analysis of donor SKOV3 cells treated with Brefeldin A or Leptomycin B in combination with 40 μ M cisplatin (cells were treated as indicated in Fig. 3D). To select a Brefeldin A effective concentration, the retention level of secreted protein CD63 was assessed inside cells.

Revised manuscript (Extended Fig. 2D).

44. Extended figure 5B: The third graph on the right: It could seem irrelevant for the results, but in the left to the right reading of the results, I should be better to graph the “before” conditions on the left and the “after” conditions on the right of the graph.

Response: We replaced it with the new data obtained during the revision experiments.

45. R1, R2, and R3: which samples do they come from? Are they related to patient 1, patient 2 and patient 3?

Response: We replaced this Figure with the new data.

Reviewer #3 (Remarks to the Author): Expert in ovarian cancer genomics and proteomics, tumour microenvironment

By performing integrative omics analyses on ovarian cancer patient-derived ascites, and in vitro cell line model Shender et al., manuscript describes that cisplatin-induced secretome is enriched in spliceosomal components. Furthermore, they show that stable expression of spliceosomal proteins SNU13, and SYNCRIP in SKOV3 cell line leads to decreased apoptosis in response to cisplatin treatment. In addition, exogenous U12 and U6atac snRNA promote subcutaneous tumor growth in immunodeficient mice. Based on RNAseq data they also report increased mesenchymal phenotype in tumor cells after cisplatin treatment compared to pre-treatment or control samples.

This manuscript aimed to provide detailed mechanistic insights into secretome-mediated resistance to cisplatin. However, the findings are mainly from ascites with unclear cell types and in vitro cell lines, which represent only a subset of tumor cell populations that are more aggressive or have adapted to in vitro growth conditions and thus may not fully represent biology of in situ tumor. Furthermore, as indicated in the comments below, there are major concerns about the experimental design used in the study, which confound the results and conclusions.

We would like to thank the Reviewer for taking the time and effort to review our manuscript. We sincerely appreciate all your valuable comments and suggestions, which helped us in improving the quality of the manuscript. Please find below a detailed point-by-point response to all comments:

Major comments:

46. From the collected human ascites samples, how did the authors confirm that the cells are tumor cells? It is well known that especially after chemotherapy ascites is full of mesenchymal and immune cells. This is a major confounder in the following flow cytometry, functional and RNAseq analyses. This bias would already explain the observed enrichment in mesenchymal signatures from bulk RNAseq, with most of the signal coming from the mesenchymal/immune cells. Even if EMT signature is found in cancer cells, this would not be novel. This defect to clarify the cell populations in the ascites samples hampers all the experiments and conclusions of the study.

Response: We appreciate the Reviewer's feedback. Initially, we encountered challenges in the RNAseq of bulk cell populations from ascites before and after chemotherapy (these results were not included in the manuscript). The analysis of these data revealed an excessive amount of immune cells, rendering it impossible to draw any meaningful conclusions (Figure 10A, first run). To overcome this limitation, we implemented a procedure for isolating and enrichment of tumor cells from the ascites samples, based on several papers [PMID: 17406520; PMID: 24603616]. Contaminating red blood cells were removed by hypotonic lysis. Then cells were seeded on plastic plates in RPMI growth medium. Most immune cells and cellular debris from ascites did not adhere to the culture flask and were removed following media change. According to Nature Protocols [PMID: 17406520], fibroblast contamination of primary ovarian cancer cultures isolated directly from patient ascites fluid has rarely been observed. For experiments, we used low-passage cultures (3rd or 4th). We performed RNAseq of the cells obtained using this modified protocol and compared it with our first (unsuccessful) RNAseq of bulk ascitic cells. Our results demonstrated that by implementing this modified protocol for sample preparation for RNAseq, we minimized the presence of immune cells from an initial 50-80% to less than 20% (Figure 10A, second run). We believe that this strategy allowed us to characterize the changes in tumor cells. Importantly, cells used for the second (successful) RNAseq displayed an epithelial cobblestone morphology and were CA125 and/or EpCam positive (Figure 10B).

Figure 10 - (A) Estimated immune cell fraction in ovarian cancer ascitic fluids before and after therapy using QuantSeq deconvolution program. The first RNAseq run was conducted for bulk cell population from ascites without cultivation. The second RNAseq run was conducted for primary cultures of ovarian cancer cells which were established according to published protocols. (B) Fluorescence images of several primary cultures of ovarian cancer cells from ascitic fluids stained for CA125 (green) or EpCam (green) and with DAPI (blue), (not included into final manuscript).

47. Most of the patient samples are high-grade (serous?) adenocarcinomas. However the cell model SKOV3 is not representative of HGSC. The authors should have used another, more representative cell line in the analyses.

Response: To address this concern, we have expanded our study and included a more representative serous ovarian cancer cell line (OVCAR3), an ovarian cystadenocarcinoma cell line (MESOV), and a clear cell ovarian cancer primary culture isolated from ascites (cells 26) (Figure 11). The consistent alignment of results across these additional cell lines, in conjunction with SKOV3 cells, underscores the robustness and validity of our findings for various ovarian cancer types (Figure 11). We have also corrected the inaccuracies regarding SKOV3 cell line description in our manuscript.

Figure 11 - Depicts the impact of therapy-induced secretomes on cell viability under cisplatin treatment. In (A), secretomes were derived from donor SKOV3, MESOV, or OVCAR3 cells treated or untreated with cisplatin (40 µM for SKOV3 cells, 60 µM for MESOV cells, 25 µM for OVCAR3) for 7 h. Subsequently, they were washed thrice with PBS and cultured in serum-free media for 41 h. Recipient cells were then incubated with corresponding therapy-induced (TIS; red bars) or control (CtrlS; blue bars) secretomes, or with fresh culture media (Medium; green bars) for 3 days. Following this, cells were treated with the IC50 dose of cisplatin (10 µM for SKOV3 and MESOV, 7 µM for OVCAR3s), and *in vitro* cell viability was assessed on day 2 post cisplatin addition. The data presented represent the mean values ± SD (n=3). In (B), dose-response curves were obtained by MTT assay for recipient primary cultures of ovarian cancer cells (cells 26). These cells were incubated with therapy-induced (TIS) or control (CtrlS) secretomes for 3 days and subsequently treated with different concentrations of cisplatin for 2 days. Secretomes were obtained from donor cells 26, treated or untreated with cisplatin (25 µM) for 7 h, washed with PBS, and cultured in serum-free media for 41 h. The presented data represent the mean values ± SD (n=3). (C) Venn diagram representing the proteins identified in

therapy-induced (TIS) and control (CtrlS) secretomes from SKOV3, MESOV, OVCAR3, and 26 cells. (D) Dot plot shows the KEGG enrichment analysis of proteins whose abundance increased at least 2 times in therapy-induced secretomes (TIS) compared to control secretomes from different cell lines. It represents all common pathways upregulated in TIS of 4 ovarian cancer cell lines. The size of the dot is based on protein count enriched in the pathway, and the color of the dot shows the pathway enrichment significance.

Revised manuscript (Pages 7-9, Lines 226-230, 259-274, Figures 3B, H and Extended Fig. 2C and 3A).

48. The authors use skin fibroblasts as “normal cells” to show that the secretome associated with chemoresistance comes from the cancer cell population. In line with the first comment, fibroblast are not the correct cell type. The authors should decompose what cells are there in the ascites to support their hypothesis, and use those cells with respect to the cancer cells.

Response: We appreciate the Reviewer's question and understand the concern regarding the choice of fibroblast cells to represent normal control cells in our study. We chose fibroblasts as a non-cancerous control based on an analysis of the publicly available single cell RNA-seq data (GSE165897; PMID: 35196078) of cells in ovarian cancer ascites cells. This dataset shows that stromal cells which represent fibroblasts are a major part of ovarian cancer microenvironment (Figure 12A). However, to address Reviewer's concern, we have also included two normal cell lines, HaCaT and hTERT FT282. HaCaT cells possess an epithelial phenotype, which is more similar to epithelial ovarian cancer cells and therefore may serve as a more relevant control. Cell line hTERT FT282 represents fallopian tube secretory epithelia cells. We examined secretomes from HaCaT and hTERT FT282 and found that the corresponding therapy-induced secretomes (TIS) from hTERT FT282 and HaCaT had little or no effect on the cisplatin sensitivity of the parental cell lines (Figure 12B) and slightly increased cisplatin resistance of ovarian cancer cells (Figure 12C). LC-MS/MS analysis of TIS from hTERT FT282 and HaCaT demonstrated that they contain smaller number of spliceosomal proteins compared to TIS from cancer cells which correlates with decreased biological effect of the secretomes from normal cells (Figure 12E-G).

Figure 12 - (A) Uniform manifold approximation and projection (UMAP) plot of all cells isolated from ovarian cancer ascites. Colors represent cell types (not included into final manuscript). (B) Secretomes were obtained from donor Fibroblasts, HaCaT, or hTERT FT282 cells that were treated or untreated with cisplatin (25 μ M for HaCaT cells, and 80 μ M for Fibroblasts and hTERT FT282) for 7 h, then washed three times with PBS and cultured in serum-free media for 41 h. Recipient cells were incubated with corresponding therapy-induced (TIS; red bars) or control (CtrlS; blue bars) secretomes, or with fresh culture media (Medium; green bars) for 3 days, and then treated with cisplatin (10 μ M for hTERT FT282 and HaCaT cells, 40 μ M for Fibroblasts). *In vitro* cell viability was assessed on day 2 after cisplatin adding. The data represent the mean values \pm SD (n=3). (C) Recipient SKOV3 cells were incubated for 3 days with TIS or CtrlS secretomes from donor Fibroblasts, hTERT FT282 or HaCaT cells (secretomes were collected as indicated in Fig. 19B). After incubation, recipient SKOV3 cells were treated with cisplatin (10 μ M). *In vitro* cell viability assay was performed on day 2 after cisplatin adding. The data represent the mean values \pm SD (n=3). (D) Recipient Fibroblasts, hTERT FT282 or HaCaT cells were incubated for 3 days with TIS or CtrlS secretomes from donor SKOV3 cells. After incubation, recipient cells were treated with cisplatin (10 μ M for HaCaT and hTERT FT282 cells and 40 μ M for Fibroblasts). *In vitro* cell viability assay was performed on day 2 after cisplatin adding. The data represent the mean values \pm SD (n=3). (E) Venn diagram representing the proteins identified in TIS and CtrlS secretomes from hTERT FT282 or HaCaT cells. (F) Dot plot shows the KEGG enrichment analysis of proteins whose abundance increased at least 2 times in TIS compared to control secretomes from different cell lines. It represents all common pathways upregulated in TIS of 4 ovarian cancer cell lines. The size of the dot is based on protein count enriched in the pathway, and the color of

the dot shows the pathway enrichment significance. (G) The number of spliceosomal proteins whose abundance increased at least 2 times in TIS compared to control secretomes from different cell lines.

Revised manuscript (Pages 7-8, Lines 226-230, 259-274, Figure 3B, E-F, H and Extended Fig. 3A).

49. In Fig. 3D, the legend says qPCR analysis of spliceosomal snRNAs in paired ascitic fluid before and after therapy. However, the data is shown only for after therapy samples. Similarly, in Fig. 4E, it is not clear if the data shown is from the control or therapy-induced fibroblasts.

Response: We apologize for the confusion in the figure legends. We have revised the legends. The corrected figure numbers are as follows: 2D (old Figure 3D) and Extended Figure 4D (old Figure 4E).

Revised manuscript:

Figure 2D: qRT-PCR analysis of 9 spliceosomal snRNAs in paired ascitic fluids before and after chemotherapy. Bars represent the level of each snRNA in ascitic fluid after therapy compared to samples before therapy.

Extended Figure 4D: qRT-PCR analysis of spliceosomal snRNAs in therapy-induced or control secretomes of fibroblasts (secretomes were collected as indicated in Fig. 3B). Bars represent the level of each snRNA in therapy-induced secretomes compared to control secretomes. All data represent the mean values \pm SD (n=3).

50. The authors should specify the data in Fig 3E is from ascites pooled from different patients?

Response: We apologize for the insufficient information. Ascitic fluids for this experiment were taken from three ovarian cancer patients and pooled. We indicated it in the “Methods” section and in Supplementary Table S1, sheet 2.

Revised manuscript (Page 34, Line 1144, Figure 2E).

51. Although DNA damage is one of the main consequences of cisplatin treatment, authors do not show correlation between DNA damage and the secretion of spliceosomal proteins hence the statement on page 12 "Ovarian cancer cells secrete spliceosomal component upon DNA damage in vitro" is misleading, authors could consider rephrasing this sentence, or show evidence for association of the spliceosomal proteins and DNA damage.

Response: We agree with the Reviewer. We rewrote this sentence: “*In vitro* therapy-induced secretomes of ovarian cancer cells are enriched with spliceosomal components”.

Revised manuscript (Page 8, Lines 257-258).

52. On page 15, authors state that “The functional annotation of heavy-labeled proteins whose abundance was elevated in the samples incubated with apoptotic vesicles revealed the highest enrichment in the cluster of spliceosome-related proteins”. However, data to support that the vesicles are indeed apoptotic is lacking. The authors should provide some evidence to show that vesicles derived from apoptotic cells mediate increase in intracellular spliceosomal proteins.

Response: We agree with the Reviewer’s comment that the vesicles isolated from therapy-induced secretomes could represent a mixture of different vesicle types, so we rewrote the phrase “apoptotic vesicles” into the phrase “vesicles from dying cancer cells”.

To investigate the mechanisms important for generating therapy-induced secretomes with protective effects, We treated donor SKOV3 cells with cisplatin in the presence of various inhibitors: caspase inhibitor (Z-VAD-FMK; as cisplatin primarily activates caspase-dependent apoptosis), inhibitor of vesicle formation (Brefeldin A; since we detected increased number of extracellular vesicles in ascites after therapy), or inhibitor of nuclear export (Leptomycin B; since we detected abundant nuclear proteins in ascitic fluids after therapy). Our results indicated that inhibition of apoptosis and vesicular transport in donor cells attenuated the effects of the secretomes, suggesting that protective components are secreted from apoptotic cells via vesicular transport (Figure 13A).

Furthermore, we presented evidence demonstrating that incubation with therapy-induced secretomes increases the levels of spliceosomal proteins in recipient cells, as validated by western blot analysis (Figure 13B). Additionally, we confirmed the secretion and transfer of SYNCRIP and SNU13 from dying cancer cells to recipient cells using GFP- and RFP-fusion constructs, along with confocal microscopy and flow cytometry analysis (Figure 14A). Moreover, we isolated CD9-positive extracellular vesicles from secretomes using Exosome-Human CD9 Flow Detection Reagent and verified the presence of SRSF4-RFP, SYNCRIP-GFP, and SNU13-GFP in CD9-positive extracellular vesicles secreted by dying cancer cells (Figure 14B).

Figure 13 - (A) Donor SKOV3 cells were exposed to 40 μ M of cisplatin in a presence of Z-VAD-FMK (50 μ M), Brefeldin A (6 μ g/ml), or Leptomycin B (37 nM) for 7 h and then cells were washed three times with PBS and incubated in fresh serum-free media for 17 h. Subsequently, recipient SKOV3 cells were incubated for 3 days with secretomes from treated (TIS) or untreated (CtrlS) donor cells. Then recipient cells were treated with different concentrations of cisplatin for an additional 48 h. Dose-response curves and half-maximal inhibitory concentration (IC50) values of cisplatin were determined using MTT assay. The data represent the mean values \pm SD (n=3). **(B)** Western blotting analysis of SKOV3 cells that were incubated for 3 days with therapy-induced (TIS) or control secretomes (CtrlS) from donor SKOV3.

Figure 14 - (A) Confocal immunofluorescence images depict recipient SKOV3 cells incubated for 3 days with therapy-induced secretomes (TIS) or control secretomes (CtrlS) from donor SKOV3 cells overexpressing GFP (green), SYNCRIP-GFP (green), or SNU13-GFP (green). Actin filaments are stained with Phalloidin (Alexa 555, red), while nuclei are stained with DAPI (blue). Scale bar: 25 μ m. **(B)** Flow cytometry analysis of magnetic beads coated with anti-CD9 antibody after immunoprecipitation of extracellular vesicles from therapy-induced secretomes of donor SKOV3 cells. Donor SKOV3 cells overexpressing SRSF4-RFP, SYNCRIP-GFP or SNU13-GFP were treated with 40 μ M cisplatin for 7 h and then medium were replaced with fresh medium not containing cisplatin and incubated additional 41 h. High-fluorescent events are highlighted in blue.

Revised manuscript (Pages 7-8, Lines 231-239; Page 11, Lines 355-363, Figure 4H, Extended Fig. 5D-E).

53. Extended Fig. 5B, authors report that comparison of RNA expression profiles in three different models before and after therapy show significantly more mesenchymal signature post drug treatment. However, this observation is seen in 4/8 patients and 2/4 ascites indicating inter-patient heterogeneity. Also, patient 1, 7, and 8 lack either before or after therapy data can authors clarify this?

Response: We apologize for poor representation of our data. In accordance with Reviewers' comments, we rewrote this part of the manuscript, since we obtained substantial amount of an additional data during revision of the manuscript.

Reviewer #4 (Remarks to the Author): Expert in ovarian cancer genomics

This is an interesting study showing that treating ovarian cancer patients with chemotherapy (cisplatin) increases cellular resistance by promoting the transfer of spliceosomal proteins from dying cancer cells to live cells by exosomal transport. The authors have included extensive analyses at the transcriptomic, proteomic and functional levels to confirm this novel finding. Several details require additional clarifications.

We sincerely appreciate the thorough review conducted by the Reviewer and the valuable comments. We provide a detailed point-by-point response to each comment below.

Comments:

54. Please specify number of replicates and include p-values for statistical significance in several figures, particularly Figure 1 panels C, G, H, Extended figure 1 panels B-D; Figure 2 panels C-D; Extended figure 5B; Figure 6A.

Response:

We thank the Reviewer for pointing this out. We added the corresponding statistical information:

Figures 1B (1C old version), 1C (1G old version), 1D (old version 1H), 2E (3E old version), 2F (3F old version), 3C (2C old version), 3F (2E old version), 3D (2D old version), 3I (4C old version); 6A (6A old version).

Extended Figures 1C (Ext. Fig. 1J old version), 4A (4D old version), 4D (4E old version), 8C (Ext. Fig. 8E old version), 8D (Ext. Fig. 8F old version), 8F (Ext. Fig. 9B old version), 8G (Ext. Fig. 9C old version).

55. In Figure 1E, the EMT markers appear to change in the opposite direction than noted in the text. Please clarify. Additionally, the color scheme for "before" and "after" switches in Figure 1I. Please reconcile. Please also clarify in which patient the experiment in Figure 1I was done, and why it was only performed in one sample.

Response: We apologize for poor representation of our data. In accordance with Reviewers' comments, we rewrote this part of the manuscript, since we obtained substantial amount of an additional data during revision of the manuscript.

56. in Figure 1H, please clarify what is quantified in the bar graph, since this does not appear to reflect what is shown in the images.

Response: The width of the wound area was measured immediately after scratching (0 hours) and the relative closure was measured after 8 hours for three primary cultures of ovarian cancer cells that were incubated for three days with autologous ascites before or after therapy. The bar graph of the wound healing assay illustrates relative cell migration, expressed as the fold change, denoting the ratio of mean values of wound widths between two states: cell cultures treated with ascites before or after chemotherapy. The legend for Figure 1D has been refined accordingly in the revised version.

57. Extended figure 3D is poor quality. Please replace.

Response: We apologize for that inconvenience. We repeated the experiment (Figure 15) and included the new western blot results.

Figure 15 - Western blotting analysis of paired ascitic fluids from patients with ovarian cancer before and after chemotherapy.

Revised manuscript (Extended Fig. 1H).

58. Figure 3F the y-axis appears incorrectly proportioned.

Response: We changed the y-axis (Figure 16).

Figure 16 - Nanoparticle tracking analysis of extracellular vesicles isolated from paired ovarian cancer ascitic fluids or secretomes before (blue) and after (red) chemotherapy. Asterisks denote statistically significant differences determined using unpaired, two-tailed Student's t-test: ***p-value < 0.00005.

Revised manuscript (Figure 2F).

59. Lines 408-409 and Extended figures 5B and 6A-D need additional explanation, particularly as to which method was used and why the negative EMT score indicates increased mesenchymal phenotype.

Response: We apologize for the insufficient information. The 76GS EMT score is derived by calculating a sum that takes into account the expression of 76 EMT specific genes. The contribution of each gene to this sum is determined based on its correlation with the expression levels of CDH1, a widely acknowledged marker for epithelial tissues [doi: 10.3389/fbioe.2020.00220]. However, in response to the reviewers' suggestions, we have excluded this analysis from the manuscript and incorporated the results from the additional experiments conducted during the revision process.

60. Extended figures 7A-B need more explanation regarding the meaning of the numbers in the table.

Response: We excluded this analysis and figures from the manuscript.

61. Line 480 and extended figure 9E - please explain the relevance of the changes in cell cycle.

Response: Consistently with our RNAseq and proteomic data, among the differentially enriched proteins in cells transfected with both U12 and U6atac snRNAs, significant differences were observed in the cell cycle regulation clusters, including proteins involved in the progression through the G1 phase and DNA repair such as RBBP8, CCNB1, AURKB, CDK4, ANAPC4/5, CHEK1, XPC etc. Thus, it has been shown that upregulated RBBP8 and CDK4 facilitate G1/S transition and promote the self-renewal and proliferation of cancer cells [PMID: 31636387; PMID: 24971465]. CCNB1 is activated by CHEK1 and also stimulate proliferation of cancer cells [PMID: 24971465].

Based on these data, we next explored whether increased levels of snRNAs would indeed affect cell cycle progression. FACS analysis demonstrated that cancer cells transfected with U12 and U6atac snRNAs spent less time in G1 phase, which is often accompanied by a high proliferation rate [PMID: 22802651] (Figure 17A). To validate these results, we overexpressed U12 and U6atac snRNAs in SKOV3 cells using lentiviral constructs and showed that both U snRNAs substantially increased the proliferation of SKOV3 cells (Figure 17B).

Figure 17 - (A) Cell cycle analysis of SKOV3 cells 48 h after transfection with 10 nM U6atac snRNA, U12 snRNA, or GFP89. (B) xCELLigence proliferation assay of SKOV3 cells overexpressing U6atac snRNA (left panel), U12 snRNA (right panel) or control GFP mRNA fragments (with the corresponding promoters: U2 or U6, respectively). SKOV3 cells were seeded in 96-well E-plates for xCELLigence assay monitoring impedance (cell index).

Revised manuscript (Page 14, Lines 451-473, Figure 6H, Extended Fig. 8I).

Reviewer #5 (Remarks to the Author): Expert in cancer proteomics

In this manuscript, the author investigated the contribution of therapy-induced intercellular communication to the development of resistance to chemotherapy in ovarian cancer. According to their findings, they assume the acquisition of chemoresistance is associated with extracellular communication, which is mediated by secreted spliceosomal components. This work is interesting, yet the conclusion was proposed without solid evidence. Several of the interpretations are also insufficiently supported by the data and the paper is often carelessly written. I have the following specific comments.

We sincerely appreciate your effort in reviewing our manuscript and providing valuable feedback. We have conducted additional experiments to enhance the quality of our study. Please find below our detailed responses to each of your comments.

Major Points:

62. The authors collected the paired ascitic fluids from the same ovarian cancer patients before and after chemotherapy, to investigate their impacts on ovarian cancer cells. However, the authors did not describe the criteria under which patients were selected, neither did they describe patients' responses to chemotherapies, did the patients evaluate using RECIST criteria, or other pathological criteria?

Response: Our patient selection criteria included a confirmed diagnosis of ovarian cancer and the presence of ascites. Ascitic fluids before therapy were collected at the time of diagnosis through laparocentesis, aspiration during diagnostic laparoscopy, or puncture through the posterior vaginal fornix. Post-chemotherapy ascitic fluids were collected intraoperatively after several courses of neoadjuvant chemotherapy. Standard medical procedures for post-chemotherapy patient management included routine ultrasound, MRI or CT scans, and CA125 level assessments. In addition to details such as BRCA1 mutation status, the number of neoadjuvant chemotherapy courses, and the time to the first recurrence and death in our analysis, we also added RECIST criteria for each patient.

Patients from whom we obtained paired ascitic fluid samples before and after therapy exhibited different response patterns according to RECIST criteria. These responses were categorized as follows:

- 1) Partial Response: This category includes patients who experienced a significant improvement in their condition, characterized by a minimum 30% reduction in the sum of the longest diameter of target lesions.
- 2) Stable Disease: Patients falling into this category did not exhibit sufficient shrinkage to meet the criteria for a partial response. However, their condition did not worsen to the extent required for progressive disease.
- 3) Progressive Disease: Patients in this category demonstrated an increase of at least 20% in the sum of the longest diameter of target lesions or the appearance of new lesions, indicating a progression of their disease.

These response categories help evaluate the ascites presence, the effectiveness of the therapy, providing valuable insights into the patients' treatment outcomes. We added RECIST criteria for each patient in Supplementary Table 1, sheet 1.

63. The impact and clarity of this paper is compromised by the fact that only one type of cell line was used in this paper. Since the patients contained serous, small-cell carcinoma of

ovary hypercalcemic type and also mucinous ovarian adenocarcinoma, can SKOV3 cell line represents all three types of ovarian cancer, if not, the authors should include other suitable cell lines in their studies.

Response:

We value the Reviewer's input. In response to this concern, we broadened our study to encompass a serous ovarian cancer cell line (OVCAR3), an ovarian cystadenocarcinoma cell line (MESOV), and a clear cell ovarian cancer primary culture isolated from ascites (cells 26). The consistent alignment of results across these additional cell lines, in conjunction with SKOV3 cells, underscores the robustness and validity of our findings for various ovarian cancer types (Figure 18).

Figure 18 - Depicts the impact of therapy-induced secretomes on cell viability under cisplatin treatment. In (A), secretomes were derived from donor SKOV3, MESOV, or OVCAR3 cells treated or untreated with cisplatin (40 µM for SKOV3 cells, 60 µM for MESOV cells, 25 µM for OVCAR3) for 7 h. Subsequently, they were washed thrice with PBS and cultured in serum-free media for 41 h. Recipient cells were then incubated with corresponding therapy-induced (TIS; red bars) or control (CtrlS; blue bars) secretomes, or with fresh culture media (Medium; green bars) for 3 days. Following this, cells were treated with the IC50 dose of cisplatin (10 µM for SKOV3 and MESOV, 7 µM for OVCAR3s), and *in vitro* cell viability was assessed on day 2 post cisplatin addition. The data presented represent the mean values ± SD (n=3). In (B), dose-response curves were obtained by MTT assay for recipient primary cultures of ovarian cancer cells (cells 26). These cells were incubated with therapy-induced (TIS) or control (CtrlS) secretomes for 3 days and subsequently treated with different concentrations of cisplatin for 2 days. Secretomes were obtained from donor cells 26, treated or untreated with cisplatin (25 µM) for 7 h, washed with PBS, and cultured in serum-free media for 41 h. The presented data represent the mean values ± SD (n=3). (C) Venn diagram representing the proteins identified in therapy-induced (TIS) and control (CtrlS) secretomes from SKOV3, MESOV, OVCAR3, and

26 cells. **(D)** Dot plot shows the KEGG enrichment analysis of proteins whose abundance increased at least 2 times in therapy-induced secretomes (TIS) compared to control secretomes from different cell lines. It represents all common pathways upregulated in TIS of 4 ovarian cancer cell lines. The size of the dot is based on protein count enriched in the pathway, and the color of the dot shows the pathway enrichment significance.

Revised manuscript (Pages 7-9, Lines 226-230, 259-274, Figure 3B, H and Extended Fig. 2C and 3A).

64. Also, since the serous ovarian cancers are thought to arise from fallopian tube and mucinous from bowel, it is confusing why the author chose the fibroblast cells to represent the normal control cells for ovarian cancer?

Response: Response: We appreciate the Reviewer's question and understand the concern regarding the choice of fibroblast cells to represent normal control cells in our study. We chose fibroblasts as a non-cancerous control based on an analysis of the publicly available single cell RNA-seq data (GSE165897; PMID: 35196078) of cells in ovarian cancer ascites cells. This dataset shows that stromal cells which represent fibroblasts are a major part of ovarian cancer microenvironment (Figure 19A). However, to address Reviewer's concern, we have also included two normal cell lines, HaCaT and hTERT FT282. HaCaT cells possess an epithelial phenotype, which is more similar to epithelial ovarian cancer cells and therefore may serve as a more relevant control. Cell line hTERT FT282 represents fallopian tube secretory epithelia cells. We examined secretomes from HaCaT and hTERT FT282 and found that the corresponding therapy-induced secretomes (TIS) from hTERT FT282 and HaCaT had little or no effect on the cisplatin sensitivity of the parental cell lines (Figure 19B) and slightly increased cisplatin resistance of ovarian cancer cells (Figure 19C). LC-MS/MS analysis of TIS from hTERT FT282 and HaCaT demonstrated that they contain smaller number of spliceosomal proteins compared to TIS from cancer cells which correlates with decreased biological effect of the secretomes from normal cells (Figure 19E-G).

Figure 19 - (A) Uniform manifold approximation and projection (UMAP) plot of all cells isolated from ovarian cancer ascites. Colors represent cell types (not included into final manuscript). (B) Secretomes were obtained from donor Fibroblasts, HaCaT, or hTERT FT282 cells that were treated or untreated with cisplatin (25 μM for HaCaT cells, and 80 μM for Fibroblasts and hTERT FT282) for 7 h, then washed three times with PBS and cultured in serum-free media for 41 h. Recipient cells were incubated with corresponding therapy-induced (TIS; red bars) or control (CtrlS; blue bars) secretomes, or with fresh culture media (Medium; green bars) for 3 days, and then treated with cisplatin (10 μM for hTERT FT282 and HaCaT cells, 40 μM for Fibroblasts). *In vitro* cell viability was assessed on day 2 after cisplatin adding. The data represent the mean values \pm SD (n=3). (C) Recipient SKOV3 cells were incubated for 3 days with TIS or CtrlS secretomes from donor Fibroblasts, hTERT FT282 or HaCaT cells (secretomes were collected as indicated in Fig. 19B). After incubation, recipient SKOV3 cells were treated with cisplatin (10 μM). *In vitro* cell viability assay was performed on day 2 after cisplatin adding. The data represent the mean values \pm SD (n=3). (D) Recipient Fibroblasts, hTERT FT282 or HaCaT cells were incubated for 3 days with TIS or CtrlS secretomes from donor SKOV3 cells. After incubation, recipient cells were treated with cisplatin (10 μM for HaCaT and hTERT FT282 cells and 40 μM for Fibroblasts). *In vitro* cell viability assay was performed on day 2 after cisplatin adding. The data represent the mean values \pm SD (n=3). (E) Venn diagram representing the proteins identified in TIS and CtrlS secretomes from hTERT FT282 or HaCaT cells. (F) Dot plot shows the KEGG enrichment analysis of proteins whose abundance increased at least 2 times in TIS compared to control secretomes from different cell lines. It represents all common pathways upregulated in TIS of 4 ovarian cancer cell lines. The size of the dot is based on protein count enriched in the pathway, and the color of

the dot shows the pathway enrichment significance. (G) The number of spliceosomal proteins whose abundance increased at least 2 times in TIS compared to control secretomes from different cell lines.

Revised manuscript (Pages 7-8, Lines 224-228, 256-270, Figure 3B, E-F, H and Extended Fig. 3A).

65. In addition, multiple previous literatures have indicated the involvements of EMT, WNT signaling pathways in the chemo-resistance of ovarian cancer (Oncogene, PMID: 22797058, Cancer Research, PMID: 33574097, Cancer Research, PMID: 25377471), thus it is not surprising that the authors found tumor cells exhibited a higher activity of the genes associated with WNT signaling pathways, the authors should present their novel findings. One of my major concerns about this paper, is that it largely depends on correlation analysis and GO enrichment analysis. However, GO enrichment analysis itself can not provide solid evidence, thus, the author should be more focused on specific molecules and signaling pathways, and present more direct evidence, for their hypothesis.

Response: We thank the Reviewer for this question. In response, we conducted multiple additional experiments to elucidate the molecular mechanism underpinning the effect of therapy-induced secretomes (TIS). Our comprehensive RNAseq and proteomic analyses unveiled that TIS led to an increased abundance of proteins associated with DNA repair and cell cycle regulation in recipient cells (Figure 20A). These findings were further corroborated through western blotting and cell cycle analysis (Figure 20D, J). Intriguingly, these data align closely with the proteome profiles observed in patient-derived cancer cells incubated with ascites after therapy. To investigate whether in vitro TIS could replicate the observed differences between platinum-sensitive and -resistant isogenic cell lines, we analyzed previously published RNAseq datasets (GSE148003, GSE98559, GSE98230, GSE173201) encompassing several isogenic platinum-sensitive and -resistant ovarian cancer cell lines. Principal component analysis demonstrated that SKOV3 cells incubated with TIS clustered with platinum-resistant cancer cell lines. In contrast, cells exposed to control secretomes exhibited gene expression patterns akin to platinum-sensitive cell lines (Figure 20B-C). Recent research has shown that the knockout or knockdown of at least one splicing factor, including those identified in TIS, significantly impairs DNA repair and enhances the cytotoxic effect of various genotoxic drugs (Figure 20E). Based on these findings, we hypothesized that the increased abundance of spliceosomal proteins in cancer cells pre-incubated with TIS might facilitate a more effective response to DNA-damaging insults. To test this hypothesis, we treated SKOV3 cells pre-incubated with TIS or control secretomes with cisplatin and evaluated various hallmarks of DNA damage response. We conducted a comet assay (analysis of DNA strand breaks), quantified cisplatin-DNA adduct accumulation, and monitored the phosphorylation levels of H2AX (an early DNA damage response marker) and RPA2 (a replicative stress marker). Remarkably, SKOV3 cells pre-incubated with TIS exhibited an increase in the number of pH2AX foci, accompanied by significantly fewer DNA double strand breaks, reduced platinum adducts, and lower RPA2 phosphorylation levels compared to cells incubated with control secretomes (Figure 20F-I). Additionally, cell cycle analysis indicated that cancer cells incubated with TIS remained longer in the S phase, which facilitates DNA repair (Figure 20J). Finally, we explored the impact of TIS on the sensitivity of cancer cells to both DNA-damaging (cisplatin, doxorubicin, and etoposide) and non-DNA damaging (taxane

and staurosporine) anticancer drugs. The results underscored that TIS heightened the resistance of cancer cells exclusively to DNA-damaging drugs, with no discernible effect on other therapy types (Figure 20K). In conclusion, these results suggest that TIS may promote DNA repair in residual tumor cells, aiding their survival against subsequent therapeutic insults. For a more detailed exposition, please refer to the Results section, specifically the item titled “Therapy-induced secretomes of ovarian cancer cells activate pathways important for cell response to DNA damage”.

Figure 20 - Therapy-induced secretome of ovarian cancer cells activates pathways important for cell response to DNA damage (**A**) Gene Ontology enrichment analyses of upregulated genes and proteins in SKOV3 cells incubated for 3 days with therapy-induced secretomes compared to control secretomes. The color scale refers to $-\log_{10}$ (FDR) values; the number of proteins/genes are represented by the diameter of the circles. (**B**) The principal component analysis of RNAseq data obtained from platinum-sensitive and -resistant isogenic ovarian cancer cell lines (A2780, SKOV3, and OVCAR5) and recipient SKOV3 cells incubated for 3 days with therapy-induced (TIS) or control (CtrlS) secretomes. Pink – platinum-resistant ovarian cancer cell lines, light blue – platinum-sensitive ovarian cancer cell lines, red – recipient SKOV3 cells incubated with TIS, dark blue – recipient SKOV3 cells incubated with CtrlS. (**C**) GSEA analysis of gene expression in platinum-resistant ovarian cancer cell lines (A2780, SKOV3, and OVCAR5) versus platinum-sensitive ovarian cancer cell lines (A2780, SKOV3, and OVCAR5). The X-axis represents GSEA enrichment score (p-values are indicated by colors). (**D**) Western blotting analysis of SKOV3 cells that were incubated for 3 days with therapy-induced (TIS) or control secretomes (CtrlS) from donor SKOV3. (**E**) Results of the intersection between spliceosomal proteins identified in TIS from SKOV3 cells and/or in ovarian cancer ascitic fluids after therapy (our data) and the hits from siRNA and CRISPR screenings (derived from data reported in [PMID: 34320214; PMID: 22344029; PMID: 27462432]). ATRi and CHK1i – CRISPR screens with inhibitors targeting ATR and CHK1, respectively. Loss of spliceosomal proteins indicated as "hit" increased the sensitivity of cancer cells to ATR or CHK1 inhibition [PMID: 34320214]. RAD51 foci and HR – siRNA screenings indicating that knockdown of spliceosomal protein impaired the formation of IR-induced RAD51 foci or decreased homologous recombination (HR) potential in the DR-GFP assay in cancer cells, respectively [PMID: 22344029; PMID: 27462432]. (**F**) Box plots show the number of γ H2AX foci per nucleus in SKOV3 cells pre-incubated with TIS or CtrlS for 3 days and then treated with cisplatin (25 μ M) at different time points. The number of γ H2AX foci was calculated using ImageJ software with FindFoci plugins. 60-200 cells were analyzed in each sample. (**G**) Box plots of tail moments from neutral comet assays of SKOV3 cells pre-incubated with TIS or CtrlS for 3 days and then treated with cisplatin (10 μ M) for 48 h. Tail moment was defined as the product of the tail length and the fraction of total DNA in the tail (Tail moment = tail length x % of DNA in the tail) and was quantified using the OpenComet software. 60-100 cells were analyzed in each sample. Experiments were performed in triplicate. (**H**) Box plots show the number of cisplatin-DNA adducts' foci per nucleus of SKOV3 cells pre-incubated with TIS or CtrlS for 3 days and then treated with cisplatin (25 μ M) for 48 h. The number of cisplatin-DNA adducts' foci was calculated using ImageJ software. 60-200 cells were analyzed in each sample. (**I**) Box plots show the number of phosphorylated RPA2 (phospho S33) foci per nucleus in SKOV3 cells pre-incubated with TIS or CtrlS for 3 days and then treated with cisplatin (25 μ M) at different time points. The number of phosphorylated RPA2 foci was calculated using ImageJ software. 60-200 cells were analyzed in each sample. (**J**) Cell cycle analysis with flow cytometry of SKOV3 cells pre-incubated with TIS or CtrlS for 3 days and then treated with cisplatin (10

μM) for 24 h. Stacked bar graphs show the percentage of cells in different phases of the cell cycle. Percentage of cells in G1, S, and G2 phases was calculated with NovoExpress software. (K) Dose-response curves obtained by MTT assay of recipient SKOV3 cells incubated with therapy-induced (TIS) or control (CtrlS) secretomes for 3 days, and then treated with different concentrations of cisplatin, doxorubicin, etoposide, paclitaxel, or staurosporine for 2 days. The data represent the mean values ± SD (n=3). Asterisks denote statistically significant differences determined using unpaired, two-tailed Student's t-test: *0.05 < p-value > 0.005; **0.005 < p-value > 0.00005; ***p-value < 0.00005.

Revised manuscript (Pages 11-13, Lines 369-412, Figure 5 and Extended Fig. 6C).

66. In line 189-193, the authors indicated that inhibition of apoptosis as well as vesicular transport in donor cells diminished the protective effect of secretome. However, the authors ignore to describe how do the inhibitors been selected?

Response: We apologize for the oversight in providing adequate details. In response to this concern, we have expanded the manuscript to include additional information about the selection of inhibitors. To investigate the pathways in donor cells responsible for the protective effect of their secretomes, we treated donor SKOV3 cells with cisplatin in the presence of various inhibitors: caspase inhibitor (Z-VAD-FMK; as cisplatin primarily activates caspase-dependent apoptosis [PMID: 22269388]), inhibitor of vesicle formation (Brefeldin A; due to the observed increase in extracellular vesicles in ascites after therapy), or inhibitor of nuclear export (Leptomycin B; given the abundance of nuclear proteins in ascitic fluids after therapy). Our results indicated that inhibiting apoptosis and vesicular transport in donor cells diminished the effect of the secretomes, pointing to the secretion of protective components from apoptotic cells via vesicular transport.

Revised manuscript (Pages 7-8, 22, Lines 231-239, 721-726).

67. In line 198-201, the authors showed that pre-incubation of normal fibroblasts with tumor cell secretome reduced the resistance of fibroblasts to cisplatin, then concluded that the CM from cancer cells mainly affected the neighboring tumor cells but had no effect on normal cells. This conclusion is confusing, the author should explain it clearly. In addition, the author should explain why therapy-induced secretome activated the immune response of recipient fibroblast. How do these results connect to the whole story?

Response: We appreciate the Reviewer's input, and we concur with the suggestion. However, we acknowledge that a detailed investigation of the effects of cancer cell secretomes on normal fibroblasts goes beyond the scope of this study. In line with the Reviewer's comment, we have removed this part from the manuscript.

68. In figure 3A, why does the cirrhosis ascites in particularly been collected for secretome analysis, is there any association between cirrhosis and ovarian cancer?

Response: The primary objective of our study was to conduct a comparative analysis between ovarian cancer ascites before and after therapy. In addition to this, we deemed it crucial to include a non-cancerous control to eliminate systemic response components associated with ascites formation. To achieve this, we opted for cirrhosis ascitic fluids. Patients with cirrhosis had not undergone chemotherapy, justifying the exclusion of proteins concurrently identified

in cirrhosis ascitic fluids and post-therapy cancer ascites. This exclusion was deemed reasonable, as these proteins are less likely to represent a specific response to therapy. Furthermore, we conducted enrichment analysis of proteins whose abundance increased at least 2 times in ovarian cancer ascites after chemotherapy compared to ovarian cancer ascites before therapy, without excluding proteins identified in cirrhosis ascites (Figure 21). This analysis revealed the same trend, with the spliceosomal cluster emerging as one of the most upregulated pathways in ascites after therapy. We added this information into the revised manuscript.

Figure 21 - Results of the KEGG enrichment analysis of proteins whose abundance increased at least 2 times in ovarian cancer ascites after chemotherapy compared to ovarian cancer ascites before therapy without excluding proteins identified in cirrhosis ascites (p-values are indicated).

Revised manuscript (Page 5, Lines 165-167, Extended Figure S1E).

69. The author indicated that one of their major findings is detecting spliceosomes in the extracellular space. However, other groups have already reported that apoptotic cancer cells can secrete apoptotic extracellular vesicles enriched with various spliceosomes, to promote survival cell proliferation and therapy resistance (Cancer Cell, PMID: 29937354). It is immediately clear how this study provides novel insights.

Response: We appreciate the Reviewer's comments and understand the concerns regarding the novelty of our study. It's important to note that the paper referenced [PMID: 29937354] was published with the participation of our group. While there are some similarities between these two studies, the present manuscript focuses on a different cancer type and provides a more in-depth exploration of the molecular mechanisms underlying this phenomenon, specifically with a focus on the development of chemoresistance. We have adjusted the discussion section to emphasize the uniqueness and novelty of this particular study.

70. The authors utilized the spectral count values to quantify the proteins identified from the ascitic fluids. However, in the majority of proteomic studies, the intensity-based absolute quantification (iBAQ) algorithm or label-free quantification algorithm (LFQ) were conducted for label-free protein quantification (Nature Communications, PMID: 30258067, Cell Metabolism, PMID: 29320704). I would recommend the authors to use iBAQ or LFQ values instead of spectral count values for protein quantification, to ensure the utility of these data by public.

Response: MaxQuant offers several quantification strategies, notably LFQ intensity and iBAQ protein intensity. LFQ intensity is represented by a normalized intensity profile that is generated according to the algorithm of MaxQuant. iBAQ protein intensity represents the sum

of peptide intensities matching a protein, divided by the number of theoretically observable peptides. A spiked-in standard of proteins with known absolute molar amounts is needed for accurate adjustment to represent absolute protein amounts.

Spectral counting is a straightforward label-free quantitation strategy [PMID: 27748581, PMID: 27546623]. We utilized the spectral count values to quantify the proteins identified in ascites because using multiple search engines (in our case Mascot and X!Tandem integrated then in Scaffold) can expand the number of identified proteins (the union of the data) and validate protein identifications (the intersection of the data) [PMID: 25346847].

According to the Reviewer's suggestion, we also conducted label-free quantification (LFQ) of our LC-MS/MS data using MaxQuant program. In total, we identified 1334 and 2187 proteins in all ascitic samples using MaxQuant and Scaffold, respectively (Figure 22A). 81% of proteins identified using MaxQuant overlapped with proteins identified with Scaffold. We also performed KEGG enrichment analysis of proteins identified with these programs exclusively in ascites after therapy (Figure 22B-C) and observed the same trends. In this regard, we decided to leave the data calculated in the Scaffold program in the manuscript. However, we also deposited all our raw LC-MS/MS data obtained in this study in PRIDE to allow researchers to analyze the data using any convenient program.

Figure 22 - (A) Venn diagram showing intersection of all proteins identified in ovarian cancer ascites using Scaffold or Mascot programs. **(B)** Venn diagram showing intersection of proteins identified in ascites before and after therapy using Scaffold or Mascot programs. **(C)** Results of the KEGG enrichment analysis of proteins which were identified only in ovarian cancer ascites after therapy using Scaffold or Mascot programs (p-values are indicated), (not included into final manuscript).

71. The authors identified 63 common spliceosomal proteins that may contribute to intercellular communication. Its is an interesting finding, yet need to be verified. Given that

so many proteomics studies have been published over the past few years, the author should compare and contrast the data collected here with public database.

Response:

To the best of our knowledge, the majority of studies focusing on proteome analyses of cancer cell secretomes have been conducted using cells under normal conditions, devoid of any treatments. Extensive data describe in vitro secretomes of sensitive and resistant cancer cells, mesenchymal and epithelial cancer cells, and cells with different metastatic potential.

While many studies reveal that various stress conditions such as hypoxia, irradiation, chemotherapy, etc., significantly increase the secretion of extracellular vesicles by dying cancer cells compared to untreated ones, proteomic profiling of these secretomes is often lacking. We have found 6 available datasets of proteome analyses of secretomes from different cell lines under different stress conditions. Three datasets representing secretomes of dying cancer cells, including lymphoma U937 cells [PMID: 31995763], melanoma cells [PMID: 32183388], and glioblastoma cells [PMID: 29937354], demonstrated an increased secretion of spliceosomal proteins to the extracellular space after treatment. Notably, three datasets representing secretomes from dying normal cells, such as pigment epithelial cells (RPE) [PMID: 31659173], endothelial cells [PMID: 28849662], or fibroblasts [PMID: 31945054], did not exhibit upregulated secretion of spliceosomal proteins. We added this information about comparison of our data with previously published data in the “Discussion” section. It is crucial to note that in most cases, the identification of proteins in the secretomes was limited, with no more than 20 proteins showing differential abundance between control and therapy-induced secretomes [PMID: 24729417].

Revised manuscript (Page 16, Lines 514-517, Supplementary Table S4, Sheet 4).

72. The authors utilized overexpression systems to verify the effects of SNU13 and SYNCRIP. However, the author should also include knockdown experiments too.

Response: In accordance with the Reviewer’s comment, we established SKOV3 cell sublines with knockdown of *SYNCRIP* or *SNU13* and showed these cell lines were more sensitive to cisplatin compared to control cell line (Figure 23). We also performed RNAseq analysis of SKOV3 cells overexpressing *SYNCRIP*, *SNU13*, or an empty vector before and after cisplatin treatment and revealed distinct responses of control and *SYNCRIP* or *SNU13* overexpressing cells to cisplatin. Enrichment analysis demonstrated that after treatment, both *SYNCRIP* and *SNU13* overexpressing cells had increased expression of genes involved in DNA repair compared to control cells. These findings suggest that exogenous splicing factors may promote DNA damage response, at least partially, by modulating the expression of DNA repair genes.

Figure 23 - (A) Western blotting analysis of SKOV3 cells stably expressing shRNAs against *SNU13* (SKOV3-SNU13 KD), *SYNCRIP* (SKOV3-SYNCRIP KD) or control non-target shRNA (SKOV3-NTC). **(B)** Flow cytometry analysis of caspase 3/7 activity and SYTOX staining of SKOV3 cells stably expressing shRNAs against *SYNCRIP*, *SNU13* or control non-target shRNA (SKOV3-NTC) and treated with 40 μ M of cisplatin for 24 hours.

Revised manuscript (Page 13, Lines 425-429, Extended Fig. 7C-D and Fig. 7E, Figures 6B-C, Supplementary Table S6).

73. The authors indicated that U12 snRNA, U6atac snRNA caused tumor cell proliferation. It is an important finding, yet, the authors should conduct some functional experiments to elucidate how these snRNAs impact the cell proliferation process, and finally lead to aggressive prognosis.

Response: Consistently with our RNAseq and proteomic data, among the differentially enriched proteins in cells transfected with both U12 and U6atac snRNAs, significant differences were observed in the cell cycle regulation clusters, including proteins involved in the progression through the G1 phase and DNA repair such as RBBP8, CCNB1, AURKB,

CDK4, ANAPC4/5, CHEK1, XPC etc. Thus, it has been shown that upregulated RBBP8 and CDK4 facilitate G1/S transition and promote the self-renewal and proliferation of cancer cells [PMID: 31636387; PMID: 24971465]. CCNB1 is activated by CHEK1 and also stimulate proliferation of cancer cells [PMID: 24971465].

Based on these data, we next explored whether increased levels of snRNAs would indeed affect cell cycle progression. FACS analysis demonstrated that cancer cells transfected with U12 and U6atac snRNAs spent less time in G1 phase, which is often accompanied by a high proliferation rate [PMID: 22802651] (Figure 24A). To validate these results, we overexpressed U12 and U6atac snRNAs in SKOV3 cells using lentiviral constructs and showed that both U snRNAs substantially increased the proliferation of SKOV3 cells (Figure 24B).

Figure 17 - (A) Cell cycle analysis of SKOV3 cells 48 h after transfection with 10 nM U6atac snRNA, U12 snRNA, or GFP89. (B) xCELLigence proliferation assay of SKOV3 cells overexpressing U6atac snRNA (left panel), U12 snRNA (right panel) or control GFP mRNA fragments (with the corresponding promoters: U2 or U6, respectively). SKOV3 cells were seeded in 96-well E-plates for xCELLigence assay monitoring impedance (cell index).

Revised manuscript (Page 14, Lines 451-473, Figure 6H, Extended Fig. 8I).

Minor points:

74. In figure 1E, the colors that presented cytometry results before and after therapy are marked incorrectly.

Response: We apologize for poor representation of our data. In accordance with Reviewers' comments, we rewrote this part of the manuscript, since we obtained substantial amount of an additional data during revision of the manuscript.

75. In line 125-127, the authors indicated the activation of EMT as well as the MYC signaling pathway, but do not describe in which group do these processes and pathways been activated?

Response: We apologize for inaccurate description of our data. In accordance with Reviewers' comments, we rewrote this part of the manuscript, since we obtained substantial amount of an additional data during revision of the manuscript.

Revised manuscript (Page 5, Lines 140-143): "Enrichment analysis of differentially expressed genes and proteins unveiled prominent upregulation DNA repair and cell cycle regulation pathways in tumor cells incubated with EOC ascitic fluids after chemotherapy (Fig. 1F-G)."

76. The extended figure 5H does not exist.

Response: Thank you for pointing this out. We corrected it in the revised version of the manuscript - Extended Figure 5C.

77. The figure 4F does not contain IL6.

Response: We sincerely apologize for this inaccuracy. We corrected the text in accordance with our proteomic data.

Revised manuscript (Pages 9-10, Lines 305-307): "In contrast, in fibroblast secretomes, most of these proteins, including GDF15 and CTGF, were exclusively detected after chemotherapy (Extended Fig. 4E)."

78. The writing of the paper should be improved. The authors do not clearly describe how specific analyses were performed. Specific methods should be cited in the main results section.

Response: We rewrote "Methods" section and several parts in the "Results" section, added additional information about analyses where needed and included a new item in the revised version of the manuscript. We also performed English editing of the manuscript.

79. The figure legends should be written with more details.

Response: We apologize for the insufficient information. We added more details in the Figure legends.

Reviewer #6 (Remarks to the Author): Expert in extracellular vesicles

In this paper, the authors show that spliceosomal components encapsulated in EV emanating from cisplatin treated cancer cells favors cell survival of recipient cells treated with aforementioned donor EVs. Experimentally, the authors used multi-omics, in vitro and in vivo biology to support their conclusion. Experimental approach is adequate and well described. Discussion is well balanced and data do not seem to be overinterpreted. Data would be of interest for cell biology and cancer biology, and reinforce the pro-tumoral role of EVs.

We highly appreciate thoughtful review of our manuscript and insightful suggestions to strengthen the data pertaining to the role of extracellular vesicles (EVs) in our study. We have carefully considered Reviewer's recommendations and have taken steps to address them effectively.

80. However, I believe that the data could be even stronger if the proof that the signal carried by EVs is responsible for the observed phenotypes. A significant improvement would be to

isolate EVs through sucrose flotation, directly demonstrate that spliceosome candidates are within the EV using well established protease protection assay et clearly assess the quality of the EV prep. Such isolated EVs are expected to trigger similar phenotype, and may even enhance it if the preparation is purer. More direct demonstration would better support direct role of EVs.

Response: We encountered challenges when considering the use of sucrose, primarily due to potential toxicity concerns associated with unwashed sucrose affecting recipient cells. Moreover, each washing step during extracellular vesicle (EV) isolation introduces a significant risk of substantial loss in both the quantity and quality of exosomes. This loss becomes particularly critical when dealing with exosomes that must maintain their biological functionality and structural integrity for subsequent incubation with living cells. Additionally, minimizing the handling time of vesicles was a priority to mitigate potential damage.

In response to the Reviewer's query regarding whether the signal conveyed by EVs is indeed responsible for the observed phenotypes, we conducted a proteinase K assay. Our results unequivocally demonstrate that spliceosomal proteins were shielded by the vesicle membrane in secretomes (Figure 25). These proteins remained intact and were only susceptible to digestion by proteinase K only when detergent was added, providing further support for our conclusion that they are enclosed within the EVs.

Figure 25 - Results of proteinase K protection assay. Western blotting analysis of secretomes from SKOV3 cells treated (TIS) or untreated (CtrlS) with cisplatin and then samples were treated or untreated with different concentrations of proteinase K in presence or absence of detergent.

In addition to the protease protection assay, we conducted a flow cytometry analysis of extracellular vesicles obtained from cells transfected with pTagGFP2-SNU13 or pTagGFP2-SYNCRIP (Figure 26). These experiments involved immune-selection with Exosome-Human CD9 Flow Detection Reagent. The results of this analysis unequivocally demonstrated the presence of fluorescent SNU13 and SYNCRIP proteins within the extracellular vesicles derived from secretomes of dying cancer cells. This provides direct evidence that these spliceosomal components are indeed encapsulated into EVs upon apoptosis induction.

Figure 26 - Flow cytometry analysis of magnetic beads coated with anti-CD9 antibody after immunoprecipitation of extracellular vesicles from therapy-induced secretomes of donor SKOV3 cells. Donor SKOV3 cells overexpressing SRSF4-RFP, SYNCRIP-GFP or SNU13-GFP were treated with 40 μ M cisplatin for 7 h. and then medium were replaced with fresh medium not containing cisplatin and incubated additional 41 h. High-fluorescent events are marked with blue color.

Revised manuscript (Page 9, Lines 283-287, Extended Fig. 4F; Page 11, Lines 357-363, Extended Fig. 5E).

REVIEWERS' COMMENTS

Reviewer #1 (Remarks to the Author):

The Authors have addressed my comments and the manuscript has substantially improved.

Reviewer #2 (Remarks to the Author):

Dear Authors,

I have reviewed the new version and the replies sent to my comments. I consider that you made a significant effort to give a satisfactory response to my comments and that the new version is much better organized both in the text and in the presentation of the results.

Reviewer #3 (Remarks to the Author):

EDITORIAL NOTE

Please note that Reviewer #2 assessed your response to Reviewer #3. With respect to this report, they have confidentially shared the following pending comments, which we have paraphrased here and would ask you to address in your revision:

* In figure 10 of your response letter, only one immunofluorescence result is shown for CA125 and EpCAM, not from several primary cultures. Each image seems to correspond to a different sample; instead, it would be better to show the same analysis for both markers in at least three different samples. EDITORIAL NOTE: we would suggest including this as a supplementary figure.

Reviewer #4 (Remarks to the Author):

The authors have performed extensive revisions to this manuscript. My concerns were addressed adequately and I have no further comments.

Reviewer #5 (Remarks to the Author):

The authors have substantially revised their manuscript and addressed the issues that I raised.

Reviewer #6 (Remarks to the Author):

The authors satisfyingly addressed my previous comments. In addition the revised ms has been significantly improved on many others point highlighted by other Reviewers.

Point-by-point Response to Reviewer's Comments

Please note that Reviewer #2 assessed your response to Reviewer #3. With respect to this report, they have confidentially shared the following pending comments, which we have paraphrased here and would ask you to address in your revision:

** In figure 10 of your response letter, only one immunofluorescence result is shown for CA125 and EpCAM, not from several primary cultures. Each image seems to correspond to a different sample; instead, it would be better to show the same analysis for both markers in at least three different samples. EDITORIAL NOTE: we would suggest including this as a supplementary figure.*

We appreciate the Reviewers' feedback. We provided our immunofluorescence images of three different primary cultures of ovarian cancer cells stained for CA125, EpCAM, and CD44 in Supplementary Fig. 1A and added the sentence in the main text: "Established primary cultures were stained for ovarian cancer markers CA125, EpCam, and CD44 (Fig. 1A, left panel, Supplementary Fig. 1A)".